# Was Australia a sink or source of CO$_2$ in 2015? Data assimilation using OCO-2 satellite measurements

Yohanna Villalobos[1,2,5], Peter J. Rayner[1,2,3], Jeremy D. Silver[1,4], Steven Thomas[1], Vanessa Haverd[5,†], Jürgen Knauer[5], Zoë M. Loh[6], Nicholas M. Deutscher[7], David W.T. Griffith[7], and David F. Pollard[8]

[1]School of Geography, Earth and Atmospheric Sciences, University of Melbourne, Australia
[2]ARC Centre of Excellence for Climate Extremes, Sydney, Australia
[3]Climate & Energy College, University of Melbourne, Australia
[4]School of Mathematics and Statistics, University of Melbourne, Australia
[5]CSIRO Oceans and Atmosphere, Canberra, 2601, Australia
[6]CSIRO Oceans and Atmosphere, Aspendale, Victoria 3195, Australia
[7]School of Earth, Atmospheric and Life Sciences, University of Wollongong, Wollongong, Australia
[8]National Institute of Water and Atmospheric Research Ltd (NIWA), Lauder, New Zealand.
[†]Deceased, 19 January 2021.
**Correspondence:** Yohanna Villalobos (Yohanna.Villaloboscortes@csiro.au)

**Abstract.**

In this study, we present the assimilation of data from the Orbiting Carbon Observatory-2 (OCO-2) (land nadir and glint data, version 9) to estimate the Australian carbon surface fluxes for the year 2015. To perform this estimation, we used both a regional-scale atmospheric transport-dispersion model and a four-dimensional variational assimilation scheme. Our results

suggest that Australia was a carbon sink of -0.41 $\pm$ 0.08 PgC y$^{-1}$ compared to the prior estimate 0.09 $\pm$ 0.20 PgC y$^{-1}$ (excluding fossil fuel emissions). Most of the carbon uptake occurred in northern Australia over the savanna ecotype and in the western region over areas with sparse vegetation. Analysis of the enhanced vegetation index (EVI) suggests that the majority of the carbon uptake over the savanna ecosystem was due to an increase of vegetation productivity (positive EVI anomalies) amplified by an anomalous increase of rainfall in summer. Further to this, a slight increase of carbon uptake in Western

Australia over areas with sparse vegetation (the largest ecosystem in Australia) was noted due to increased land productivity in the area caused by positive rainfall anomalies. The stronger carbon uptake estimate in this ecosystem was partially due to the land surface model (CABLE-BIOS3) underestimating the gross primary productivity of the ecosystem. To evaluate the accuracy of our carbon flux estimates from OCO-2 retrievals, we compare our posterior concentration fields against the column-averaged carbon retrievals from the Total Carbon Column Observing Network (TCCON) and ground-based *in-situ*

monitoring sites located around our domain. The validation analysis against TCCON shows that our system is able to reduce bias mainly in the summer season. Comparison with surface *in situ* observations was less successful, particularly over oceanic monitoring sites that are strongly affected by oceanic fluxes and subject to less freedom by the inversion. For stations located far from the coast, the comparison with *in situ* data was more variable, suggesting difficulties matching the column-integrated and surface data by the inversion, most likely linked to model vertical transport. Comparison of our fluxes against the OCO-2

model intercomparison (MIP) was encouraging. The annual carbon uptake estimated by our inversion falls within the ensemble of the OCO-2 MIP global inversions and presents a similar seasonal pattern.

## 1 Introduction

Australia's carbon budget has been investigated by several large scale global ecosystem models (Sitch et al., 2015, Carbon cycle model intercomparison project (TRENDY)) and by the Regional Carbon Cycle Assessment and Processes project (RECCAP) (Haverd et al., 2013a). However, although they have contributed to a more refined knowledge of the Australian carbon cycle, the estimated fluxes still diverge significantly. In the latest RECCAP report (Haverd et al., 2015), the net biome production (NBP) estimate for the country was a net carbon source of $59 \pm 35$ $TgC\,y^{-1}$ between 1990–2011. A large component of the uncertainty in this carbon budget was attributed to the estimate of net primary productivity (NPP) over grassland (Haverd et al., 2013b), with a large contribution to the land cover type they used to force their simulations (e.g., The Advanced Very High Resolution Radiometer AVHRR (1990–2006) (Donohue et al., 2009)) and The Moderate Resolution Imaging Spectroradiometer (MODIS) (2000–2011). Given this uncertainty, it is essential to bring any other observations we have to bear on the Australian carbon balance.

Data assimilation (also called atmospheric transport inversion), along with an increase of remotely-sensed concentrations of carbon dioxide $CO_2$ data, have been revolutionary for quantifying land-ocean-atmosphere $CO_2$ flux exchange in the last decade. Satellite data from the Greenhouse Gases Observing Satellite (GOSAT) (Yokota et al., 2009), launched in 2009, and the Orbiting Carbon Observatory-2 (OCO-2) (Eldering et al., 2017) launched in 2014 have been used by several studies (Basu et al., 2013; Chevallier et al., 2014; Deng et al., 2014; Maksyutov et al., 2013; Crowell et al., 2019) to infer carbon $CO_2$ sources and sinks at continental scales. Few regional studies have been performed and "none in Australia" while the global inversions show large differences for this region. For example, a study based on six satellite-based inversions using GOSAT (Chevallier et al., 2014, Fig.1) shows that Australia was a carbon sink (-0.7 $PgC\,y^{-1}$) for 2010. For the same year, Basu et al. (2013) inferred it to be a net carbon source (0.4 $PgC\,y^{-1}$)[1].

The accuracy of flux inversions using global atmospheric transport models has been the subject of discussion due to errors related to modelled transport (Chevallier et al., 2014; Basu et al., 2018). Transport model error in global inversions often emerges because inversions are run at horizontal resolutions of $1°$– $5°$. Increasing the model resolution (Law et al., 2004), potentially reduces the representation errors found in global-scale models. Regional-scale inversions arose about a decade ago. They rely on mesoscale transport models (run at $1°$ down to $10\,km$ resolution); for example, Broquet et al. (2011) performed a regional-scale variational inversion of the European biogenic $CO_2$ fluxes at $50\,km$ resolution. Another example of regional-scale inverse modelling is found in Villalobos et al. (2020), who performed an inversion at $81\,km$ resolution over Australia. Finer resolution models have the potential to be more successful, since they can offer a better representation of surface $CO_2$

---

[1]In this paper we adopt the atmospheric convention where a negative flux indicates removal from the atmosphere (a sink, hereafter quoted with a negative sign), and a positive value indicates an addition to the atmosphere (source).

fluxes and variability, as well as a better simulation of the processes driving high-frequency variability of transport (Schuh et al., 2010).

Australia has recently been subject to attention from the global carbon cycle community due to a large terrestrial carbon sink anomaly recorded in 2011 (Poulter et al., 2014). Poulter et al. (2014) found that Australia's flux anomaly was -0.66 for 2011 (relative to the 2003–2012 mean). Trudinger et al. (2016) also found a similar carbon sink anomaly for this period (ranging

between -0.40 to -0.61 PgC y$^{-1}$). These studies suggest that Australia's ecosystems might act as strong sinks of $CO_2$ in the future during extreme wet periods. However, the efficiency and the spatial distribution of these carbon sinks remain largely uncertain (Ma et al., 2016). Some studies (i.e., Ma et al., 2016) found that the anomalous carbon uptake recorded in Australia in 2011 rapidly diminished in the following year ($\sim 0.08$ PgC y$^{-1}$), suggesting that semi-arid ecosystem can act as carbon sink in the short term but not over longer periods compared to tropical forest ecotypes. An important unanswered question in

carbon cycle research remains regarding the carbon sink strength of semi-arid ecosystems in non-wet years.

In this study, we present a regional inversion to infer $CO_2$ fluxes over Australia for 2015 based on the Community Multiscale Air Quality (CMAQ) model and OCO-2 satellite retrievals. In 2015, Australia was affected by the El Niño Southern Oscillation (ENSO), and although some parts of the continent were impacted by rainfall deficiency, other regions such as northern and south-eastern Australia rainfall was above average (Annual climate statement, Bureau of Meteorology, 2015).

This manuscript is structured into five sections. Section 2 describes the flux inversions system and the datasets used. Section 3 presents the main results of the Australian carbon budget, as well as an analysis of the enhanced vegetation index (EVI) and rainfall anomalies, and a comparison between our posterior $CO_2$ concentration and the Total Carbon Column Observing Network (TCCON) and *in-situ* measurements. In Section 4, we present a discussion of our results, as well as a comparison of our optimized fluxes against the ensemble mean of nine different global inversions that participate in the OCO-2 model

intercomparison (MIP). In Section 5 we summarise our findings.

## 2 Methodology and data

To estimate the Australian $CO_2$ surface fluxes for 2015 we followed the same four-dimensional variational assimilation scheme described in Villalobos et al. (2020). In this section, we will present a brief description of the system, and an update of all changes we made to the data used for our inversion.

### 2.1 Bayesian Inverse system

Finding the optimal value ($x^a$) of the $CO_2$ flux estimates involves identification of the best fits between both observations ($y$) and a prior (or background) estimate ($x^b$) of these fluxes (Ciais et al., 2010; Rayner et al., 2019). Using Bayes' theorem and under the hypothesis of unbiased Gaussian-distributed errors of $x^b$ and $y$, the best estimate of $x^a$ (likelihood maximum

a posteriori) is equivalent to finding the minimum of the cost function $J(\boldsymbol{x})$ shown in Eq. 1. Notation in this study follows Rayner et al. (2019).

$$J(\boldsymbol{x}) = \frac{1}{2}\left[(\boldsymbol{x}-\boldsymbol{x}^b)^T\mathbf{B}^{-1}(\boldsymbol{x}-\boldsymbol{x}^b)\right] + \frac{1}{2}\left[(\mathbf{H}(\boldsymbol{x})-\boldsymbol{y})^T\mathbf{R}^{-1}(\mathbf{H}(\boldsymbol{x})-\boldsymbol{y})\right] \tag{1}$$

In Eq. 1, $\mathbf{H}$, represents the application of the forward model and the "observation operator", which allows us to map the model variables (e.g., fluxes) to observations. $\mathbf{R}$ represents the error covariance matrix of the observations $\boldsymbol{y}$, including the transport model error. $\mathbf{R}$ is defined as a diagonal matrix (details Section 2.5). $\boldsymbol{x}$ represents the control vector of unknowns. $\boldsymbol{x}$ includes not only $CO_2$ surface fluxes, but also initial and boundary conditions (details Section 2.2). $\mathbf{B}$ is the associated error covariance matrix of $\boldsymbol{x}^b$, boundary and initial concentrations, and includes off-diagonal terms. In these off-diagonal values, we only include spatial and non-temporal correlations of the prior fluxes (details of the structure of the prior error covariance matrix is found in Section 2.4 in Villalobos et al. (2020)).

We calculate the minimum of $J(\boldsymbol{x})$ by an iterative process, and not by an analytical expression. This numerical problem requires the value of the cost function gradient $\nabla_{\mathbf{x}}J(\mathbf{x})$.

$$\nabla_x J = \mathbf{B}^{-1}(\boldsymbol{x}-\boldsymbol{x}^b) + \mathbf{H}^T(\mathbf{R}^{-1}\left[\mathbf{H}(\boldsymbol{x})-\boldsymbol{y})\right]) \tag{2}$$

We compute $\mathbf{H^T}$ using the adjoint of the CMAQ model (version 4.5.1; Hakami et al., 2007). We can see in Eq. 2 that $\mathbf{H^T}$ is applied to the vector $\mathbf{R}^{-1}(\mathbf{H}(\boldsymbol{x})-\boldsymbol{y})$, which is often called the "adjoint forcing", and represents the error-weighted differences between the forward model and the observed concentrations. Applying the adjoint model to the adjoint forcing, running backward in time from $t_{i-1}$ to $t_0$, allows us to construct the gradient of the cost function, $\nabla_x J(\boldsymbol{x})$. The algorithm that our inverse system uses to optimize the $J(\boldsymbol{x})$ is the limited-memory BFGS (L-BFGS-B), implemented in the `scipy` python module (Byrd et al., 1995). Figure 1 shows a simplified version of how our inversion system works to find the optimal values of $CO_2$ surface fluxes.

The error statistics of $\boldsymbol{x}^a$ are embodied in the posterior error covariance matrix ($\mathbf{A}$). In this study, $\mathbf{A}$ was computed by a series of observing system simulation experiments (OSSEs) carried out by Villalobos et al., 2020, Section 2.4. However, here we adjusted the prior and observation uncertainties assumed in Villalobos et al. (2020) by a factor of 1.2. We made this adjustment to satisfy the theoretical assumption in the variational optimization, which indicate the value of the cost function in its minimum has to be approximately equal to half of the number of observations (for more details see Section 3.1). In general, errors assumed in the inversion are not gaussian and independent but rather have errors correlated in time and space (including flat biases) that render the statistical assumptions made in deriving the estimation method invalid and lead to a higher cost function than expected. A description of how the prior and observation uncertainties were assumed in our study is found in Sections 2.3 and 2.5. Appendix D (Figs. D1 and D2) show the spatial distribution of the prior and posterior that we reference in this study.

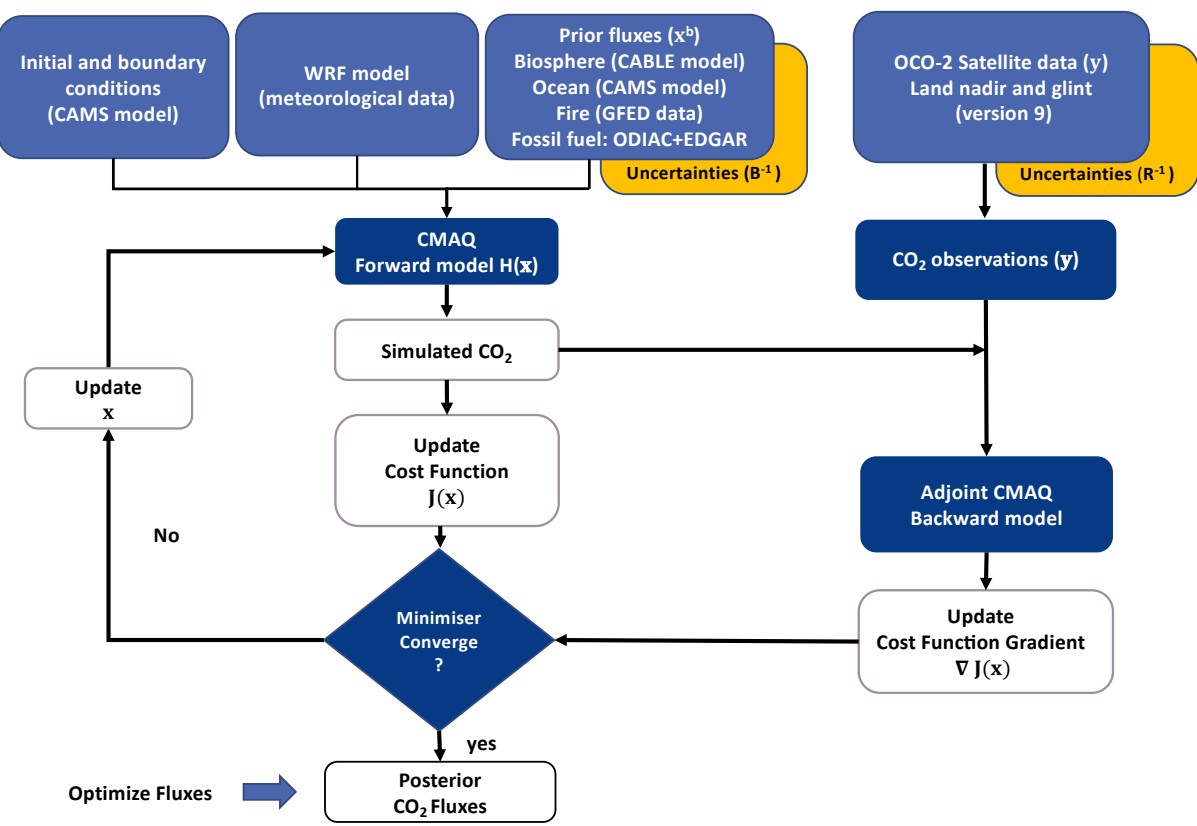

**Figure 1.** A simplified diagram of the four-dimensional variational data assimilation we used to estimate $CO_2$ surface fluxes over Australia.

## 2.2 Defining the Control Vector

In our data assimilation system, we solve for monthly-average surface fluxes at 81 km grid-cell scale resolution as the multipliers of the principal eigenvectors of the prior error covariance matrix $\mathbf{B}$, computed as $\mathbf{W^T w}^{-1/2}(\boldsymbol{x} - \boldsymbol{x}^b)$, where $\mathbf{W}$ was defined as the matrix of eigen-vectors and $\mathbf{w}$ as a diagonal matrix of corresponding eigenvalues (Villalobos et al., 2020, Section 2.2). In order to avoid the impact of the initial conditions (ICs) and boundary conditions (BCs) on our assimilated fluxes, we also solved them within the control vector $\boldsymbol{x}$. We did not optimize them in the same way as the fluxes in order to not increase

the control vector size, so we treat the unknowns related to the BCs and ICs as scaling factors of the emissions added to the CMAQ model. Lateral BCs were solved as eight boundary regions divided by the upper and lower boundary areas within the CMAQ domain (south, east, north, and west). In Fig. 2, we provide a representation of these boundaries. In this figure, we can see that our study domain not only covers the Australian continent (AUS) but additionally other countries such as Indonesia (IND), Papua New Guinea (PNG) and New Zealand (NZ). The extension of this domain was created as an extra precaution to

minimize the influence of the boundaries over Australia.

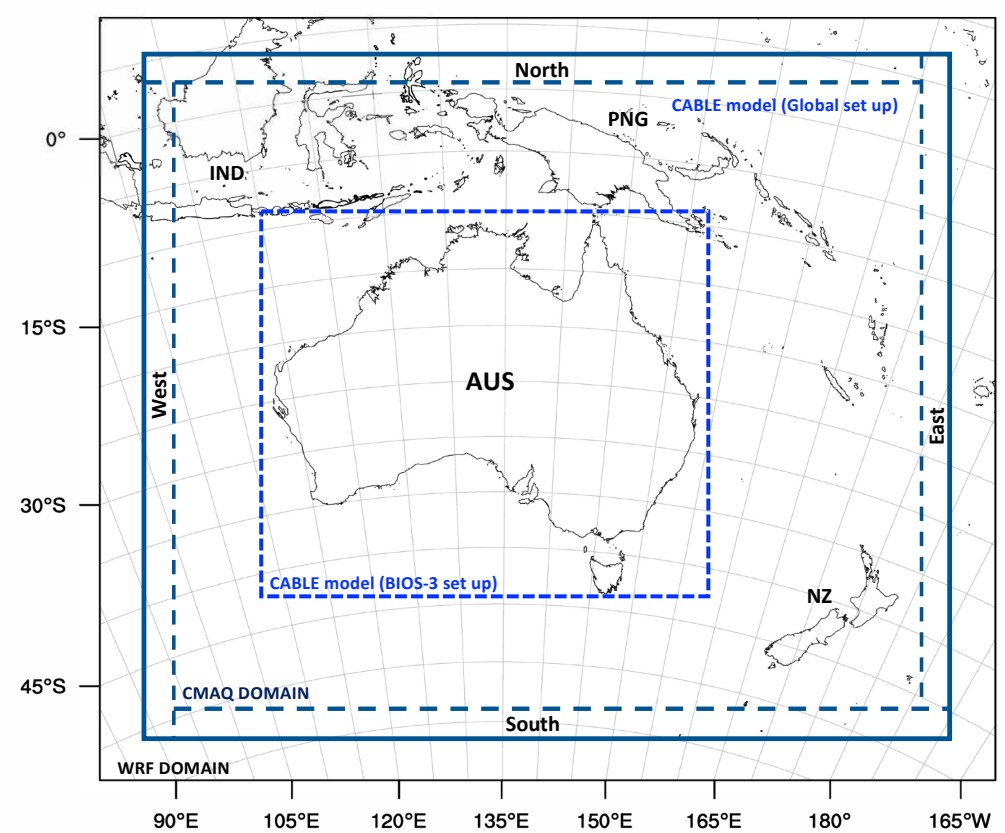

**Figure 2.** Representation of the horizontal WRF domain (black rectangle) and CMAQ model domain (dark blue rectangle). Boundary components (south, east, north, and west) are represented as between the outer domain of CMAQ domain and the dotted dark blue lines. Land biosphere emissions incorporated over Australia are represented by the small dotted blue lines (CABLE model in BIOS-3 set up). Outside this area, land biosphere emissions come from CABLE global product

Lower boundary layers were defined to cover from $\sigma = 1$ to $\sigma = 0.96$, which correspond (on average) to a pressure of 972.5 hPa, while the upper boundary layer was defined to cover from 972.5 up to 50 hPa. As mentioned before, our inversion system solves for these lateral boundary components, while surface fluxes are also being optimized. Boundary conditions are provided to the CMAQ model as daily averages, but we optimize them as monthly averages. BCs and ICs dataset were taken from the CAMS global $CO_2$ atmospheric inversion product (version v19r1) (Chevallier, 2019). Uncertainties for the initial condition were set at 1% (approximately 4 ppm), and the uncertainties in the lateral boundary conditions were assumed as the standard deviation ($1\sigma$ uncertainty) of CAMS concentration data in the perimeter of the boundary.

## 2.3 Prior information and its uncertainties

We updated the prior $CO_2$ fluxes described in Section 2.4 of Villalobos et al. (2020). Biosphere carbon fluxes were derived using a modified version of the Community Atmosphere-Biosphere Land Exchange model (CABLE) (Haverd et al., 2018), which was forced by Australian regional drivers and observations (BIOS3 set-up). CABLE consists of a biophysical core: the Carnegie-Ames-Stanford Approach, Carbon, Nitrogen, Phosphorus (CASA-CNP) biogeochemical model (Wang et al., 2010), the POP module for woody demography and disturbance-mediated landscape heterogeneity (Haverd et al., 2013d), and a module for land use and land management (POPLUC; Haverd et al. (2018)). For our case, Haverd (2020) ran the CABLE model in the BIOS3 set-up (hereafter CABLE-BIOS3) at a resolution of 0.25 degree. We calculated 3-hourly biosphere $CO_2$ fluxes by combining two data sets: daily net ecosystem exchange (NEE) fluxes with 3-hourly Gross Primary Production (GPP). Given that the BIOS3 product did not cover our whole CMAQ model domain, we also incorporated monthly biosphere fluxes from CABLE-POP global simulations as shown in Fig. 2. These CABLE-POP simulations were used in the Carbon cycle model intercomparison project (TRENDY version 8) for the 2019 global carbon budget (Friedlingstein et al., 2019). Biosphere flux uncertainties in our system were assumed to be equal to the Net Primary Production (NPP) simulated by CABLE, with a ceiling of 3 $\mathrm{gC\,m^{-2}\,day^{-1}}$ following Chevallier et al. (2010).

Anthropogenic fluxes were created by the combination of two different inventory data sets: the Open-source Data Inventory for Anthropogenic $CO_2$ (ODIAC) (Oda et al., 2018) and the Emissions Database for Global Atmospheric Research (EDGAR), version 5 (Crippa et al., 2020). The combination of these two anthropogenic inventories (each used to cover different source sectors) was necessary because the version of the ODIAC selected did not contain emissions from aviation. The EDGAR emissions combined with ODIAC were aviation climbing and descent, aviation cruise, and aviation landing and take-off datasets. Aviation emissions were also distributed across the vertical layers of the CMAQ domain. EDGAR is a gridded product with spatial resolution of $0.1° \times 0.1°$ with monthly temporal resolution. ODIAC (version 2019) is also a gridded product, which has a spatial resolution of $1 \times 1$ km. Monthly ODIAC fluxes were modified by incorporating a diurnal scale factor estimated by Nassar et al. (2013). The ODIAC data product selected did not include bunker emissions. Fossil-fuel carbon emission uncertainties were created by multiplying the emissions dataset with a factor of 0.44. This factor was calculated by a linear regression between the mean fluxes and the spread of an ensemble of 25 realizations of posterior emissions estimated by the fossil fuel data assimilation system (FFDAS) (Asefi-Najafabady et al., 2014).

Prior ocean fluxes were taken from the CAMS greenhouse gas flux inversion (version v19r1) (Chevallier, 2019). The prior fluxes that CAMS uses in its inversion also includes EDGAR emissions over the ocean; so we did not include this anthropogenic flux over the ocean to avoid double counting. We assumed that the ocean uncertainties were uniform, and set up a value of 0.2 $\mathrm{gC\,m^{-2}\,day^{-1}}$ over ocean, as in Chevallier et al. (2010). We also used monthly fire emissions from the Global Fire Emission Database GFED4.1 (van der Werf et al., 2017), which includes small fire emissions. Fire emission uncertainties were assumed as 20% of the biomass burning carbon emissions. All datasets mentioned above (terrestrial biosphere exchange, fossil-fuel, fires and ocean fluxes) were interpolated to the spatial resolution of the CMAQ model.

As described in (Villalobos et al., 2020, Section 2.4), we included spatial correlations into our prior error covariance matrix $\mathbf{B}$ following (Basu et al., 2013, Section 3.1.1). The correlation length between grid-points over land was assumed to be 500 km and over ocean 1000 km. We assume that fossil fuel uncertainties were not correlated, so we only use the diagonal values of the matrix. In our eigen-decomposition of $\mathbf{B}$, the eigen-spectrum (eigenvectors of the covariance matrix) retain 99% of the explained variance (eigenvalues).

## 2.4 Atmospheric transport model

The inversion was based around the CMAQ modelling system (version, v5.3) and its adjoint (version 4.5.1; Hakami et al., 2007). The CMAQ modelling system is an Eulerian (gridded) mesoscale Chemical Transport Model (CTM). We added $CO_2$ into the CMAQ model as an inert chemical species, whose concentration is determined by atmospheric transport, fluxes, initial and boundary concentrations. The CMAQ model was driven by meteorological fields from the Weather Research and Forecast model (WRF) Advance Research Dynamical Core WRF-ARW (henceforth, WRF) version V4.1.1 (Skamarock et al., 2008), which data was processed by the Meteorology-Chemistry Interface Processor (MCIP) version 4.2 (Otte and Pleim, 2010). WRF configuration details are shown in Table 1. Our WRF model was set up at a spatial resolution of 81 km with 32 vertical layers from the surface up to 50 hPa. The numerical simulation was carried out on a single domain (i.e., non-nested). WRF initial conditions were taken from the ERA-Interim global atmospheric reanalysis (Dee et al., 2011), which has a resolution of approximately 80 km on 60 vertical levels from the surface up to 0.1 hPa. Sea surface temperatures were obtained from the National Centers for Environmental Prediction/Marine Modeling and Analysis Branch (NCEP/MMAB). The WRF model was run with a spin-up period of 12 hours.

**Table 1.** Physics parameterisations used in WRF model setup

| Category | Selected schemes |
| --- | --- |
| Microphysics | Morrison double-moment (Morrison et al., 2009) |
| Short wave radiation | Rapid Radiative Transfer Model (RRTMG) scheme (Iacono et al., 2008) |
| Long-wave radiation | Rapid Radiative Transfer Model (RRTMG) scheme (Iacono et al., 2008) |
| Surface layer | Monin-Obukhov (Monin and Obukhov, 1954) |
| Land/water surface | The NOAH land-surface model and the urban canopy model (Tewari et al., 2007) |
| Planetary Boundary Layercs (PBL) | Mellor–Yamada–Janjic scheme (Janjić, 1994)) |
| Cumulus | The Grell-Devenyi ensemble scheme (Grell and Dévényi, 2002) |

## 2.5 OCO-2 satellite information and its uncertainties

We assimilated satellite data from OCO-2 level 2 (Lite file version 9) for 2015, which is distributed by the National Aeronautics and Space Administration (NASA) (OCO-2 Science Team/Michael Gunson, Annmarie Eldering, 2018). OCO-2 was launched in 2014, and since then has provided nearly global coverage of column-averaged dry air mole fraction of $CO_2$ that has been

used by several carbon cycle researchers to estimate surface carbon fluxes at global and regional scales. OCO-2 provides data in three modes: nadir, glint, and target mode. In nadir mode, OCO-2 instrument points straight down at the surface of the Earth (surface solar zenith angle is less than $85°$); in glint mode, OCO-2 instrument points to the bright glint spot on Earth where solar radiation is directly reflected off the Earth's surface (local solar zenith angle is less than $75°$); and target mode, the instrument is configured to scan about a particular point on the ground as it passes overhead. In this study, we used the combination of both land nadir and land glint observations (LNLG), because there are no systematic offsets between the two datasets (O'Dell et al., 2018). We also performed an inversion using the combination of land (nadir and glint) and ocean glint observations (LNLGOG). However, this inversion was treated as an independent experiment (see Appendix F, Table. F1), and the assimilated fluxes estimated by using LNLGOG were not included in our main results. We decided to not incorporate them because ocean glint retrievals still have undetermined biases (O'Dell et al., 2018) that may complicate or confound the Australia flux estimation. We discussed the impact of assimilation LNLGOG in the validation of our inversion with independent data (see Section 3.5 for more details). We only used OCO-2 retrievals with quality flag 0 and only soundings that were bias-corrected, as described by Kiel et al. (2019). The spatial distribution of OCO-2 soundings (LNLG and LNLGOG) across the CMAQ domain for 2015 are shown in Appendix C, Figs. C1 and C2 respectively.

Given that multiple OCO-2 soundings cross one grid-cell over the CMAQ domain, we had to average them before doing any comparison with the CMAQ model simulations. This averaging process was carried out in two steps. First, we averaged all OCO-2 soundings that fall within 1 second intervals, and then these 1-second averages were averaged again within the CMAQ vertical columns (approximately 11 seconds average) across 81 km $\times$ 81 km grid-cells. The 1-second weighted averaging process is described in detail in Section 2.3. of Villalobos et al. (2020). In summary, to obtain the uncertainties of these 1-second averaging processes, we considered three different forms of uncertainty calculation, similar to Crowell et al. (2019). First, we averaged OCO-2 uncertainties assuming that these were correlated in a 1-second span (uncertainties defined as $\sigma_s$). Second, given that the average of OCO-2 uncertainties is sometimes low because they neglect systematic errors, we also used the spread of the OCO-2 retrievals in the 1-second average (uncertainties defined as $\sigma_r$). Third, we also defined baseline uncertainties (defined as $\sigma_b$) for cases where the number of soundings was not enough to compute a realistic spread. The values for our baseline uncertainties were assumed to be 0.8 ppm over land and 0.5 ppm over ocean. Finally, we selected the maximum value between these three uncertainties ($\sigma_s$, $\sigma_r$, and $\sigma_b$). We also added (in quadrature) to this term 0.5 ppm as the contribution of the model uncertainty (defined as $\sigma_m$).

Solving the cost function shown in Eq. 1 requires convolving the vertical levels of the CMAQ model with the retrieval profile from OCO-2. For this, we used Eq. 3 derived by Connor et al. (2008) as follows:

$$x^m_{\text{CO}_2} = x^a_{\text{CO}_2} - \sum_j \boldsymbol{h}_j \boldsymbol{a}_{\text{CO}_2,j} \boldsymbol{x}_a + \sum_j \boldsymbol{h}_j \boldsymbol{a}_{\text{CO}_2,j} \boldsymbol{x}^m_j, \tag{3}$$

where, $x^a$ is the OCO-2 a priori, $\boldsymbol{h}$ is a vector of pressure weights, $\boldsymbol{h_j}$ is the mass of dry air in layer $j$ divided by the mass of dry air in the total column, $\boldsymbol{a}_{\text{CO}_2}$ is the averaging kernel of OCO-2, $\boldsymbol{x}_a$ is the OCO-2 a priori profile, and $\boldsymbol{x}^m$ is the simulated

profile from the CMAQ model. In our inversion system, the OCO-2 averaging kernel is defined on 20 pressure levels and we interpolate these to the CMAQ vertical levels.

### 2.5.1 TCCON measurements

To validate our posterior $CO_2$ CMAQ concentrations, we used ground-based remote sensing data from the Total Carbon Column Observing Network (TCCON) (Wunch et al., 2011). There are three TCCON stations in our domain (see Table 2 for
references and Fig. 3 for coordinate locations). A TCCON instrument is a Fourier Transform Spectrometer (FTS) developed to record direct solar spectra in the near-infrared spectral region. TCCON provides accurate and precise column-averaged concentrations of $CO_2$ and other greenhouse gases. This instrument represents the "gold standard" for surface-based remote-sensing estimates of the total-column concentration of these gases. Data from TCCON is widely used by carbon cycle researchers, in particular for global flux inversion (Byrne et al., 2020) and validation of satellite data products (such as from OCO-2). To
perform a quantitative comparison against CMAQ simulations, we averaged all the TCCON retrievals to create hourly average XCO2 values, which were consistent with the CMAQ hourly simulations. After calculating the average of these retrievals, we interpolated the TCCON column averaging kernels and TCCON a priori $CO_2$ profile to the CMAQ vertical levels. After the interpolation, we followed the equation Eq. 3 to compute the TCCON column-mixing ratios simulated by CMAQ. The statistical analysis of CMAQ model–TCCON differences was based on monthly mean concentration, which were calculated
by taking local time averages (10:00 a.m. $-$ 02:00 p.m.), where the solar radiation intensity is most stable (Kawasaki et al., 2012).

**Table 2.** Reference of the TCCON stations used in this work for evaluation of our inverse model system

| TCCON station | Reference |
| --- | --- |
| Darwin, Australia | Griffith et al. (2014a) |
| Wollongong, Australia | Griffith et al. (2014b) |
| Lauder, New Zealand | Sherlock et al. (2014) |

### 2.5.2 Ground-based *in-situ* measurements

Additional data sets used to validate our posterior concentrations were taken from four ground-based *in-situ* monitoring sites forming part of the Global Atmosphere Watch (GAW) Programme of the World Meteorological Organisation (WMO): Cape
Grim, Gunn Point, Burncluith and Ironbark. Coordinates of these locations are shown in Fig. 3. All these data sets were supplied by Loh (2019) at hourly temporal resolution. For the comparison with our model simulation, we used hourly data from these monitoring sites, but the monthly mean averaged data shown in Section 3.5.2 were calculated using local time averages between midday and 05:00 p.m.

    Measurements of atmospheric $CO_2$ concentration at the Gunn Point, Ironbark and Burncluith sites were made continuously
at high frequency ($\sim$0.3 Hz) using CSIRO Picarro cavity ring-down spectrometers (model G2301 at Gunn Point and Ironbark,

and G2401 at Burncluith) all with inlets placed at the height of 10 m. Details of the Ironbark and Burncluith installation are described by Etheridge et al. (2016), and are broadly similar to the installations elsewhere, including Gunn Point. Cape Grim also operates a Picarro G2301 analyser, with the inlet positioned at a height of 70 m.

The instrumental precision for these analysers is better than $\pm$ 0.1 ppm for $CO_2$ (Etheridge et al., 2014) and all measurements are calibrated to the WMO X2007 $CO_2$ mole fraction scale (Zhao and Tans, 2006), ensuring comparability between all measurements used.

Cape Grim is a significant monitoring station in the GAW Programme because it samples air with some of the least recent anthropogenic and terrestrial influence in the world, representing hemispheric background concentrations. These air masses, known as "baseline", have blown straight off the Southern Ocean and have often been used in modelling studies. However, in this study, we used all Cape Grim data because our inversion assimilates only data that was collected over land and carries terrestrial signals.

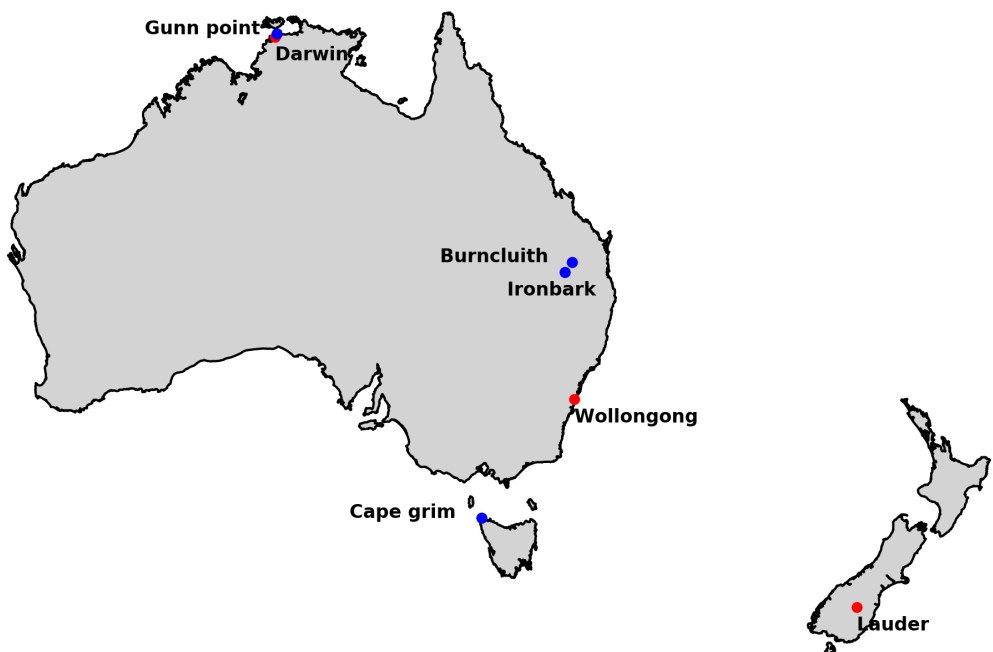

**Figure 3.** Total Carbon Column Observing Network (TCCON) and *in-situ* location sites. Red dots indicate TCCON locations. TCCON Darwin and Wollongong are located over Australia, while TCCON Lauder is located in New Zealand. Blue dots represent *in-situ* location around Australia (Gunn Point, Burncluith, Ironbark and Cape Grim).

### 2.6 Auxiliary data

In this study, we also use auxiliary data such as the Enhanced Vegetation Index (EVI), rainfall, and gross primary productivity (GPP) from the CABLE BIOS3 model to understand the difference between the prior and posterior fluxes over Australia in 2015.

#### 2.6.1 The Enhanced Vegetation Index (EVI)

To understand if there was higher than expected growth of vegetation across Australia in 2015, we evaluated the monthly EVI anomalies relative to the long-term mean from 2000-2014. We used the EVI from the Moderate-Resolution Imaging Spectroradiometer (MODIS) MOD13C1 version 6 data product from the NASA satellite Terra (Didan, 2014). This gridded EVI MODIS product has a temporal resolution of 16 days composite and 0.05-degree spatial resolution. We constructed the EVI anomalies by subtracting the long-term mean (2000-2014) for each month of 2015. The spatial distribution of the EVI anomalies is shown in Supplementary Fig. S1. EVI measures the greenness of vegetation and can be used as a proxy for monitoring the density or productivity of the vegetation biomass. EVI indices range from -0.2 to 1, where values less than 0 indicate a lack of green vegetation or arid areas. These monthly EVI MODIS products were regridded to the CMAQ grid to calculate the spatial correlation between prior and posterior flux differences (see Section 3.3).

#### 2.6.2 Australian Water Availability Project (AWAP)

Monthly rainfall data was taken from the Australian Water Availability Project (AWAP), Bureau of Meteorology (Jones et al., 2009). We used data for the period 2000–2015. AWAP data is obtained from a spline interpolation technique, which interpolates all available *in situ* rainfall observations onto grid-cells of 0.05 degrees (more details can be found in Jones et al. (2009)). AWAP rainfall anomalies were calculated in the same way as EVI anomalies, by subtracting their long-term mean from 2000 to 2014 (see Supplementary Fig. S2).

#### 2.6.3 MODIS Gross Primary Production (GPP)

We compared the MODIS Terra Gross Primary Productivity (GPP) MOD17A2H version 6 product for 2015 (Running et al., 2015) against CABLE-BIOS3 model GPP predictions (see Appendix E, Fig. E1). The MODIS GPP product has a spatial resolution of 500 m and a temporal resolution of eight days. The 8-day composite was averaged to monthly resolution and aggregated to the CMAQ grid for comparison with the CABLE-BIOS3 model GPP.

#### 2.6.4 MIP *in-situ* and OCO-2 satellite-derived fluxes

For validation, we compared our posterior Australian biosphere $CO_2$ flux estimates (excluding fossil fuel) against the ensemble monthly mean of nine OCO-2 satellite-based and *in situ* global inversions (see Section 4 for details). *In situ* and OCO-2 satellite-derived fluxes were consolidated by the OCO-2 Model Inter-comparison Project (MIP) (Crowell et al., 2019), which used the OCO-2 satellite version 7 product. In this study, we used the latest update of the MIP OCO-2 product (Peiro et al., 2021),

which uses OCO-2 data lite file version 9, with an improve bias correction approach (Kiel et al., 2019) compared to version 7 product (Crowell et al., 2019). Within the MIP design, *in situ* carbon flux estimates are derived by utilizing five collections in ObsPack observations (Masarie et al., 2014). A description of these data can be found in Peiro et al. (2021) (Section 2.3).

Table 3 shows a summary of these global inversions. This table shows that MIP global inversions were performed using different prior flux estimates, and the transport models were run at different spatial scales. Within MIP, prior estimates also include fossil fuel data, which was fixed and derived from ODIAC. With regard to fires estimates, they use different versions of the GFED dataset. Some of them used version 4, while other modellers use version 3. The main difference between these two datasets is that GFED version 3 does not include small fire burned areas.

**Table 3.** Summary of the configuration of the MIP OCO-2 (version 9) design.

| Acronym | Contact (Institutions) | Grid spacing Degree | Transport Model | meteorological fields | Prior Fluxes |
|---|---|---|---|---|---|
| AMES | Matthew Johnson (NASA Ames Research Center) | $4° \times 5°$ | GEOS-Chem | MERRA-2 | CASA-GFED4.1s |
| Baker | David Baker (Colorado State University) | $6.7° \times 6.7°$ | PCTM | MERRA-2 | CASA-GFED3 |
| CAMS | Frederic Chevallier (LSCE France) | $1.9° \times 3.75°$ | LMDz | ERA-Interim | ORCHIDEE |
| CMS-Flux | Junjie Liu (NASA JPL) | $4° \times 5°$ | GEOS-Chem | GEOS-FP | CARDAMOM |
| CSU | Andrew Schuh (Colorado State University) | $1° \times 1$ | GEOS-Chem | MERRA-2 | SIB4/ MERRA-2 |
| CT | Andy Jacobson (University of Colorado and NOAA GML) | $3° \times 2°$ $1° \times 1°$ | TM5 | ERA-Interim | CT2019 CASA GFED4.1s |
| OU | Sean Crowell (Colorado State University) | $4° \times 6°$ | TM5 | ERA-Interim | CASA-GFED3 |
| TM5-4DVAR | Sourish Basu (University of Maryland and NASA GMAO) | $2° \times 3°$ | TM5 | ERA-Interim | SIB-CASA |
| UT | Feng Deng University of Toronto | $4° \times 5°$ | GEOS-Chem | GEOS-FP | BEPS |

## 3 Results

### 3.1 Inversion Evaluation: Analysis of the residual between CMAQ simulation and OCO-2

As described in Eq. 1, the main purpose of the inversion is to optimize fluxes by minimizing the mismatch between the model simulation and observations. In order to evaluate the performance of the inversion, we compared the $CO_2$ concentrations obtained when forcing the CMAQ model with the prior and posterior fluxes (for convenience, we will call these the prior and posterior $CO_2$ concentrations, respectively). Fig. 4 shows the bias and root-mean-square error (RMSE) between the prior and posterior CMAQ simulations against the OCO-2 observations for 2015. This figure shows that the biases and RMSE in the posterior concentration were reduced by the inversion and indicate the inversion system leads to an overall improvement of the representation of OCO-2 observations. Our findings indicate that the prior concentrations overestimate OCO-2 from March to April, and from July to September. Prior biases in these months were reduced by more than 90%. In March, for example, the monthly mean bias was reduced from 0.56 to 0.05 ppm, with a decrease in the RMSE from 1.11 to 0.84 ppm. In April, we see similar results to March, the prior bias was reduced from 0.40 (RMSE = 1.05) to 0.03 (RMSE = 0.88). On the other hand, in January, February, May and December prior biases were negligible, showing a good agreement with OCO-2. In a consistent system, we know that the theoretical value of the cost function at its minimum should be close to half the number of assimilated observations, assuming all error statistics are correctly specified (Tarantola, 1987, p. 211). In our inversion, after iteration 27, we obtained a cost function of 4392.15, which was close to half of the total number of OCO-2 assimilated observations for 2015 (N = 9556). In general, we also see a modest reduction in the prior RMSE each month during 2015, and its variability is proportional to the number of assimilated observations. Thus, a slight prior RMSE decrease corresponds to a month with a reduced number of OCO-2 data available.

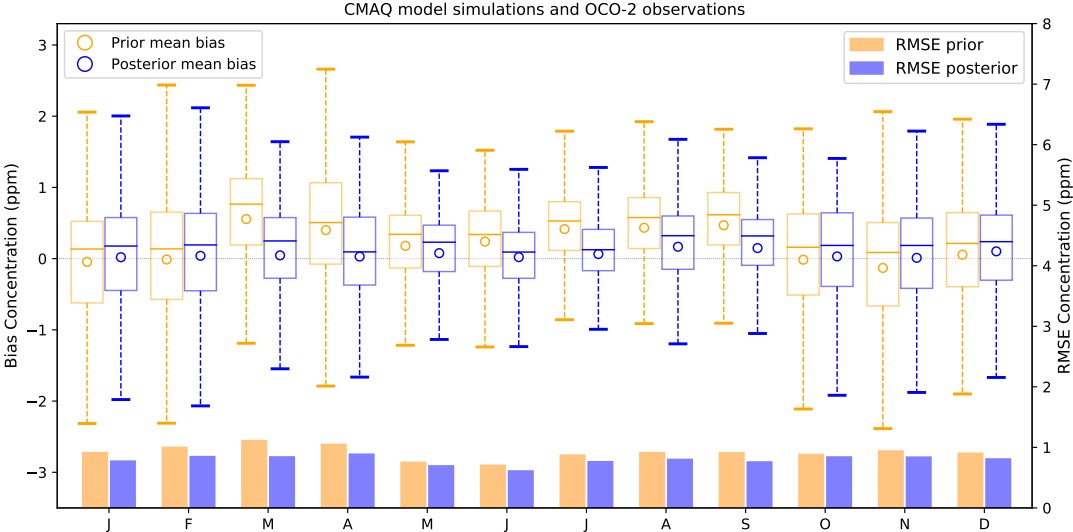

**Figure 4.** Bias and root mean square error (RMSE) between OCO-2 and the prior and posterior concentrations simulated by CMAQ model. Orange and purple circles represent prior and posterior concentration biases, and orange and blue bars represent the RMSE (Units: ppm). The top edge of the box represents the 75th percentile and the bottom edge represents the 25th percentile. The top and bottom whiskers represent the 95th and the 5th percentile.

## 3.2 Australian carbon flux estimate

In this section, we will only discuss the results of carbon fluxes that were assimilated by the combination of LNLG OCO-2 retrievals, not the carbon fluxes estimated using LNLGOG observations. We decided not to discuss the results based on LNL-GOG because ocean glint observations (version 9) still have undetermined biases (O'Dell et al., 2018) that might contaminate the Australian carbon flux estimate. However, we include these findings in the Appendix F, Table F1. Adding ocean OCO-2 glint observations to our inversion system does not significantly alter the terrestrial annual mean flux estimate for Australia

(-0.36 PgC y$^{-1}$) compared to estimates made by only using OCO-2 LNLG observations (-0.41 PgC y$^{-1}$). Results based on LNLGOG will be further discussed in Section 4.

Fig. 5a represents the terrestrial prior and posterior annual mean flux for 2015 (excluding fossil fuel). As mentioned previously, our assimilated carbon fluxes using LNLG indicate that the Australian annual terrestrial flux for 2015 was a slight carbon sink of -0.41 $\pm$ 0.08 PgC y$^{-1}$ (1-sigma uncertainty) compared to the prior terrestrial estimate of 0.09 $\pm$ 0.17 PgC y$^{-1}$. Our

prior fossil fuel estimates from ODIAC and EDGAR, which is about 0.06 $\pm$ 0.01 PgC y$^{-1}$ (mostly constant for each month in 2015) over Australia represent only 25% of the annual posterior flux. We decided to exclude these emissions from our analysis because variations in land uptake cause most of the variation in our posterior fluxes.

Fig. 5b shows the seasonal cycle of the prior and posterior fluxes along with its uncertainties. As mentioned in Section 2.3, the prior and posterior uncertainties included in Fig. 5a and Fig. 5b were calculated from an ensemble of five different OSSE

experiments adjusted by a factor of 1.2 in this study. We also plotted the spatial distribution of the prior and posterior fluxes (Figs 6 and 7), and the difference between them (Appendix A, Fig. A1).

Fig. 5b shows that the posterior flux estimates generally refine the prior with the exception of March and the period July to September. In January and February, the posterior fluxes were not modified much by the inversion. In January, for example, the terrestrial posterior flux was -0.84 $\pm$ 0.18 PgC y$^{-1}$ compared to the prior -0.89 $\pm$ 0.75 PgC y$^{-1}$. The agreement follows from
the small residual between prior simulated concentration and observation (Fig. 4). From April to May, we also see the posterior is shifted from the prior, although not significantly considering the prior uncertainty. In April, for instance, the prior flux (0.24 $\pm$ 0.69 PgC y$^{-1}$) was slightly shifted to a posterior carbon sink (-0.28 $\pm$ 0.25 PgC y$^{-1}$). However, these two estimates do not disagree because they fall within 1-sigma uncertainties.

As mentioned in the previous paragraph, March, July, August, and September were the exceptions to this general agreement.
In March, we see a prior flux of 0.12 $\pm$ 0.74 PgC y$^{-1}$ compared to the posterior carbon sink of -0.82 $\pm$ 0.17 PgC y$^{-1}$. The difference between the posterior (Fig 7b) and the prior flux (Fig 6b) at grid-cell scale (see Appendix A, Fig. A1, panel c) suggests that most of the posterior sink comes from the north and south-east corner of Australia. July represents the month where the posterior is most shifted from the prior. In this month we see a posterior flux of -1.75 $\pm$ 0.34 PgC y$^{-1}$ compared to the prior flux of 0.09 $\pm$ 0.62 PgC y$^{-1}$. The spatial distribution of the posterior and prior flux difference at grid-cell scale for
July (Fig. A1, panel g) indicates that the shift largely comes from northern and south-eastern Australia. The stronger posterior sink seen in July decreased in August (-0.93 $\pm$ 0.27 PgC y$^{-1}$) and September (-0.78 $\pm$ 0.20 PgC y$^{-1}$), and changed sign in October and November. In November, the posterior flux was 1.75 $\pm$ 0.31 PgC y$^{-1}$ compared to the prior, which was 0.53 $\pm$ 0.58 PgC y$^{-1}$. The largest difference in this month is found in the north and south east coast of Australia (Appendix A, Fig. A1, panel k). The carbon release from land in northern Australia is likely attributed to a combination of fire anomalies
(Supplementary Fig S3, panel k) and the lack of rainfall seen in Australia in 2015 (Supplementary Fig S2, panel k). In December we see that the posterior source seen in November changed to a posterior carbon neutral (0.003 $\pm$ 0.15 PgC y$^{-1}$). A further analysis which explains the reasons for this shift is given in the following section.

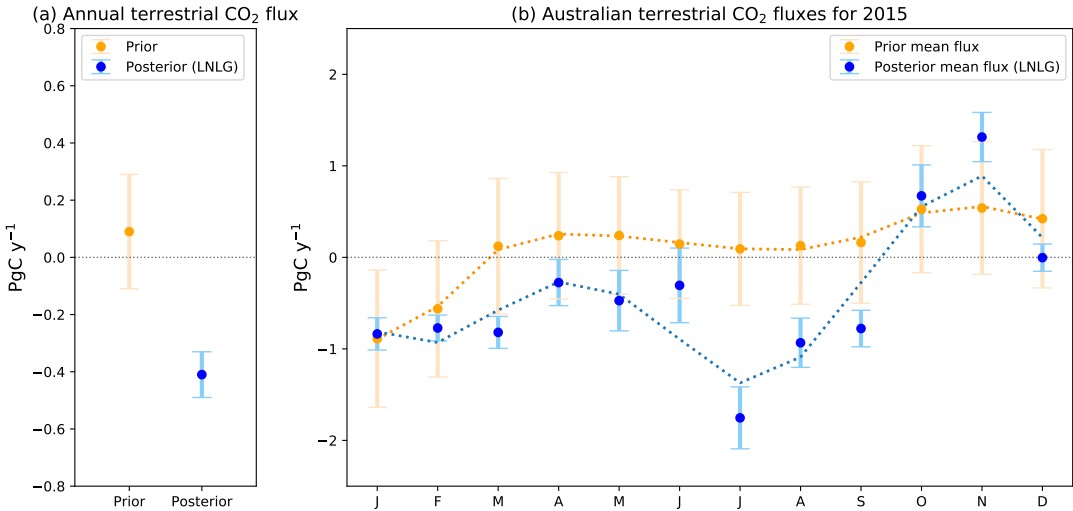

**Figure 5.** Time series of monthly mean prior (orange dots) and posterior (blue dots) $CO_2$ fluxes and their uncertainties in PgC y$^{-1}$ over Australia for 2015. The orange and blue dashed line represents a smooth line for the prior and posterior fluxes respectively.

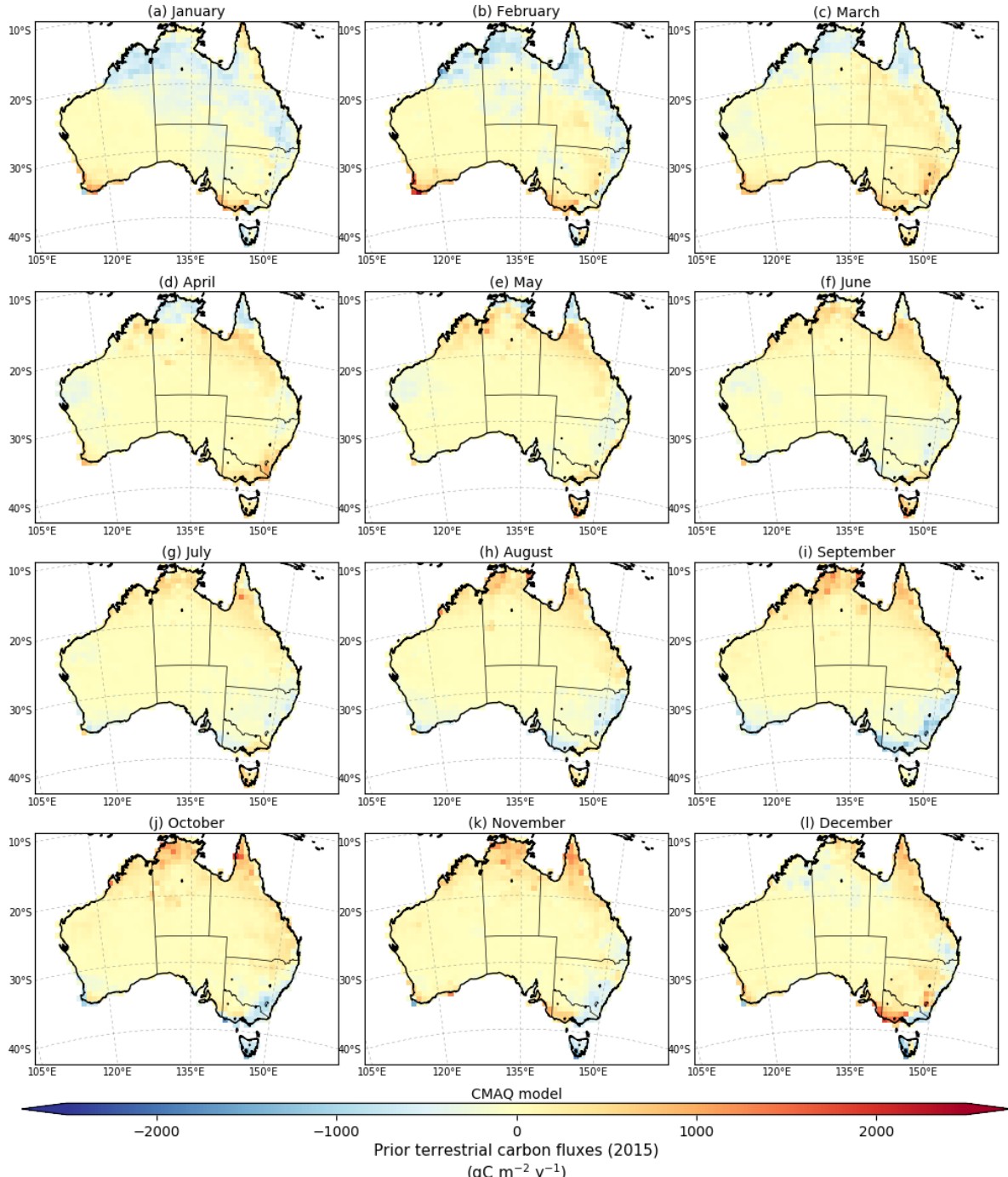

**Figure 6.** Prior fluxes derived by the CABLE model in the BIOS3 set-up in combination with fires emissions selected by GFED for 2015 (Fossil fuel emissions are excluded).

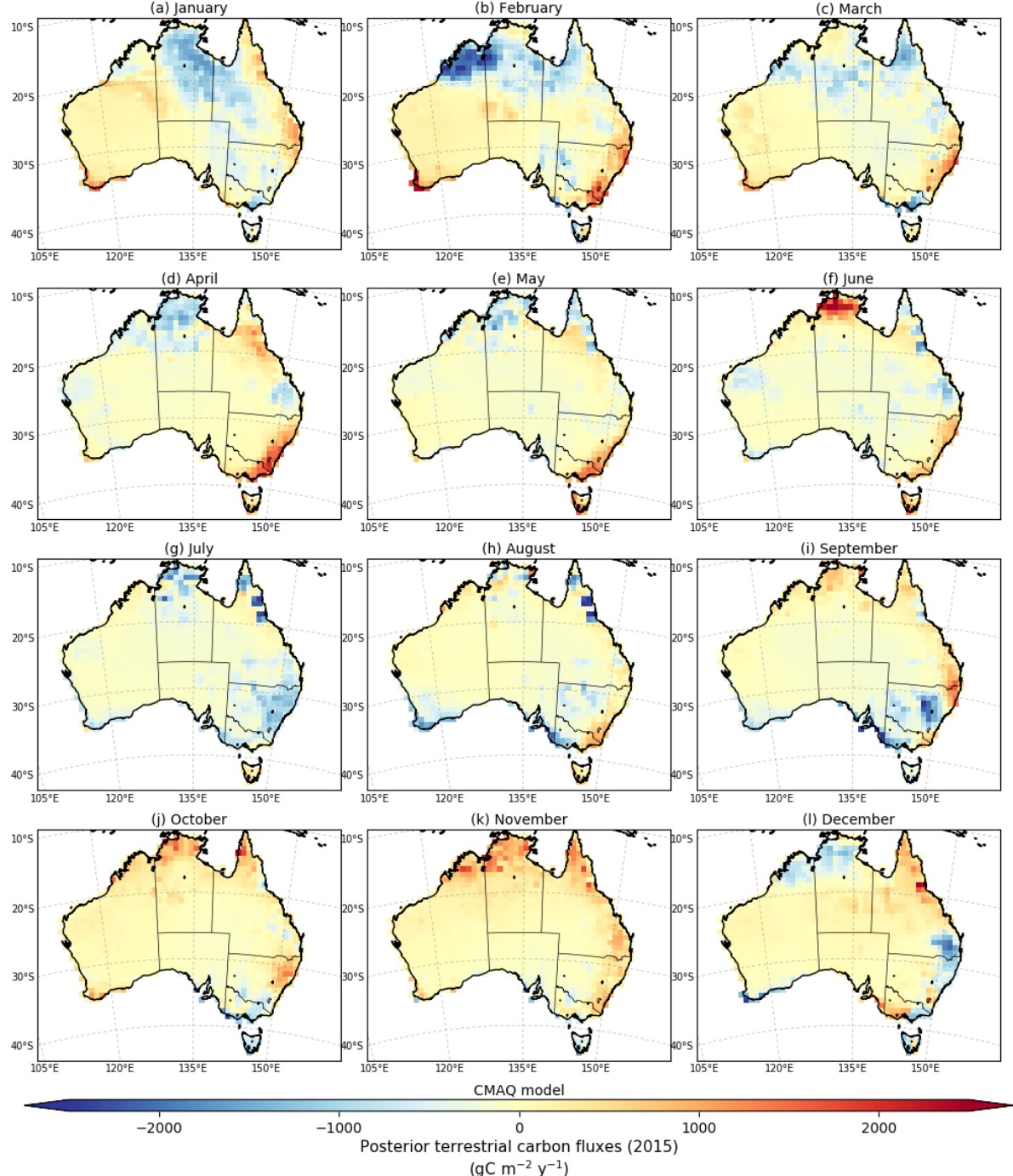

**Figure 7.** Posterior fluxes assimilated using LNLG OCO-2 satellite observations for 2015 (fossil fuel emissions are excluded).

### 3.3 Spatial patterns of the EVI and rainfall anomalies in Australia

To investigate why our inversion led to a higher carbon uptake (relative to the prior flux) in some months in 2015, we studied the spatial pattern of monthly EVI anomalies and rainfall anomalies. EVI anomalies were calculated relative to 2000−2014 over Australia from the MOD13C1 version 6 data product and rainfall anomalies from AWAP data relative to 2000−2014 (Supplementary Figs. S1-S2). We indicated in Section 3.2 that the posterior sink recorded in January and February agrees with the prior estimates but disagrees with the prior flux estimates from March to September, where the most considerable difference is seen in March and from July to September. From the inversion viewpoint, the significant shift between prior and posterior fluxes occurs because the prior column averaged simulated by the CMAQ model overestimates the column-average retrieval by OCO-2 in these periods (see Fig. 4).

As indicated in Section 3.2, most of the posterior carbon uptake seen in March comes from the northern part of Australia (except coastal regions) and the south-eastern area (Appendix A; Fig. A1, panel c). We found that the higher posterior uptake relative to the prior in the northern part of the continent was not attributed to an increase of greenness in vegetation (negative EVI anomalies). These results suggest that most of the posterior carbon uptake observed in March is likely associated with positive EVI anomalies seen in January and February affected by the positive rainfall anomalies recorded in January. High anomalous rainfall in January is not unexpected because it is the wet season in the northern region of Australia (tropical monsoonal climate).

The spatial pattern of the difference between the posterior and prior flux estimate recorded in July indicates that the majority of the posterior carbon uptake estimated by the inversion comes from the south-eastern and northern region of Australia (Appendix A; Fig. A1, panel g). We found that the posterior sink estimated in south-eastern Australia was likely driven by a higher than expected greenness of vegetation (Supplementary Fig. S1, panel g), probably induced by anomalously positive rainfall in that period (Supplementary Fig. S2, panel g). We cannot conclude the same results in the northern region that we found in the south. We see in Appendix A; Fig. A1, panel g that positive EVI anomalies were slight compared to the one found in south-eastern Australia. In the following section, we will show that the underestimation of the GPP by the CABLE-BIOS3 model might be the likely reason for the difference between prior and posterior in this region.

An increase in carbon uptake estimated by our inversion in August comes from the northern and southern region of Australia (with the exception of coastal areas in the southern-east corner of Australia), which mainly shows a release of carbon (Appendix A; Fig. A1, panel h). The release of carbon by the land in this coastal region is likely attributed to a decrease in land productivity (Supplementary information, Fig. S1, panel h). The subtle decrease of photosynthesis activity in the coastal area is likely associated to a decrease of rainfall seen in June and July (Supplementary Fig S2, panels f and g).

In September, the posterior carbon uptake primarily comes from the south-east corner of Australia (with a slight exception seen in the south-east and east coast of Australia), which shows a release of carbon into the atmosphere (Appendix A; Fig. A1, panel i). The carbon uptake seen in the south-east of Australia aligns with a higher than usual increase in land productivity, as reflected by the positive EVI anomalies in that region (Supplementary Fig S1, panel h), likely benefited by the positive rainfall anomalies seen in August in that area. In September, we also see that positive EVI anomalies were not as strong as in July and

August. These findings are probably associated with the fact that rainfall in September decreased considerably for most parts of the country, where rainfall was lower than average (negative rainfall anomalies) (Supplementary Fig S2, panel i).

Anomalies in EVI in Australia are closely related to fluctuations in rainfall, which is one of the most important drivers of ecosystem dynamics and productivity. This is the case in (e.g., semi-arid) regions where rainfall is the limiting factor for plant growth, which is indeed the case in much of Australia. These results are consistent with findings of previous studies (e.g., Weltzin et al., 2003) that Australia's semi-arid ecosystems are water resilient and can respond to favourable rainfall conditions by capturing large amounts of carbon.

## 3.4 Australian carbon flux estimate classified by bioclimatic zones

To understand which Australian ecosystem contributed most to our posterior carbon sink estimate, we divided the continent into six bioclimatic classes: tropical, savanna, warm temperate, cool temperate, Mediterranean, and sparsely vegetated (Fig. 8). We used the same six bioclimatic regions at a 0.05 degree spatial resolution as in Haverd et al. (2013a). The six bioclimatic classes used in this study correspond to an aggregation of the 18 agro-climatic zones generated by Hutchinson et al. (2005). The climatic classification in Hutchinson et al. (2005) was adapted from an existing global agro-climate classification (Hutchinson, 1992), which was refined and closely aligned with natural vegetation formations and common land uses across Australia using 182 weather climate stations and the Interim Biogeographic Regionalisation for Australia (IBRA). In Fig. 8, we can see that Australian tropical land only covers the northern coastal part of Australia. Savanna extends across the northern tropics to the south-eastern subtropical zone. Warm temperate land covers the south-east Australian coast, while cool temperate land covers the south-eastern corner of Australia. The Mediterranean region is confined to the south-western corner of Australia and the gulf region of South Australia. The sparsely vegetated ecosystem represents the biggest ecosystem over Australia, which extends from the northern subtropical zone to southern Australia.

Fig. 8 also includes the prior and posterior annual flux aggregated into these bioclimatic regions. It is evident that savanna and sparsely vegetated ecosystems were the regions across Australia that most contribute to posterior carbon sink estimated for 2015. The annual posterior carbon flux for savanna was -0.17 $\pm$ 0.03 PgC y$^{-1}$ compared to the prior annual flux (0.09 $\pm$ 0.11 PgC y$^{-1}$), and the annual posterior carbon sink over sparsely vegetated was even higher (-0.25 $\pm$ 0.07 PgC y$^{-1}$) compared to prior annual flux (-0.01 $\pm$ 0.07 PgC y$^{-1}$). These results are not unexpected because the sparsely vegetated ecosystem represents the largest bioclimatic region in Australia, and a slight shift of carbon fluxes across this area causes a significant impact on the total annual flux for this ecoregion, and for total annual flux estimated for Australia.

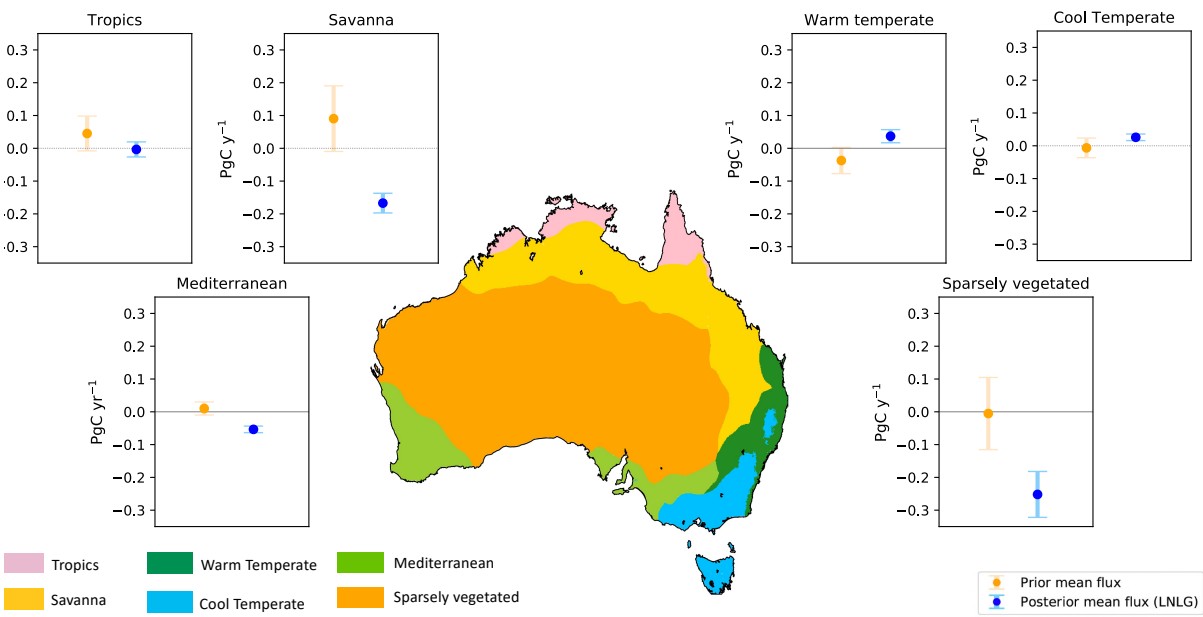

**Figure 8.** Annual prior and posterior flux estimates aggregated into six bioclimatic classes (tropics, savanna, warm temperate, cool temperate, Mediterranean, and sparsely vegetated) over the Australian region. Fossil fuel emissions are excluded.

Fig. 9 shows the monthly time series of the prior and posterior terrestrial flux aggregated into these bioclimatic regions. Over the savanna ecosystem (Fig 9b), our inversion indicates that from January to June, this ecosystem acted as carbon sink. In February, in this ecosystem, we see that the prior sink (-0.48 $\pm$ 0.40 PgC y$^{-1}$) strengthens to a posterior of -1.07 $\pm$ 0.10 PgC y$^{-1}$. The stronger carbon sink (relative to the prior) from January to March coincides with an increase of greenness in vegetation (positive EVI anomalies) in this ecosystem (see Supplementary; Fig. S1, panel a and b), benefited by anomalous rainy conditions in January (see Supplementary; Fig. S2, panel a). Thus, it seems that the anomalous increase of rainfall in northern Australia in January benefits the increase in vegetation growth and carbon uptake recorded in February. However, it is difficult to draw conclusions about the posterior carbon uptake seen in months subsequent to March because of unfavourable raining conditions and negative EVI anomalies in these periods.

Another noticeable difference between prior and posterior flux estimates over savanna is seen in July and August. In July, we cannot conclude if the prior was a sink or source of carbon (0.19 $\pm$ 0.28 PgC y$^{-1}$). However, our inversion indicates that savanna was acting as a carbon sink of -0.35 $\pm$ 0.11 PgC y$^{-1}$. In August, the prior source (0.25 $\pm$ 0.20 PgC y$^{-1}$) becomes a posterior carbon sink of -0.22 $\pm$ 0.08 PgC y$^{-1}$. To understand the difference between the prior and posterior estimate in this period, we calculated the GPP estimated by the CABLE BIOS3 model and the GPP generated by MODIS (see Appendix E; Fig. E1, panel b). The temporal correlation between CABLE-BIOS3 and MODIS GPP was moderate (R = 0.69). According to MODIS estimates, the CABLE-BIOS3 GPP is overestimated from January to March, and underestimated from May to October. The underestimation of the GPP flux by the CABLE-BIOS3 model might explain why we find a stronger posterior sink estimated by our inversion in this category.

Over the warm temperate region, from February to April our posterior estimate suggests a carbon source (Fig 9c). For this period, we cannot determine if the prior flux estimate was a carbon sink or source due to its uncertainty range. In February, the prior flux (-0.05 $\pm$ 0.08 PgC y$^{-1}$) becomes a posterior carbon source of 0.17 $\pm$ 0.06 PgC y$^{-1}$. In March, the prior estimate was nearly neutral (0.04 $\pm$ 0.05 PgC y$^{-1}$) compared to the posterior carbon source estimate (0.17 $\pm$ 0.05 PgC y$^{-1}$). The reduced carbon uptake estimated by the inversion in this period does not agree with the positive EVI anomalies seen in this region, however it is likely that this extra carbon release to the atmosphere is related to an increase of leaf respiration in response to high temperatures recorded in 2015 for the majority of Australia (Annual climate statement, Bureau of Meteorology, 2015). Another possible reason for the relatively small shift from the prior in this period was most likely because the CABLE-BIOS3 GPP over-estimates MODIS GPP (Appendix E1, Fig. E1, panel c). For the warm temperate category, the correlation of CABLE-BIOS3 and MODIS GPP flux is high (R = 0.86).

We also see a subtle disagreement between prior and posterior estimates over the cool temperate ecosystem in April and May (Fig 9d). In this period, our posterior estimate indicates that this category was a stronger carbon source than the prior flux estimate. In April, the inversion strengthened the prior source (0.12 $\pm$ 0.1 PgC y$^{-1}$) to a posterior of 0.47 $\pm$ 0.05 PgC y$^{-1}$. In May, we cannot define if the prior was a sink or a source (0.06 $\pm$ 0.09 PgC y$^{-1}$), however, our assimilated fluxes indicate this category was acting as a posterior carbon source (0.36 $\pm$ 0.04 PgC y$^{-1}$). The most likely reason for a larger carbon release in this period is related to negative EVI anomalies seen across this ecosystem. While it is true that April and May see positive EVI anomalies (Supplementary, Fig. S1, panels d and e), in April, we notice predominantly negative EVI anomalies in the southern corner of the Australian (mainland) and Tasmania. The analysis of GPP between the CABLE-BIOS3 model and MODIS also shows some discrepancies (see Appendix E; Fig. E1, panel d). For this category, in general, the CABLE-BIOS3 GPP is overestimating the productivity of the land for the whole year. For example, the absolute difference between both GPP data sets in April and June is about 0.2 PgC y$^{-1}$. For the cool temperate category, the correlation between CABLE-BIOS3 and MODIS GPP is moderate (R = 0.73).

Another disagreement between the prior and posterior terrestrial flux estimate is seen over the Mediterranean ecoregion in August (Fig 9e). Our posterior estimate is a flux of -0.35 $\pm$ 0.08 PgC y$^{-1}$ compared to the prior of -0.12 $\pm$ 0.12 PgC y$^{-1}$. An increase in vegetation productivity may also be the reason for the increase in carbon uptake by this category (positive EVI anomalies, supplementary, Fig S1, panel h). This larger carbon uptake was likely a consequence of an increase of rainfall in this category (greater than 60% on average (relative the mean 2000–2014) for some areas of this ecosystem, supplementary, Fig S3, panel h). We also found that CABLE-BIOS3 underestimates MODIS GPP by 0.2 PgC y$^{-1}$.

We found a noteworthy discrepancy between the prior and posterior flux estimate over sparsely vegetated ecosystem from March to September (Fig 9f). In this period, in general, the absolute difference between the prior and posterior mean was around 0.4 PgC y$^{-1}$. The largest difference found in July and September was about 0.6 PgC y$^{-1}$. This highly unexpected and counter-intuitive difference is not because we see a significant increase of positive EVI anomalies across this ecoregion (Supplementary, Fig S1). On the contrary, it is because a "small shift" in the carbon fluxes over this large ecosystem causes an important impact on the total carbon net flux calculated for the whole country. We clearly demonstrate this fact in Appendix B, Fig. B1. This Figure in the appendix shows the fluxes divided by area. In the western region of this category, we see evident

positive EVI anomalies which start from April and last all the way through to September, which again line up with positive rainfall anomalies in that period.

Analysis of the GPP also shows the stronger posterior sink estimated by our inversion might be associated with a underestimation of the GPP by CABLE-BIOS3 in this category. The absolute difference between the CABLE-BIOS3 GPP and MODIS GPP was almost the same between May and September, with range $0.8-1.1$ PgC y$^{-1}$. This underestimation in GPP also suggests an underestimation of the land productivity. In this same category the posterior sink estimated in September disappears in October. For this period, our posterior source estimate ($0.14 \pm 0.08$ PgC y$^{-1}$) did not change much from the prior ($0.18 \pm 0.13$ PgC y$^{-1}$). In November and December, our posterior source was strengthened by the inversion. In November, we estimated a carbon source of $0.21 \pm 0.08$ PgC y$^{-1}$ in comparison with the prior, which was $0.12 \pm 0.13$ PgC y$^{-1}$. The extra carbon release estimated by the inversion in November might likely be associated with the combination of fires (Supplementary, Fig.S4, panel k) located in the west and central north west region of Australia (Supplementary, Fig.S4, panel k) and due to high temperatures recorded across Australia in summer (Annual climate statement, Bureau of Meteorology, 2015). These conditions certainly intensified the wildfires seen in that period.

In summary, our results showed that OCO-2 produced a shift in the carbon flux (relative to the prior) over the savanna and sparsely vegetated region. We found strong negative correlations (R> 0.8) at grid-cell scale between the EVI anomalies and the posterior and prior difference in northern Australia (savanna ecosystem) and in the western region of the sparsely vegetated ecosystem, which align with the spatial pattern of rainfall in that area. These results suggest that our OCO-2 inversion might likely be better capturing the anomalies in comparison with the biosphere land model.

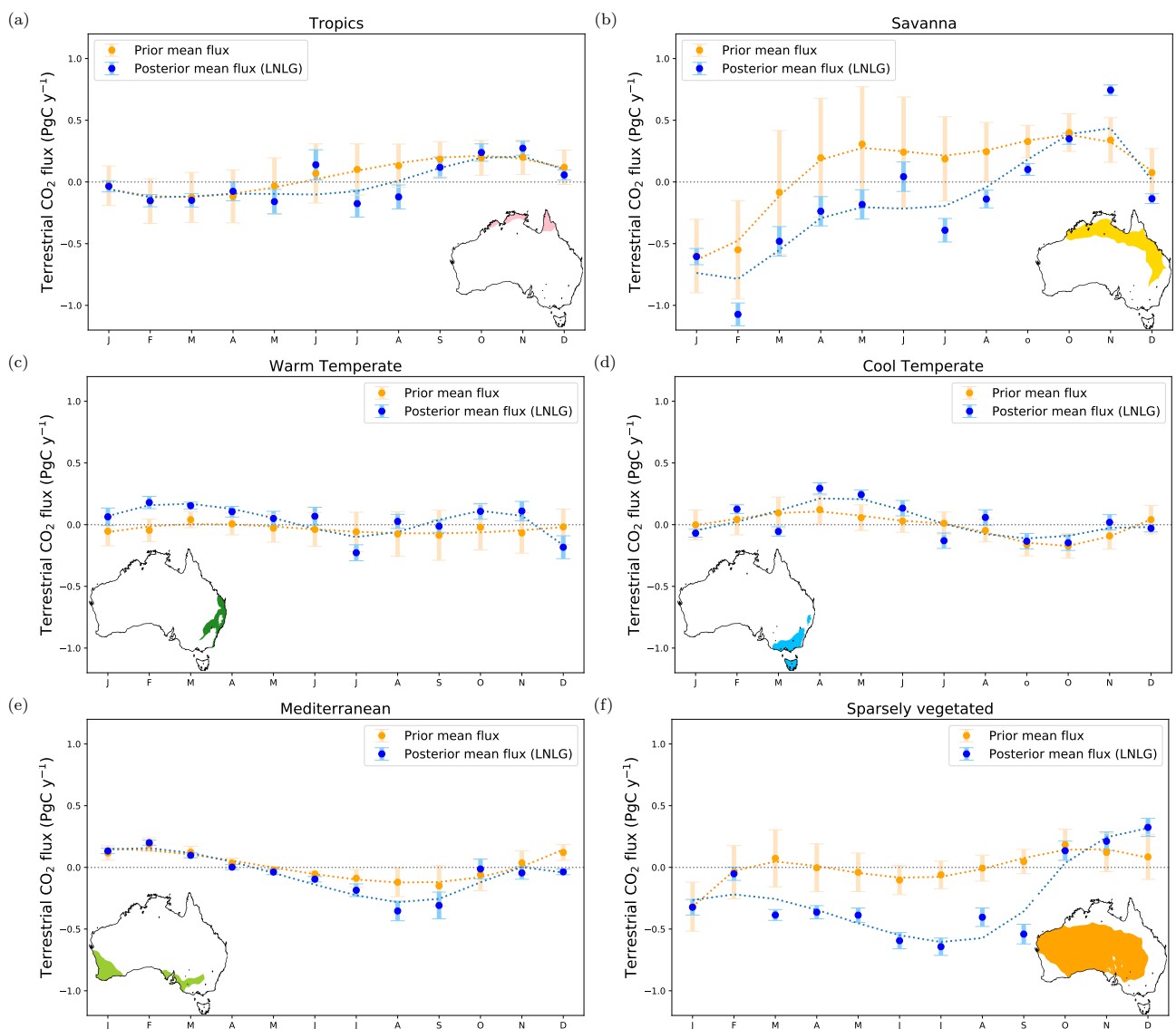

**Figure 9.** Monthly time series of the Australian land biosphere prior and posterior $CO_2$ flux and their uncertainties in PgC $y^{-1}$ aggregated over six bioclimatic regions. The prior and posterior estimates do not include fossil fuel emissions.

## 3.5 Evaluation of the inversion with independent data

In this section, we evaluate the accuracy of our posterior fluxes by comparing the residual between the prior and posterior concentrations simulated by CMAQ against TCCON and *in-situ* observations. In this comparison, we simulate the posterior concentration with fluxes that were not only assimilated by nadir and glint satellite observations (LNLG) but also by the

combination of both land nadir and land and ocean glint observations (LNLGOG). We decided to examine whether biases in
our posterior concentration could improve when incorporating glint ocean observations into the inversion.

### 3.5.1 Comparison with TCCON observations

As mentioned in Section 2.5.1, we selected TCCON observations from three different sites (Darwin, Wollongong, and Lauder; see Fig. 3). The comparison between the monthly mean column-averaged from TCCON sites and the prior and posterior column averaged concentration simulated by CMAQ, including both bias and root mean square error (RMSE) are shown in Figs. 10 and 11 respectively.

In Fig. 10a, we see that in late spring, summer, and early autumn in Australia (November to March) the posterior column-average simulated by CMAQ model (LNLG) is in better agreement with TCCON Darwin estimates than the prior. In this period, prior mean biases were reduced by approximately 30-80%. For example, in November and December, the prior concentration biases were reduced from -0.25 (RMSE = 0.51) to 0.11 (RMSE = 0.42), and from -0.34 (RMSE = 0.48) to -0.02 (RMSE = 0.31) respectively. Large seasonal differences (approximately 1 ppm) are seen between the TCCON observation, the prior and posterior column-average concentrations (LNLG) from June to September. Despite the fact that we see an improvement of the prior biases in this period, assimilating OCO-2 data does not significantly reduce them. The remaining posterior concentration biases of about +1 ppm might be explained by spurious OCO-2 soundings affected by biomass burning aerosols seen in that period. In northern Australia, winter occurs in dry season, and it is highly impacted by wildfires (see Supplementary; Fig. S4). OCO-2 spectrometers measure reflected sunlight from Earth's surface, and regions heavily affected by fires can lead to a modification of the light path length because the instrument struggles to distinguish between photons reflected by intermediate scatterers and photons reflected from Earth's surface (O'Dell et al., 2018). In terms of the posterior bias improvement when fluxes have been assimilated by OCO-2 LNLGOG data, we can see that the improvements are negligible, and in some periods such as January or May, the posterior biases get worse. This result suggests that the uncharacterized OCO-2 glint ocean bias degrades the performance of the inversion. We also found that our posterior column-average concentrations were better correlated with TCCON in comparison with the prior concentration (Appendix G, Table G1).

At Wollongong site, in general, we see a consistent overestimation of the prior and the posterior column average (LNLG) simulated by CMAQ (Fig 10b). We observe a relatively slight reduction of the prior biases in February, March, and November (spring, summer, and early autumn in Australia, Fig 11b). In November, for example, prior negative biases of about -0.74 ppm (RMSE = 1.22) were reduced to -0.40 ppm (RMSE = 1.13). The small reduction of the biases in this period is likely associated with strong winds coming from the ocean to the TCCON station (Supplementary, Fig. S7). Wollongong TCCON site is strongly affected by ocean fluxes, which are less restricted by our inversion when we only use LNLG observations. In late autumn and winter at Wollongong, we see high significant positive posterior biases (range between 1.1 and 1.61 ppm). Biases in winter season are likely related to OCO-2 than TCCON biases. It has been found that passive satellite instruments have difficulties measuring at high and middle latitudes in winter because the sun stays low in the sky (Wunch et al., 2017). A low solar altitude angle corresponds to a high solar zenith angle and high air mass, which means it takes longer for the sunlight to reach the satellite instrument. Biases related to high airmasses ("long path length") can be obtained because the absorption

spectra tend to saturate at the line centre, causing the column line shape of the absorption line to be more sensitive (Jacobs et al., 2020). TCCON retrievals contain an airmass-dependent bias, which is corrected using the method described by Wunch et al. (2011) and Deutscher et al. (2010). To evaluate whether any residual airmass-dependent bias is present in the TCCON retrievals, which would cause the seasonal posterior biases seen here, we filtered the TCCON dataset to contain only selected solar zenith angles $\geq 40$ and $\leq 50$ (see Appendix G, Table G3). We found only a slight improvement of the posterior biases, which means TCCON retrieval bias is not likely the reason for the biases seen in winter. Similar to the Darwin site, we did not find an improvement of the prior biases by adding ocean glint data to the inversion. Besides, ocean glint data (as it is shown in Appendix C, Fig. C2, panel d-h) is quite sparse around the Wollongong site providing little constraint on carbon fluxes around this location.

Similar to the Darwin and Wollongong sites, we also found a systematic overestimation of our posterior column-average (LNLG) concentration at TCCON Lauder site (Fig. 10c). Prior biases are less than 0.8 ppm. A slight improvement was seen in June, July, September, and November (Fig. 11c). In June and July (winter season) the reduction of the biases was only about 7%. Improvement of the biases in November and September were better (10 and 18%), and we did not find improvement in the correlation for these months (see Appendix G, Table G4). The small or negligible improvement of the prior biases at this site is likely due to a combination of New Zealand's size and shape, the prevailing wind direction, and the fact that we do not allow much freedom for ocean fluxes by specifying a small prior uncertainty. Adding ocean glint observation to the inversion did not improve the accuracy of the biases at this site. The spareness of OCO-2 soundings over the ocean around New Zealand in the period from May to September might explain the lack of improvement in this bias. Higher resolution models and smaller correlation lengths (allowing more flexibility in spatial fluxes) would be required for good performance over New Zealand.

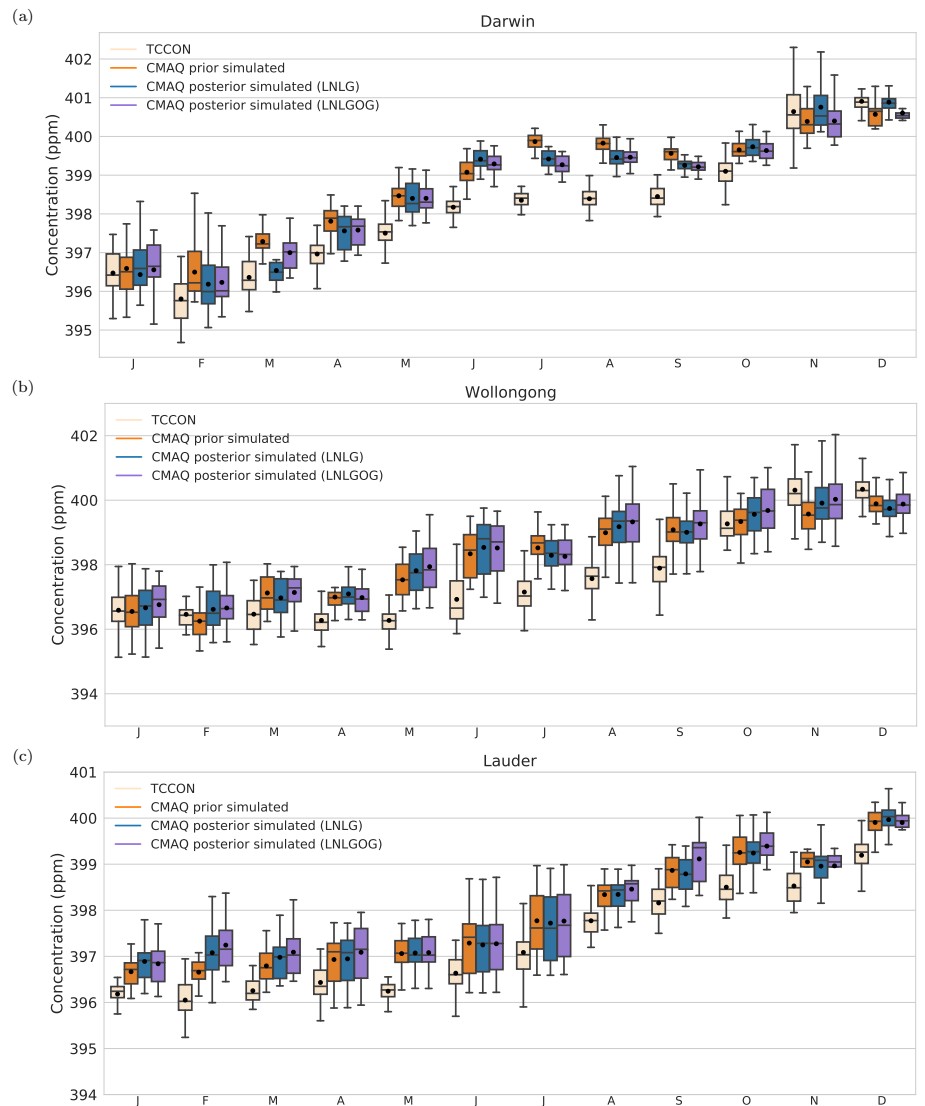

**Figure 10.** Box plot diagrams show the monthly mean average of $CO_2$ concentration at Darwin (a), Lauder (b) and Wollongong (c) TCCON site for 2015. The top edge of the box represents the 75th percentile and the bottom edge represents the 25th percentile. The top and bottom whiskers represent the 95th and the 5th percentile. The horizontal black line shows the median and the circle indicates the mean. Mean values are indicated by blue circles and median values by black line.

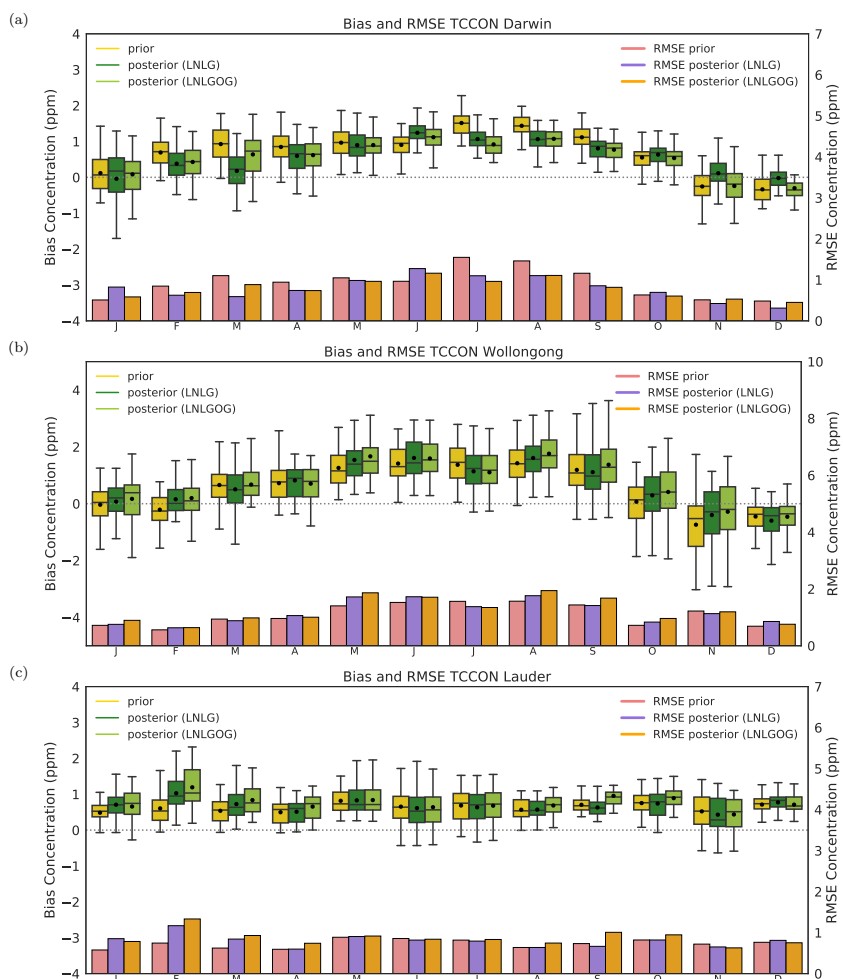

**Figure 11.** CMAQ prior and posterior concentration bias and root mean square error at TCCON sites (a) Darwin, (b) Wollongong and (c) Lauder for 2015. Yellow-green and green boxes represent prior posterior concentration biases and coral and purple bars represent the RMSE. Box plots represent the 75th percentile and the bottom edge represents the 25th percentile. The top and bottom whiskers represent the 95th and the 5th percentile. The horizontal black line shows the median and the circle indicates the mean. Mean values are indicated by black circles and median values by black line.

### 3.5.2 Comparison with *in-situ* measurements

Figs. 12 and 13 show the comparison between ground-based *in situ* measurements (Gunn Point, Burncluith, Ironbark and Cape Grim) and our prior and posterior concentrations simulated by CMAQ at the surface.

As illustrated in Fig. 12a, the inversion using only LNLG OCO-2 observations does not match Gunn Point observed concentrations well except in September. Most biases are negative, indicating that the posterior simulation at the surface of the CMAQ model underestimates the observations. The prior concentration indicates a better agreement, but biases are still sig-

nificant. One possible explanation for the large negative posterior biases in January, February, March, and December might be related to strong westerly winds that blow from the ocean to this site location (see Supplementary; Fig. S9). Using only LNLG observations restricts our inversion to optimized ocean fluxes because the uncertainties set-up it over ocean were lower compared to land. However, when we added ocean glint observations the posterior concentration biases get better. These results are not unexpected because Gunn Point is a coastal site largely affected by ocean carbon fluxes. In February, and when the wind comes from the ocean, the posterior bias using LNLGOG shows a significant improvement compared to prior bias concentration. Here, we see a reduction of the bias from 1.93 (RMSE = 4.21) to 0.65 (RMSE = 3.79).

In winter (June to August), we see that the posterior biases using either OCO-2 LNLG or LNLGOG show no improvement, and the prior biases are in better agreement with the observations. One possible explanation might be related to the fact the column-integrated $CO_2$ measurements are less sensitive to near-surface dynamics compared to *in-situ* measurements (Lauvaux and Davis, 2014), or to remaining bias in the OCO-2 data. Despite the fact that version 9 has an improvement in the bias correction, in the recent study performed by OCO-2 MIP (Peiro et al., 2021) shows that LNLG data still has large negative latitudinal biases in the Southern Hemisphere. Another potential explanation could be associated with an inaccurate representation of vertical transport within the planetary boundary layer in winter by the CMAQ model. Incorrect vertical transport might lead to erroneous horizontal distributions of air masses (Lauvaux and Davis, 2014). Therefore, correcting the prior column-average simulated by CMAQ to match OCO-2 might not improve near surface simulations.

Improvement of the bias using LNLG observations at Ironbark are only seen in January, February, May, September and November (Fig. 12b). We found high negative posterior biases in June and July. The negative posterior bias in June, -2.79 ppm (RMSE = 3.53), might be associated with the small number of OCO-2 soundings located around Ironbark (see Appendix C, panel f) and the wind direction in that region. We can see in Fig. S7 (Supplementary) that prevailing winds blow from the south-east, an area with no OCO-2 soundings to constrain fluxes. In July, posterior biases are larger than prior (-0.35 ppm to -2.33 ppm). Again, biases in winter might be associated with error in the transport model, or remaining biases in LNLG OCO-2 observations. At this site, we do not see an improvement of the posterior bias when we added glint ocean data to the inversion.

Results for the Burncluith station (Fig. 3c) are similar to Ironbark. This is not surprising given the stations' proximity. The posterior simulation performs better in July, October, and December at Burncluith than Ironbark.

The posterior LNLG simulation at Cape Grim, shown in Fig. 12d, is in better agreement with the observations than the prior concentrations for the austral Autumn and early Winter of 2015. By contrast, high posterior negative biases are seen from September to December ($> 2$ ppm). This seasonality of bias is likely related to the seasonality of wind direction. The predominantly northerly flow in winter brings air from mainland Australia where fluxes have been constrained by OCO-2 observations. The southerly flow later in the year brings air from the Southern Ocean, unconstrained by observations. Similar to the Gunn Point site, we found that adding ocean glint observations to the inversion improved the prior mean concentration bias considerably. In January, for example, we see a reduction of the bias from -2.31 (RMSE = 3.380) to -0.54 (RMSE = 2.46) ppm using LNLGOG in the inversion. Again, these findings are not unexpected because Cape Grim is an oceanic station strongly influenced by oceanic carbon fluxes.

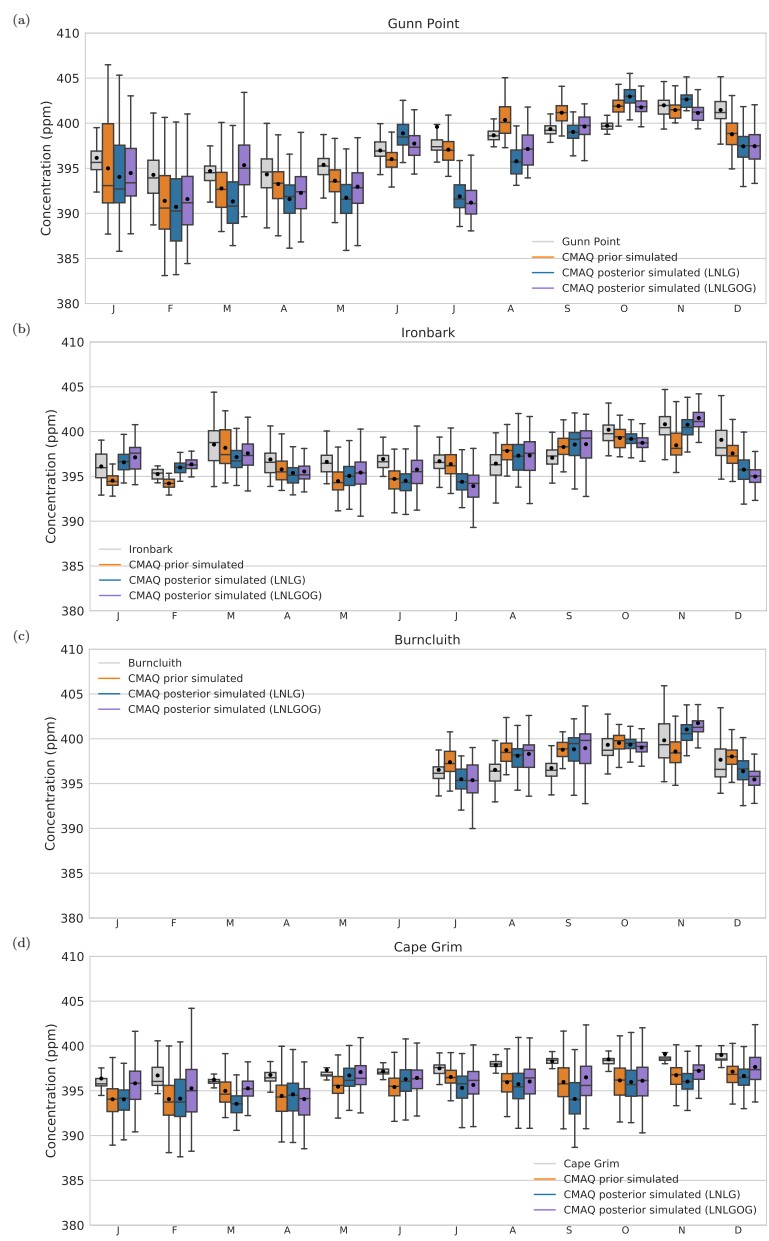

**Figure 12.** Box plot diagrams show the monthly mean average of $CO_2$ concentration at (a) Gunn point, (b) Ironbark, (c) Burncluith, and (d) Cape grim for 2015. For details of what the different components of the box-plot represent, see the caption of Fig. 10.

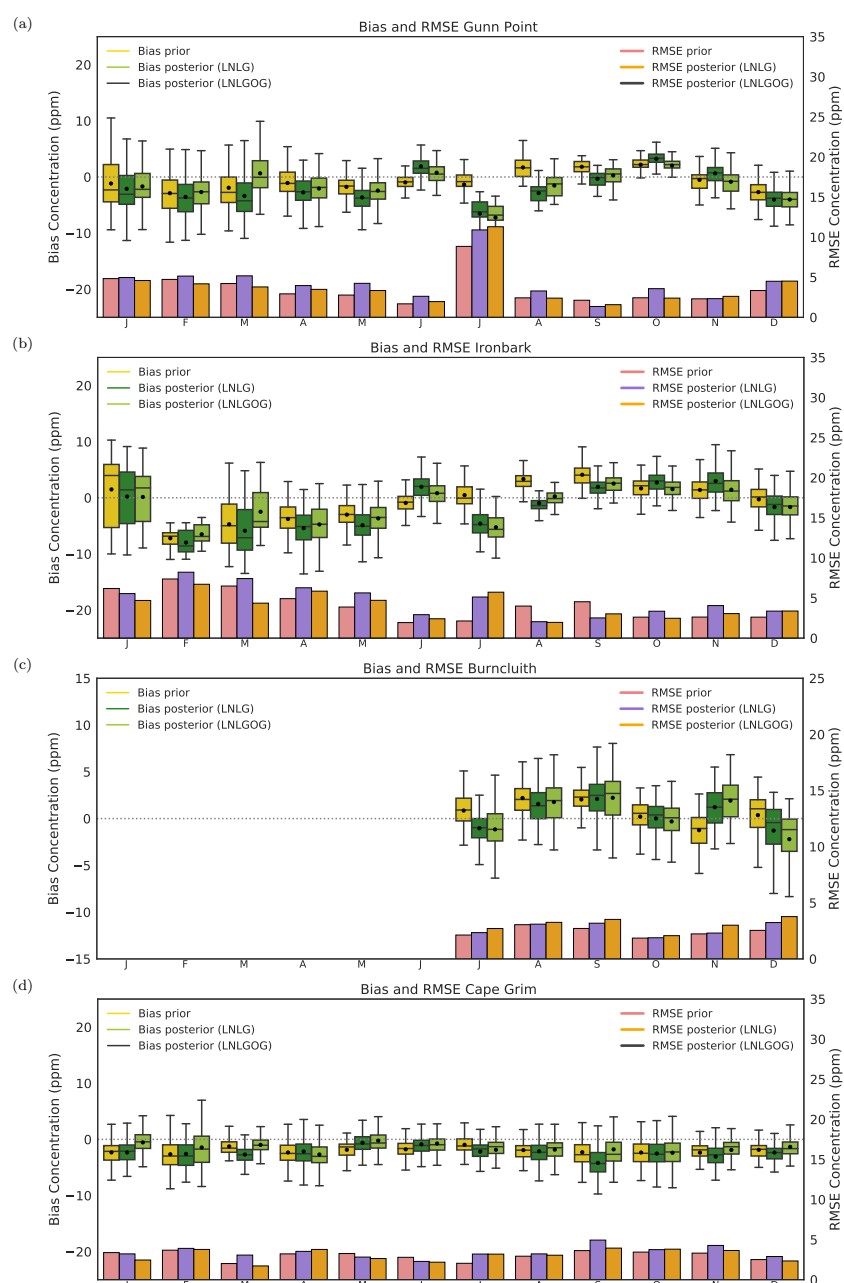

**Figure 13.** CMAQ prior and posterior concentration bias and root mean square error at (a) Gunn Point, (b) Ironbark, (c) Burncluith, and (d) Cape Grim for 2015. For details of what the different components of the box-plot represent, see the caption of Fig. 11. Note: Bias and RMSE in Gunn Point in July exclude the highest concentration value for that period, which was 558.408 ppm, we did this to better represent the Figure. Prior and posterior concentration bias for this period were -4.9 (RMSE = 16.8) and -5.1 ppm (RMSE = 17.1), respectively.

## 4 Discussion

We saw that semi-arid ecosystems in Australia, such as savanna and areas with sparse vegetation, are responsible to some extent for the stronger carbon sink (relative to the prior flux) recorded in 2015. We associated this carbon uptake with an increase in vegetation productivity (positive EVI anomalies) and an underestimation of the GPP flux by the CABLE land surface model. We speculate that the LAI estimated by the land-surface model CABLE-BIOS3 fails to capture the abrupt response of the terrestrial biosphere to rainfall over areas with sparse vegetation. This hypothesis could be tested by comparing CABLE-BIOS3 LAI with satellite-LAI. However, this is beyond the scope of this study and will be taken up in a forthcoming article.

We compared our findings against the ensemble mean of nine *in situ* and OCO-2 (LNLG) MIP global inversions (AMES, PCTM, CAMS, CMS-Flux, CSU, CT, OU, TM5$-$4DVAR, UT) for 2015 (Fig. 14). We can see in Fig. 14a that the OCO-2 MIP and our posterior inversion suggest that Australia was a carbon sink for 2015. Our flux annual mean carbon flux estimate for Australia (-0.41 $\pm$ 0.08 PgC y$^{-1}$) falls within the annual ensemble mean estimate of the MIP OCO-2 (LNLG) flux inversion (-0.23 $\pm$ 0.12 PgC y$^{-1}$). Contrary to these findings, the ensemble mean of the in situ MIP inversions suggest that Australia was a carbon source of 0.20 $\pm$ 0.22 PgC y$^{-1}$ for 2015. However, this fact cannot be concluded with high confidence because of the spread ensemble mean of in situ MIP global inversions.

In terms of seasonality, we can see in Fig. 14b that our inversion produces a similar seasonal pattern to the ensemble monthly mean of MIP-OCO2 (LNLG) (except for July), and produces an almost identical flux estimate for several months in 2015 (Fig. 14b). For example, in February and March, the monthly ensemble MIP OCO-2 (LNLG) was -0.77 $\pm$ 0.48 PgC y$^{-1}$ and -0.96 $\pm$ 0.21 PgC y$^{-1}$ compared to our posterior flux estimate, which was -0.77 $\pm$ 0.14 PgC y$^{-1}$ and -0.82 $\pm$ 0.17 PgC y$^{-1}$ respectively. These findings give confidence that the posterior carbon fluxes estimated in this study are reliable.

As we saw in Section 3.5 the validation of our inversion against Ironbark and Burncluith sites suggest that the anomalous sink seen in July might likely be related to errors in the transport model. If we interpolate July between June and August, we reproduce a monthly mean (-0.62 $\pm$ 0.34 PgC y$^{-1}$ ), which is closer to the ensemble mean of the OCO-2 MIP (-0.47 $\pm$ 0.58 PgC PgC y$^{-1}$), considering the range of the spread the model and the uncertainties of our posterior fluxes (see Fig. 15b). By doing this interpolation, we shift our posterior annual flux from -0.41 to -0.32 PgC y$^{-1}$, which is also closer to the annual ensemble mean of MIP-OCO-2 (-0.23 PgC y$^{-1}$).

The individual analysis of seasonal variations for the nine global carbon flux estimates, either derived by *in situ* or OCO-2 (LNLG) observations, show a large disagreement between them (Appendix I; Fig. I1 and Fig. I2). However, the variation of the seasonal cycle between *in situ* global flux inversions is more evident. We can see in Fig. I1 that the seasonal cycle derived by *in situ* global inversions over Australia is highly uncertain. One reason for the large disagreement between the *in situ* global inversions in Australia is the sparsity of observations. The existing Australian monitoring stations are around six, and not all are operational (Ziehn et al., 2016). Besides, these global *in situ* inversions rely on measurements that come from monitoring stations such as Cape Grim, a station designed to sample background maritime air masses much of the time, thus providing minimal constraint on Australian fluxes (Haverd et al., 2013c). The MIP OCO-2 disagreement is likely driven by the choice

of the prior flux, transport and data assimilation methodology used in the inversion (Crowell et al., 2019). For example, prior flux estimates used in global inversions rely on biosphere models such as CASA (van der Werf et al., 2017) or ORCHIDEE (Krinner et al., 2005). These models do not simulate well the NPP for grasslands (Wang et al., 2016) and hence underestimate the seasonality of the net ecosystem exchange for important ecosystems such as savanna and sparsely vegetated. This last point is critical for flux estimates over Australia because most of the land ecosystem is grassland and shrubs.

The analysis of the peak-to-peak seasonal variability of the nine OCO-2 global inversions individually shown in Table 4 indicates that, in general, November was the month in Australia with the highest carbon release to the atmosphere, similar to our posterior estimates. However, there is no unanimous agreement between them about the month with the largest carbon uptake. Analysing the ensemble mean of MIP OCO-2, we can see in Fig 14b that February and March were the months with the largest carbon uptake (-0.77 and -0.96 PgC $y^{-1}$ respectively), a close estimate to our posterior fluxes, which values were -0.77, -0.82 $y^{-1}$.

To further analyse our results, we also assess agreement between our posterior monthly spatial maps and these ten global inversions individually. We plotted monthly maps for each global inversion (see Section 7 in the Supplementary information). We found that our posterior flux distribution across Australia agree well with at least four global inversions (TM5, CAMS, PCTM, AMES; Supplementary, Figs. S14, S15, S16 and S18). We believe that this inter-comparison is valuable for Australia because it shows that our results are reliable with a better spatial resolution than global inversion.

The comparison with MIP-OCO2 strengthens our confidence that our inversion is capturing fluxes across Australia. It supports the surprising result that Australia was a carbon sink in 2015 despite the significant El Niño (ECMWF, 2020). El Niño is only one of several large-scale drivers of the Australian climate, and we have already noted positive rainfall anomalies associated with some strong sinks.

Evaluating the carbon fluxes assimilated in this study through the validation of our assimilated posterior field is difficult. We saw in Section 3.5.1 that most of the improvement of the prior concentration biases was seen in TCCON site (mainly Darwin site in northern Australia) in summer season compared to *in situ* observations. It is difficult to validate our posterior concentration field against monitoring stations located in coastal areas, such as Gunn Point or Cape Grim. These sites are strongly affected by oceanic fluxes, which are less restricted by the inversion when they are assimilated by only using LNLG observations. We demonstrated that adding OCO-2 glint observations to the inversion improves the biases considerably at these sites, but not for TCCON sites or sites located far way the ocean such as Burncluith and Ironbark.

We also found that adding OCO-2 ocean glint data to the inversion does not significantly alter the annual carbon sink estimated for the continent (0.36 PgC $y^{-1}$) compared to the estimate made by only using LNLG OCO-2 observations (0.41 PgC $y^{-1}$), suggesting that adding ocean glint observations does not strongly drive the continental carbon budget.

To assess the impact of biases in the lateral boundaries of the CMAQ domain, we performed two sensitivity experiments. In both experiments we add a constant offset of 0.25 ppm to each grid-cell of the BCs. In the first experiment we solve for the BCs and use LNLG. This induces a bias in our posterior annual flux of -0.8 PgC. Solving for the BCs and using LNLGOG observations reduces the bias further to (-0.4 PgC $y^{-1}$). Adding 0.25 ppm everywhere is an extreme test since the global assimilation fields we use are unlikely to have such systematic errors against data they assimilate. The results do highlight

the importance of solving for the BCs in a regional inverse system and also the importance of large domains with enough observations in a buffer region around our area of interest. These results are also reinforced by the good agreement that our assimilated fluxes have with MIP OCO-2.

There are still several methodological choices that are somewhat arbitrary in this study. Most important is the implied spatial resolution. This is determined by the correlation length used in the prior uncertainty as much as the resolution of CMAQ. Villalobos et al. (2020) showed the impact of this correlation length on posterior uncertainty and our choice makes a compromise between the information available from observations and avoidance of aggregation errors (Kaminski et al., 2001). A more important limitation is the restriction to one year. This will be addressed in a forthcoming study extending over the
OCO-2 data set.

Higher-resolution flux inversions assimilating satellite retrievals of greenhouse gas concentrations, as illustrated by this study, will be increasingly important in a world seeking climate solutions and a better understanding of the global carbon cycle. They will likely play a role in not just addressing questions of scientific interest but also in ongoing monitoring and assessment of emission targets. Australia, as a large and geographically isolated land-mass, with a terrestrial biosphere highly
responsive to climate drivers, offers an ideal testing ground for such flux inversions. The overall success of this study suggests great promise, especially in regions with sparse *in situ* networks.

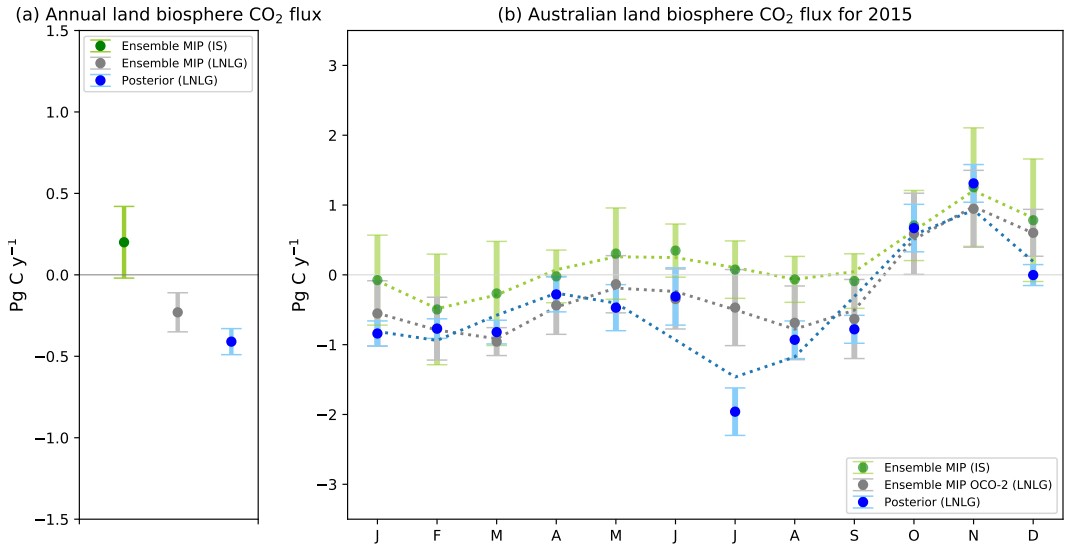

**Figure 14.** (a) Shows the ensemble annual mean of carbon fluxes derived by the *in situ* (green dot) and OCO-2 (LNLG) MIP (grey dot) global inversions, and the annual mean of the posterior fluxes estimated in this study (blue dot). The green and grey error bar represents the annual ensemble spread of the global models. In contrast, the blue error bar represents the uncertainties in the posterior flux calculated by different OSSE experiments estimated by Villalobos et al. (2020). (b) Shows the ensemble mean seasonal cycle of MIP *in situ* (green dots) and OCO-2 (LNLG) (grey dots) and the seasonal cycle of the posterior fluxes estimated in this study (blue dots) (All fluxes in this comparison are without fossil fuel emissions).

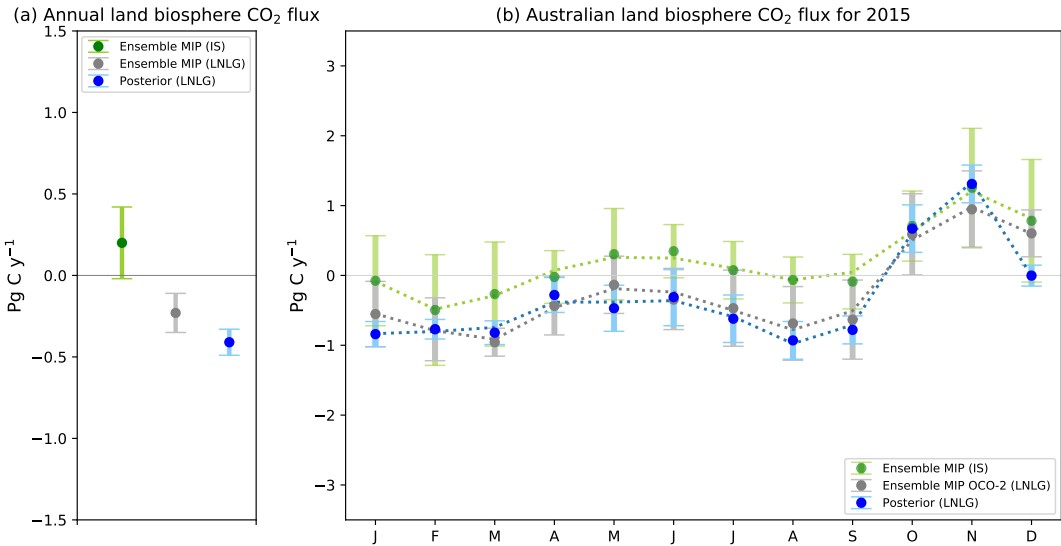

**Figure 15.** As in Fig. 14, but in this case, the monthly mean posterior flux (blue dot) for July has been interpolated between June and August.

**Table 4.** Summary of the peak-to-peak amplitude of our posterior terrestrial fluxes, prior fluxes, and terrestrial fluxes from nine different *in situ* and OCO-2 (LNLG) MIP global inversions. Units PgC y$^{-1}$

| Biosphere terrestrial fluxes | Acronym | Amplitude | Maximum month | Minimum month |
|---|---|---|---|---|
| MIP *in situ*[1] | AMES | 2.59 | December | February |
| | PCTM | 4.04 | November | March |
| | CAMS | 2.48 | July | February |
| | CMS-Flux | 2.83 | November | February |
| | CSU | 2.68 | November | January |
| | CT | 0.65 | June | September |
| | OU | 1.81 | December | September |
| | TM5-4DVAR | 2.59 | November | August |
| | UT | 0.98 | November | March |
| | Ensemble | 1.75 | November | February |
| MIP OCO-2 (LNLG)[2] | AMES | 2.27 | October | March |
| | PCTM | 1.96 | October | February |
| | CAMS | 2.19 | November | March |
| | CMS-Flux | 2.87 | November | March |
| | CSU | 2.85 | November | July |
| | CT | 0.81 | July | August |
| | OU | 1.90 | December | September |
| | TM5-4DVAR | 0.89 | November | August |
| | UT | 3.71 | November | March |
| | Ensemble | 1.90 | November | March |
| CMAQ OCO-2 (LNLG) | Posterior | 3.07 | November | July |
| | Posterior (July interpolated) | 2.13 | November | March |
| CABLE BIOS3 | Prior | 1.43 | November | January |

[1] Fig. I1, Appendix I, shows the seasonal cycle of the Australian carbon fluxes derived by the models contributing to *in situ* MIP inversions.

[2] Fig. I2, Appendix I, shows the seasonal cycle of the Australian carbon fluxes derived by the models contributing to the OCO-2 (LNLG) MIP global inversions.

## 5   Conclusions

We performed a four-dimensional variational data assimilation inversion to estimate Australian $CO_2$ fluxes for 2015. The inversion was based around the Community Multiscale Air Quality (CMAQ) transport-dispersion model and satellite data from the Orbiting Carbon Observatory-2 (OCO-2) (land nadir and glint data, version 9). Our regional inversion suggests that Australia was a carbon sink of -0.41 $\pm$ 0.03 PgC y$^{-1}$ compared to the prior estimate of 0.09 $\pm$ 0.20 PgC y$^{-1}$. We found a higher than average increase of land productivity (relative to 2000-2014) over the Savanna ecosystem (northern Australia) during summer, leading to most of the carbon uptake in this ecosystem. Sparsely vegetated ecosystem is the most extensive ecosystem over Australia, and also showed a slight increase of land productivity in the Autumn and Winter seasons in the western region of Australia, which was also driven by an increase of vegetation productivity in response to positive rainfall anomalies in this period. We also found that the higher carbon uptake by our inversion (relative to the prior) was due to an underestimation of GPP simulated by the CABLE-BIOS3 model.

Evaluation with the TCCON Darwin site shows that our inversion is able to reduce biases mainly in the summer period compared to the winter season. Reduction of the biases at TCCON Lauder and Wollongong showed a very slight systematic decrease, mostly because both sites are strongly affected by ocean winds and the reduced number of OCO-2 sounding passing over these sites in some periods 2015. Posterior column-integrated simulations at coastal monitoring sites are challenging to validate because they are strongly influenced by ocean fluxes, which were assigned small uncertainties in our inversion. Comparison with *in situ* data was also a challenge mainly over oceanic monitoring stations such as Cape Grim and Gunn Point sites, which are also strongly impacted by ocean fluxes. Comparison with monitoring stations over land such as Ironbark and Burncluith also shows difficulties in simultaneously matching column-integrated and surface data, most likely linked to model vertical transport. The scarcity of *in situ* observations across the Australian continent, mainly over the savanna and sparsely vegetated ecosystem, restrict our ability to conclude with confidence whether the stronger carbon sink (relative to the prior) found in those ecosystems are real or not. However, the comparison with the annual and monthly ensemble means of the OCO-2 MIP intercomparison are encouraging and support our results.

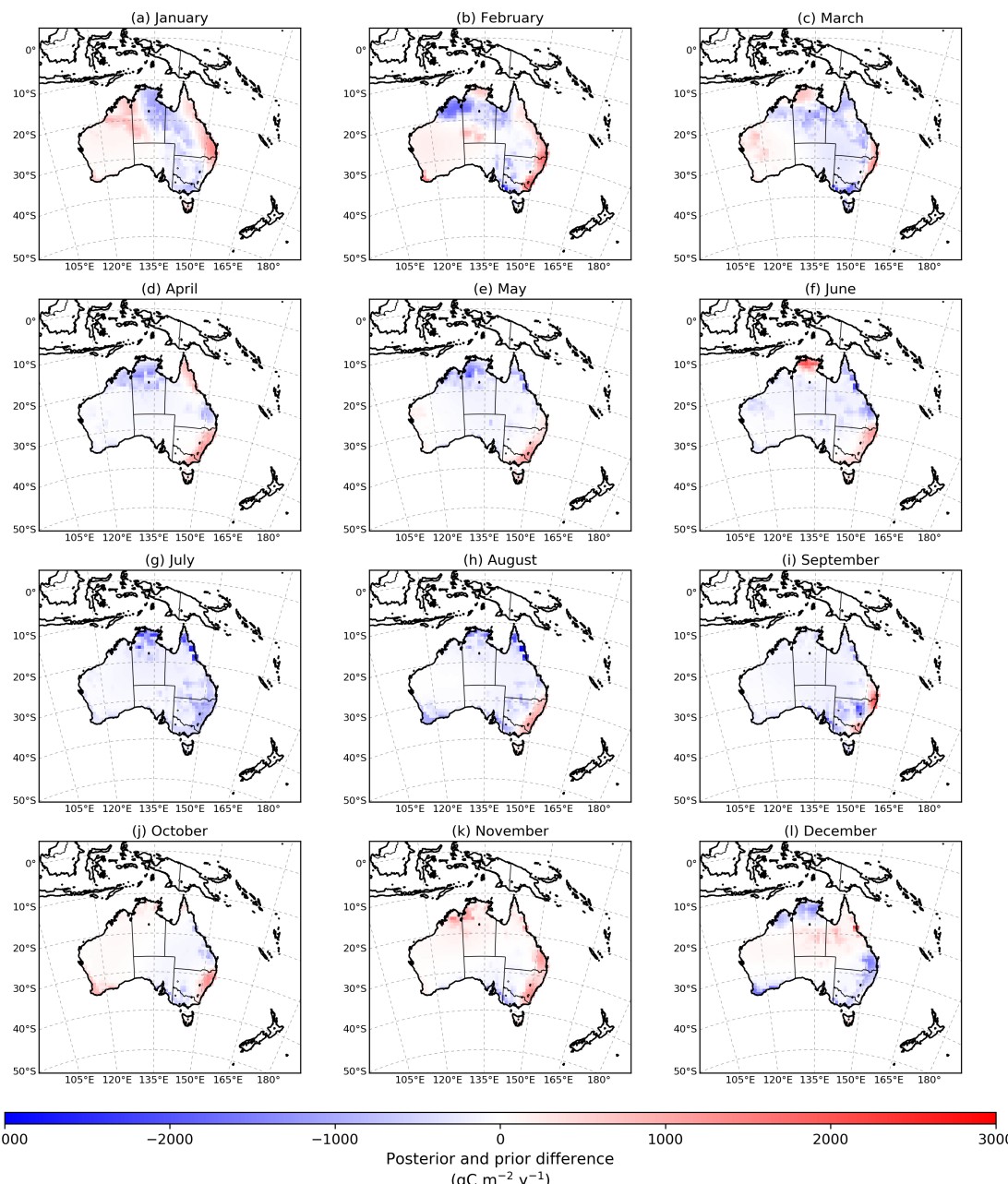

**Figure A1.** Spatial pattern of the differences between posterior and prior fluxes for 2015.

## Appendix B: Monthly time series of the Australian land biosphere prior and posterior CO$_2$ flux over six bioclimatic regions

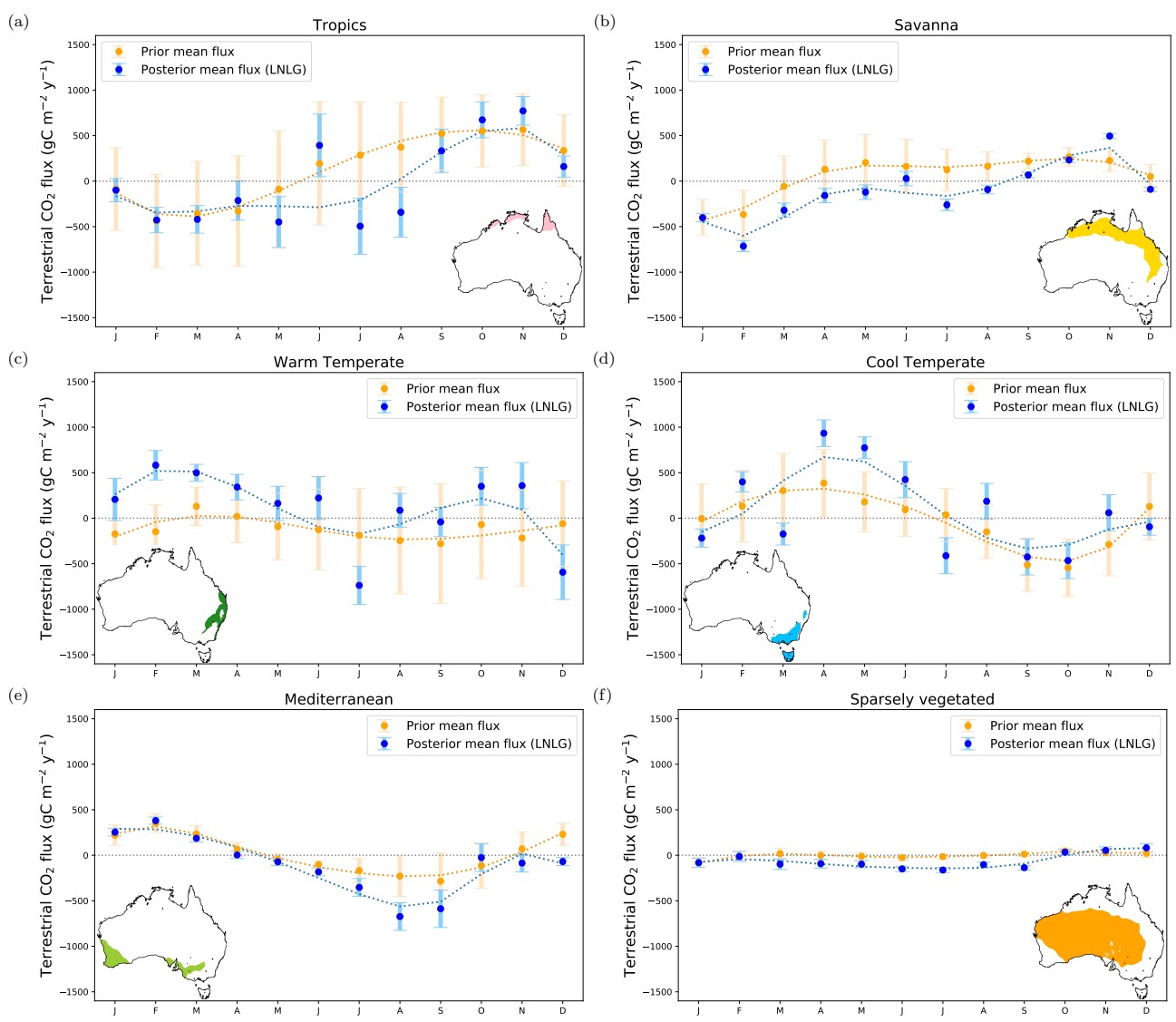

**Figure B1.** Monthly time series of the Australian land biosphere prior and posterior CO$_2$ flux and their uncertainties in gC m$^{-2}$y$-1$ over six bioclimatic regions. The prior and posterior estimates do not include fossil fuel emissions.

## Appendix C: Spatial distribution of OCO-2 data across Australia

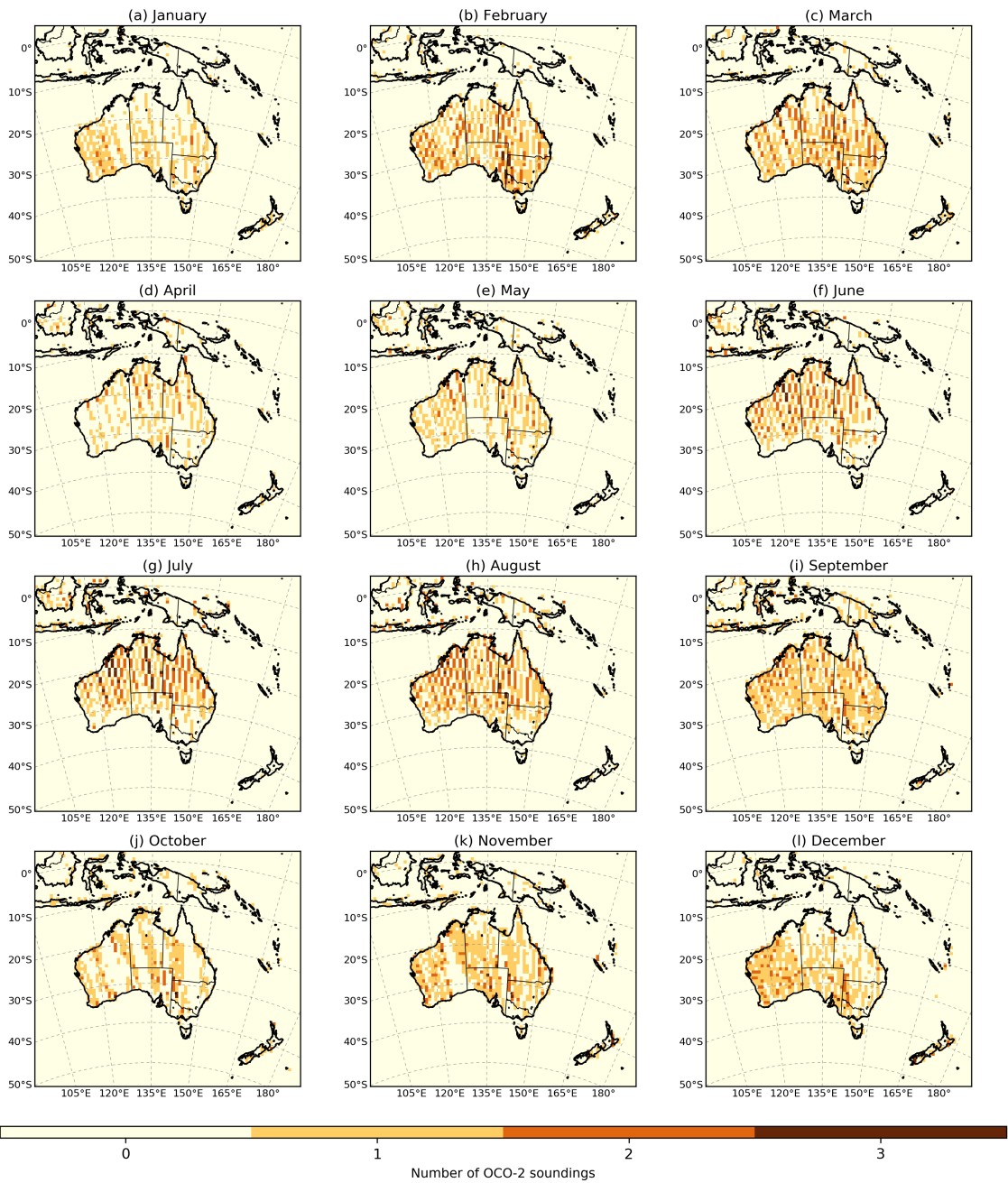

**Figure C1.** Spatial distribution of OCO-2 soundings (LNLG) over the CMAQ domain for 2015.

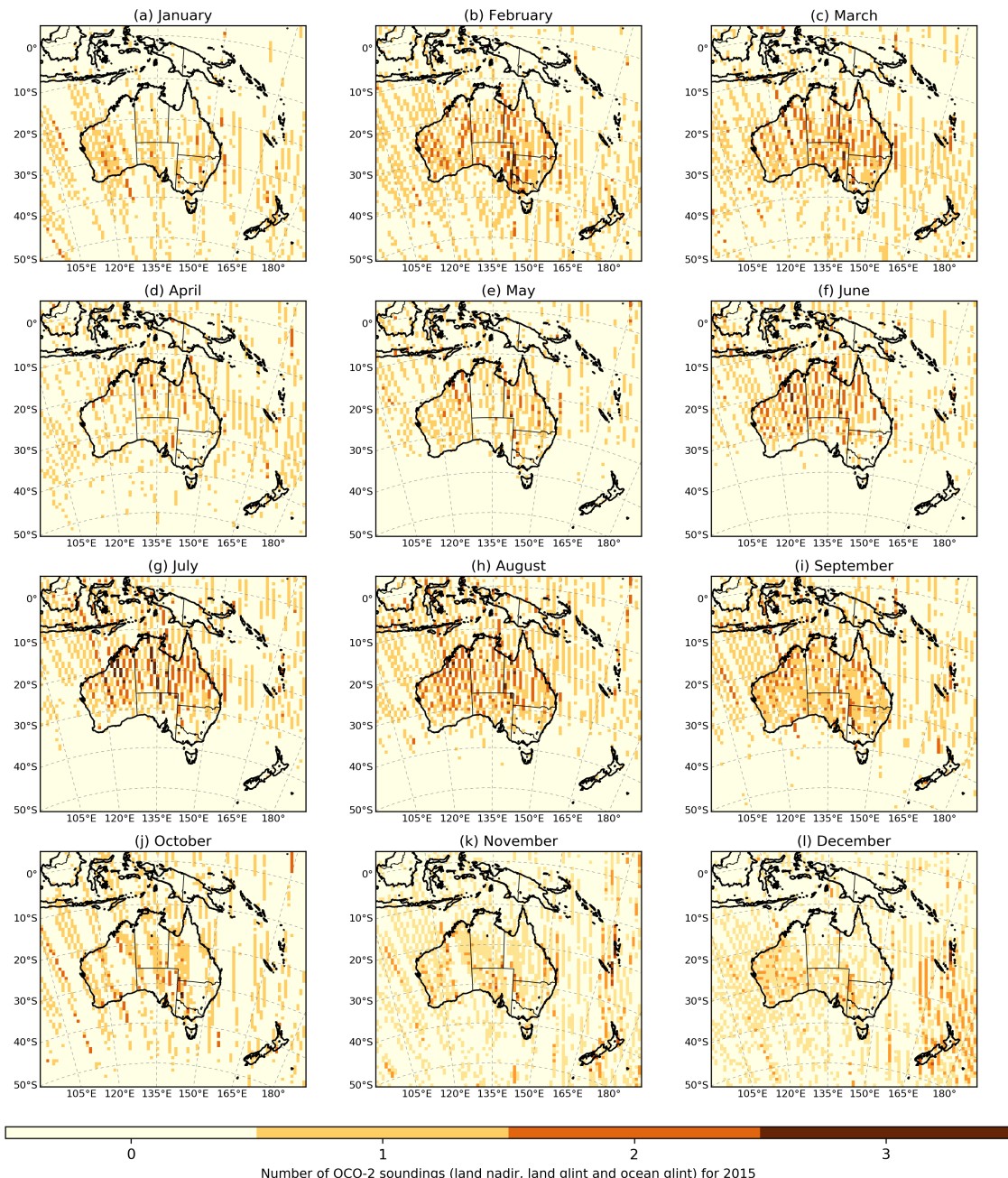

**Figure C2.** Spatial distribution of OCO-2 soundings (LNLGOG) over the CMAQ domain for 2015.

## Appendix D: Spatial distribution of the prior and posterior uncertainties across Australia.

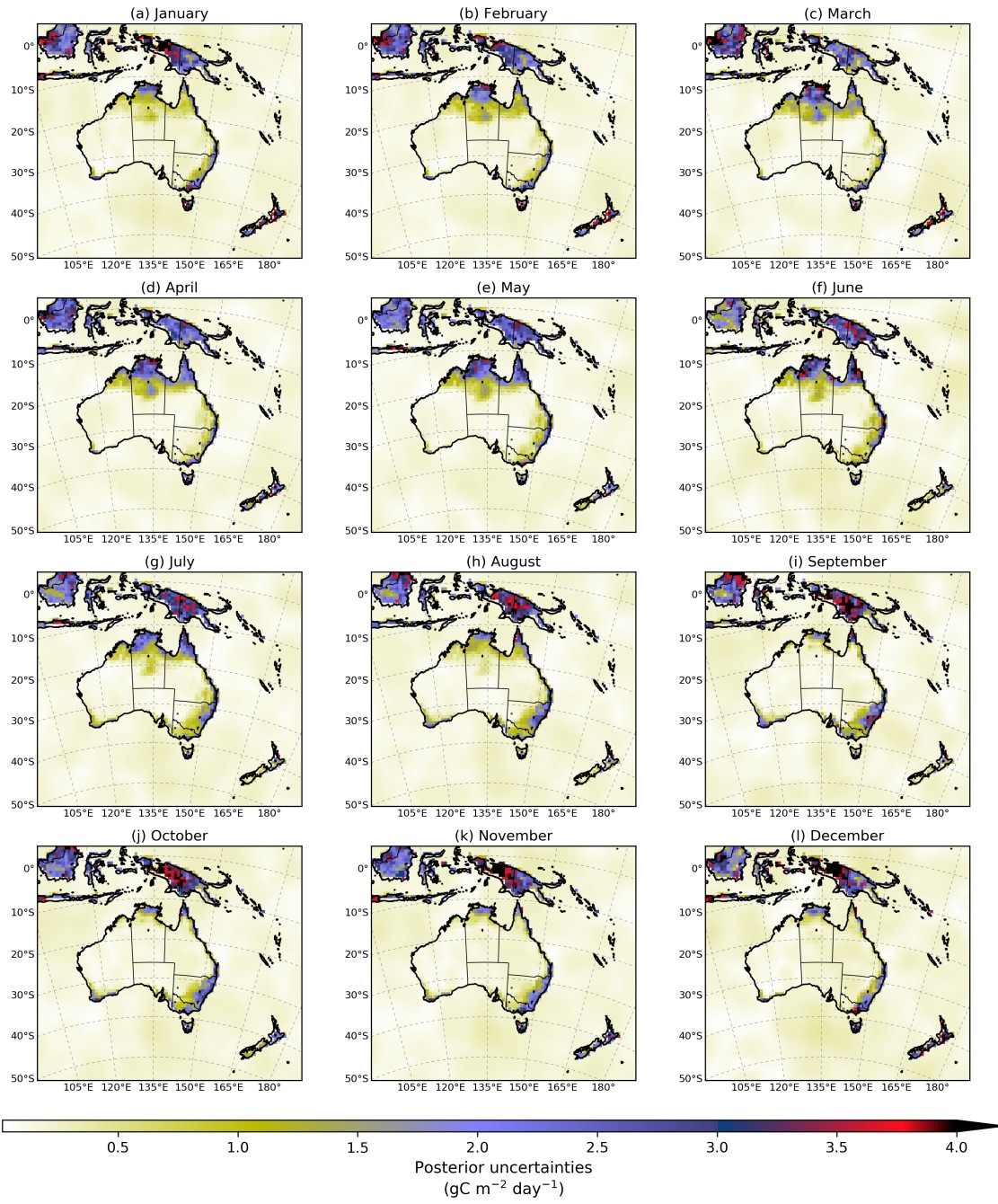

**Figure D1.** Prior uncertainties accounting for the major terms in the $CO_2$ budget (anthropogenic fluxes, fires, land, and ocean exchange).

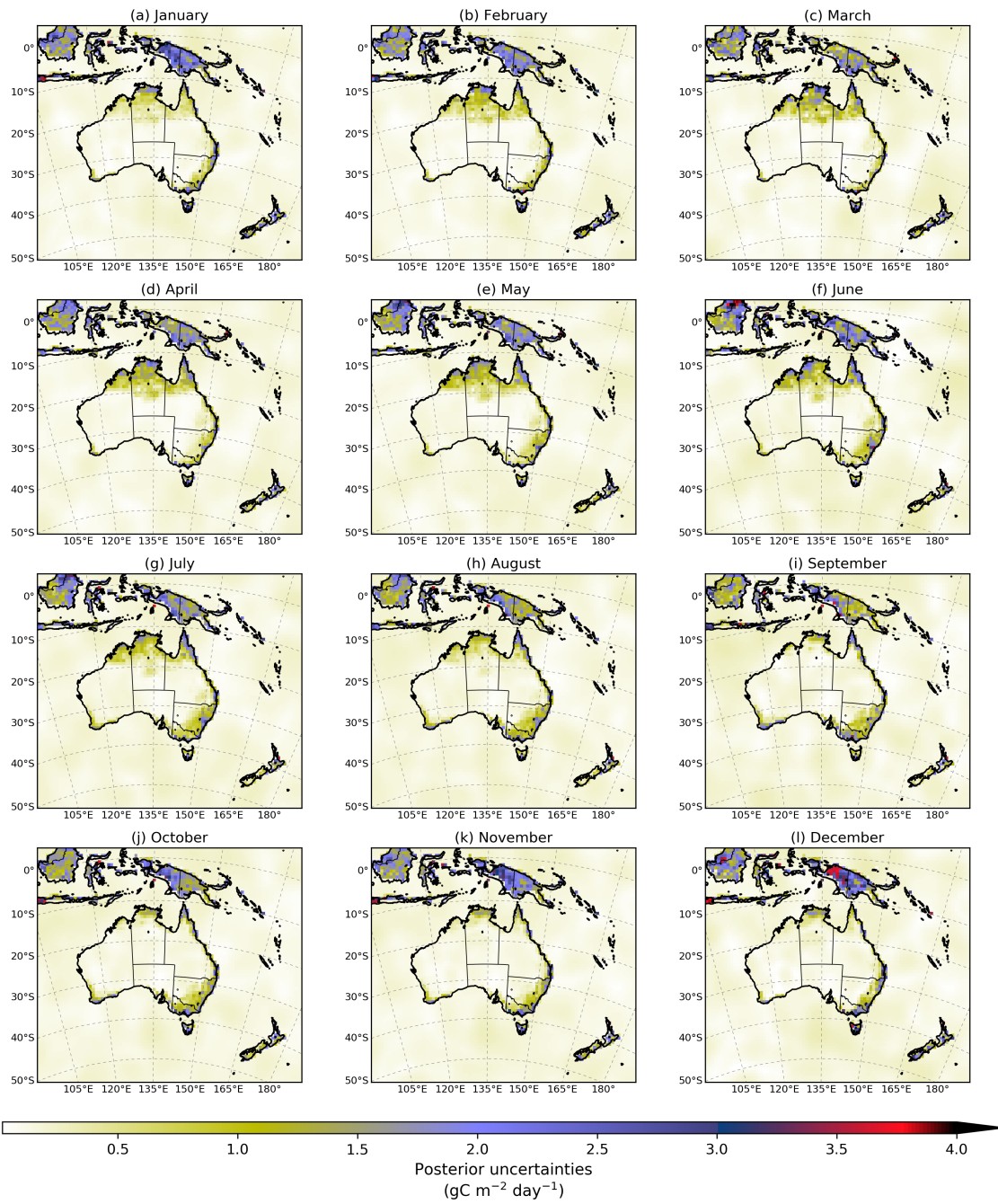

**Figure D2.** Posterior uncertainties calculated by OSSEs experiments in Villalobos et al. (2020)

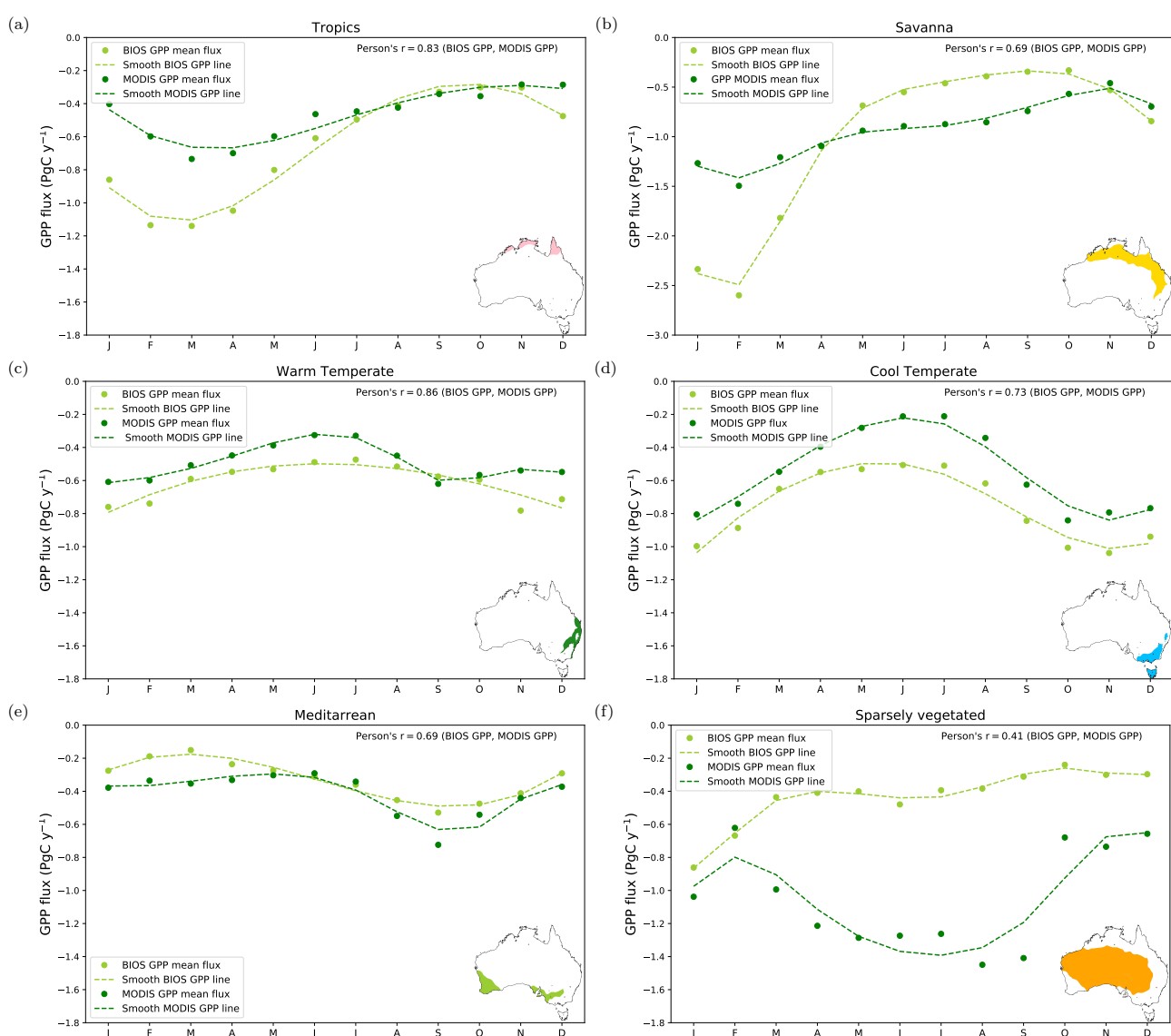

**Figure E1.** Time series of monthly mean of CABLE-BIOS3 GPP and MODIS GPP.

# Appendix F: Posterior fluxes assimilated by using LNLGOG satellite observations

**Table F1.** Australia terrestrial carbon fluxes estimated using LNLG and LNLGOG for 2015 (Units: PgC $y^{-1}$).

| Months | | Prior | Posterior | | Posterior |
| --- | --- | --- | --- | --- | --- |
| YYYY-MM | Prior | Uncertainties | LNLG | LNLGOG | Uncertainties |
| 2015-01 | -0.89 | 0.75 | -0.84 | -0.51 | 0.18 |
| 2015-02 | -0.56 | 0.75 | -0.77 | -1.13 | 0.14 |
| 2015-03 | 0.12 | 0.74 | -0.82 | -0.59 | 0.17 |
| 2015-04 | 0.24 | 0.69 | -0.28 | -0.23 | 0.25 |
| 2015-05 | 0.15 | 0.64 | -0.47 | -0.23 | 0.33 |
| 2015-06 | 0.15 | 0.59 | -0.31 | -0.54 | 0.41 |
| 2015-07 | 0.09 | 0.62 | -1.75 | -1.96 | 0.34 |
| 2015-08 | 0.13 | 0.64 | -0.93 | -0.97 | 0.27 |
| 2015-09 | 0.16 | 0.66 | -0.78 | -0.24 | 0.20 |
| 2015-10 | 0.53 | 0.69 | 0.67 | 0.20 | 0.34 |
| 2015-11 | 0.54 | 0.73 | 1.31 | 1.56 | 0.27 |
| 2015-12 | 0.422 | 0.76 | 0.00 | 0.40 | 0.15 |

## Appendix G: TCCON comparison

**Table G1.** Analysis of the residual between CMAQ prior and posterior simulation and TCCON Darwin site for 2015. Averaged bias (Bias), Root-mean-square error (RMSE) and Pearson's coefficient (R).

| | | Darwin | | | | | | | |
|---|---|---|---|---|---|---|---|---|---|
| Months | | Prior | | Posterior LNLG | | | Posterior LNLGOG | | |
| YYYY-MM | Bias | RMSE | R | Bias | RMSE | R | Bias | RMSE | R |
| 2015-01 | 0.12 | 0.51 | 0.81 | -0.04 | 0.82 | 0.75 | 0.08 | 0.58 | 0.61 |
| 2015-02 | 0.69 | 0.85 | 0.78 | 0.38 | 0.63 | 0.78 | 0.43 | 0.69 | 0.70 |
| 2015-03 | 0.93 | 1.10 | 0.14 | 0.18 | 0.59 | 0.29 | 0.64 | 0.88 | 0.26 |
| 2015-04 | 0.85 | 0.94 | 0.38 | 0.60 | 0.74 | 0.42 | 0.62 | 0.74 | 0.42 |
| 2015-05 | 0.97 | 1.05 | 0.37 | 0.90 | 0.99 | 0.52 | 0.90 | 0.96 | 0.58 |
| 2015-06 | 0.90 | 0.97 | 0.21 | 1.24 | 1.27 | 0.23 | 1.12 | 1.16 | 0.12 |
| 2015-07 | 1.51 | 1.55 | -0.18 | 1.07 | 1.10 | 0.22 | 0.92 | 0.96 | 0.17 |
| 2015-08 | 1.44 | 1.46 | 0.34 | 1.06 | 1.10 | 0.35 | 1.07 | 1.11 | 0.33 |
| 2015-09 | 1.12 | 1.16 | 0.02 | 0.81 | 0.86 | 0.10 | 0.77 | 0.82 | 0.21 |
| 2015-10 | 0.55 | 0.63 | 0.53 | 0.63 | 0.69 | 0.62 | 0.54 | 0.60 | 0.64 |
| 2015-11 | -0.25 | 0.51 | 0.66 | 0.11 | 0.42 | 0.75 | -0.24 | 0.53 | 0.64 |
| 2015-12 | -0.34 | 0.48 | 0.18 | -0.02 | 0.31 | 0.26 | -0.30 | 0.45 | 0.28 |

**Table G2.** Analysis of the residual between CMAQ prior and posterior simulation and TCCON Wollongong site for 2015. Averaged bias (Bias), Root-mean-square error (RMSE) and Pearson's coefficient (R).

| Wollongong | | | | | | | | | |
|---|---|---|---|---|---|---|---|---|---|
| Months | Prior | | | Posterior LNLG | | | Posterior LNLGOG | | |
| YYYY-MM | Bias | RMSE | R | Bias | RMSE | R | Bias | RMSE | R |
| 2015-01 | -0.04 | 0.72 | 0.21 | 0.07 | 0.75 | 0.23 | 0.17 | 0.89 | -0.04 |
| 2015-02 | -0.21 | 0.56 | 0.48 | 0.16 | 0.63 | 0.51 | 0.20 | 0.64 | 0.44 |
| 2015-03 | 0.66 | 0.94 | 0.19 | 0.51 | 0.88 | 0.16 | 0.67 | 0.98 | 0.15 |
| 2015-04 | 0.72 | 0.96 | 0.07 | 0.82 | 1.06 | 0.15 | 0.71 | 1.00 | 0.05 |
| 2015-05 | 1.26 | 1.40 | 0.07 | 1.54 | 1.72 | 0.02 | 1.67 | 1.86 | 0.05 |
| 2015-06 | 1.41 | 1.53 | 0.68 | 1.61 | 1.72 | 0.68 | 1.59 | 1.71 | 0.66 |
| 2015-07 | 1.37 | 1.56 | 0.32 | 1.14 | 1.38 | 0.28 | 1.10 | 1.35 | 0.28 |
| 2015-08 | 1.42 | 1.57 | 0.25 | 1.61 | 1.76 | 0.28 | 1.76 | 1.94 | 0.35 |
| 2015-09 | 1.19 | 1.44 | 0.16 | 1.11 | 1.42 | 0.19 | 1.37 | 1.68 | 0.22 |
| 2015-10 | 0.07 | 0.72 | 0.03 | 0.29 | 0.83 | 0.00 | 0.41 | 0.96 | -0.06 |
| 2015-11 | -0.74 | 1.22 | -0.08 | -0.40 | 1.13 | -0.05 | -0.28 | 1.19 | -0.07 |
| 2015-12 | -0.45 | 0.69 | 0.14 | -0.60 | 0.85 | -0.03 | -0.46 | 0.76 | 0.04 |

**Table G3.** Analysis of the residual between CMAQ prior and posterior simulation and TCCON Wollongong site for 2015. Averaged bias (Bias), Root-mean-square error (RMSE) and Pearson's coefficient (R).

| Wollongong* | | | | | | | | | |
|---|---|---|---|---|---|---|---|---|---|
| Months | | Prior | | | Posterior LNLG | | | Posterior LNLGOG | |
| YYYY-MM | Bias | RMSE | | Bias | RMSE | R | Bias | RMSE | R |
| 2015-01 | 0.01 | 0.75 | 0.14 | 0.10 | 0.79 | 0.15 | 0.21 | 0.94 | -0.11 |
| 2015-02 | -0.21 | 0.56 | 0.48 | 0.16 | 0.63 | 0.51 | 0.20 | 0.64 | 0.44 |
| 2015-03 | 0.65 | 0.93 | 0.19 | 0.50 | 0.87 | 0.17 | 0.67 | 0.97 | 0.16 |
| 2015-04 | 0.69 | 0.92 | 0.12 | 0.79 | 1.02 | 0.20 | 0.68 | 0.96 | 0.10 |
| 2015-05 | 1.12 | 1.26 | 0.24 | 1.39 | 1.54 | 0.19 | 1.50 | 1.66 | 0.18 |
| 2015-06 | 1.18 | 1.28 | 0.78 | 1.36 | 1.46 | 0.78 | 1.33 | 1.43 | 0.77 |
| 2015-07 | 1.17 | 1.35 | 0.45 | 0.94 | 1.18 | 0.40 | 0.91 | 1.16 | 0.39 |
| 2015-08 | 1.30 | 1.44 | 0.31 | 1.51 | 1.66 | 0.32 | 1.68 | 1.87 | 0.37 |
| 2015-09 | 1.18 | 1.43 | 0.16 | 1.10 | 1.41 | 0.20 | 1.36 | 1.67 | 0.22 |
| 2015-10 | 0.07 | 0.72 | 0.03 | 0.29 | 0.83 | 0.00 | 0.41 | 0.96 | -0.06 |
| 2015-11 | -0.74 | 1.17 | 0.02 | -0.41 | 1.05 | 0.07 | -0.31 | 1.10 | 0.04 |
| 2015-12 | -0.44 | 0.71 | 0.16 | -0.59 | 0.88 | 0.01 | -0.46 | 0.78 | 0.09 |

[*] Wollongong TCCON data filtered by solar zenith angles $\leq 40$ and $\geq 50$ degrees.

**Table G4.** Analysis of the residual between CMAQ prior and posterior simulation and TC-CON Lauder site for 2015. Averaged bias (Bias), Root-mean-square error (RMSE) and Pearson's coefficient (R).

| | | Lauder | | | | | | | |
|---|---|---|---|---|---|---|---|---|---|
| Months | | Prior | | | Posterior LNLG | | | Posterior LNLGOG | |
| YYYY-mm | Bias | RMSE | R | Bias | RMSE | R | Bias | RMSE | R |
| 2015-01 | 0.48 | 0.58 | 0.31 | 0.71 | 0.85 | 0.06 | 0.66 | 0.78 | 0.23 |
| 2015-02 | 0.61 | 0.74 | 0.22 | 1.03 | 1.17 | 0.34 | 1.19 | 1.33 | 0.41 |
| 2015-03 | 0.54 | 0.62 | 0.51 | 0.73 | 0.84 | 0.53 | 0.84 | 0.93 | 0.57 |
| 2015-04 | 0.50 | 0.59 | 0.77 | 0.51 | 0.60 | 0.79 | 0.66 | 0.74 | 0.82 |
| 2015-05 | 0.82 | 0.89 | 0.30 | 0.83 | 0.90 | 0.23 | 0.89 | 0.96 | 0.23 |
| 2015-06 | 0.65 | 0.86 | 0.60 | 0.61 | 0.82 | 0.56 | 0.64 | 0.84 | 0.55 |
| 2015-07 | 0.69 | 0.82 | 0.79 | 0.64 | 0.79 | 0.76 | 0.68 | 0.83 | 0.76 |
| 2015-08 | 0.57 | 0.64 | 0.64 | 0.57 | 0.64 | 0.66 | 0.69 | 0.75 | 0.62 |
| 2015-09 | 0.71 | 0.73 | 0.83 | 0.63 | 0.67 | 0.83 | 0.95 | 1.01 | 0.72 |
| 2015-10 | 0.75 | 0.82 | 0.65 | 0.74 | 0.82 | 0.59 | 0.89 | 0.95 | 0.61 |
| 2015-11 | 0.52 | 0.72 | 0.36 | 0.43 | 0.65 | 0.37 | 0.44 | 0.63 | 0.34 |
| 2015-12 | 0.71 | 0.76 | 0.79 | 0.77 | 0.81 | 0.81 | 0.71 | 0.75 | 0.79 |

# Appendix H: *in-situ* comparison

**Table H1.** Analysis of the residual between CMAQ prior and posterior simulation and Gunn Point site for 2015. Averaged bias (Bias), Root-mean-square error (RMSE) and Pearson's coefficient (R).

| | | Gunn Point | | | | | | | |
|---|---|---|---|---|---|---|---|---|---|
| Months | | Prior | | Posterior LNLG | | | Posterior LNLGOG | | |
| YYYY-MM | Bias | RMSE | R | Bias | RMSE | R | Bias | RMSE | R |
| 2015-01 | -1.16 | 4.83 | 0.37 | -2.11 | 4.96 | 0.26 | -1.68 | 4.59 | 0.06 |
| 2015-02 | -2.88 | 4.73 | 0.41 | -3.55 | 5.14 | 0.47 | -2.68 | 4.18 | 0.49 |
| 2015-03 | -1.93 | 4.21 | -0.06 | -3.36 | 5.17 | -0.06 | 0.65 | 3.79 | 0.04 |
| 2015-04 | -1.07 | 2.92 | 0.33 | -2.74 | 3.96 | 0.28 | -2.05 | 3.49 | 0.36 |
| 2015-05 | -1.76 | 2.78 | 0.35 | -3.65 | 4.24 | 0.53 | -2.43 | 3.34 | 0.54 |
| 2015-06 | -0.96 | 1.68 | 0.29 | 1.90 | 2.62 | 0.31 | 0.77 | 1.96 | 0.34 |
| 2015-07 | -1.34 | 8.84 | 0.00 | -7.71 | 17.93 | 0.06 | -8.40 | 18.20 | 0.08 |
| 2015-08 | 1.70 | 2.43 | 0.41 | -2.88 | 3.29 | 0.25 | -1.52 | 2.39 | 0.25 |
| 2015-09 | 1.81 | 2.13 | 0.28 | -0.32 | 1.35 | 0.04 | 0.28 | 1.58 | -0.04 |
| 2015-10 | 2.19 | 2.44 | 0.15 | 3.24 | 3.57 | -0.03 | 2.05 | 2.40 | -0.02 |
| 2015-11 | -0.52 | 2.30 | -0.67 | 0.66 | 2.33 | -0.63 | -0.85 | 2.61 | -0.75 |
| 2015-12 | -2.69 | 3.34 | 0.38 | -4.03 | 4.50 | 0.36 | -4.02 | 4.51 | 0.34 |

**Table H2.** Analysis of the residual between CMAQ prior and posterior simulation and Ironbark site for 2015. Averaged bias (Bias), Root-mean-square error (RMSE) and Pearson's coefficient (R).

| | | Ironbark | | | | | | | |
|---|---|---|---|---|---|---|---|---|---|
| Months | | Prior | | | Posterior LNLG | | | Posterior LNLGOG | | |
| YYYY-MM | Bias | RMSE | R | Bias | RMSE | R | Bias | RMSE | R |
| 2015-01 | -1.61 | 2.28 | 0.32 | 0.43 | 2.33 | -0.12 | 1.00 | 3.03 | -0.42 |
| 2015-02 | -1.07 | 1.30 | 0.44 | 0.71 | 1.14 | 0.32 | 1.07 | 1.31 | 0.50 |
| 2015-03 | -0.34 | 2.85 | 0.35 | -1.32 | 3.61 | -0.08 | -0.87 | 3.64 | -0.04 |
| 2015-04 | -1.11 | 2.13 | 0.50 | -1.48 | 2.52 | 0.39 | -1.32 | 2.22 | 0.54 |
| 2015-05 | -2.15 | 2.77 | 0.37 | -1.56 | 2.45 | 0.29 | -1.18 | 2.29 | 0.33 |
| 2015-06 | -2.12 | 2.63 | 0.46 | -2.29 | 3.26 | 0.02 | -1.06 | 2.78 | -0.01 |
| 2015-07 | -0.35 | 1.66 | 0.49 | -2.33 | 2.77 | 0.54 | -2.80 | 3.24 | 0.61 |
| 2015-08 | 1.44 | 2.55 | 0.26 | 0.92 | 2.84 | 0.01 | 0.95 | 2.80 | 0.21 |
| 2015-09 | 1.27 | 1.83 | 0.55 | 1.58 | 2.40 | 0.49 | 1.63 | 2.65 | 0.46 |
| 2015-10 | -0.81 | 2.04 | 0.28 | -0.90 | 2.04 | 0.29 | -1.34 | 2.35 | 0.16 |
| 2015-11 | -2.28 | 2.86 | 0.53 | 0.05 | 1.93 | 0.48 | 0.80 | 2.27 | 0.36 |
| 2015-12 | -1.50 | 2.77 | 0.50 | -3.33 | 4.34 | 0.28 | -4.10 | 5.00 | 0.21 |

**Table H3.** Analysis of the residual between CMAQ prior and posterior simulation and Burn-cluith for 2015. Averaged bias (Bias), Root-mean-square error (RMSE) and Pearson's coefficient (R).

| | Burncluith | | | | | | | | |
|---|---|---|---|---|---|---|---|---|---|
| Months | Prior | | | Posterior LNLG | | | Posterior LNLGOG | | |
| YYYY-MM | Bias | RMSE | R | Bias | RMSE | R | Bias | RMSE | R |
| 2015-01 | - | - | - | - | - | - | - | - | - |
| 2015-02 | - | - | - | - | - | - | - | - | - |
| 2015-03 | - | - | - | - | - | - | - | - | - |
| 2015-04 | - | - | - | - | - | - | - | - | - |
| 2015-05 | - | - | - | - | - | - | - | - | - |
| 2015-06 | - | - | - | - | - | - | - | - | - |
| 2015-07 | 0.86 | 2.13 | 0.41 | -1.04 | 2.34 | 0.29 | -1.15 | 2.71 | 0.28 |
| 2015-08 | 2.20 | 3.05 | 0.39 | 1.56 | 3.10 | 0.10 | 1.77 | 3.26 | 0.19 |
| 2015-09 | 2.05 | 2.71 | 0.43 | 2.10 | 3.18 | 0.27 | 2.22 | 3.52 | 0.22 |
| 2015-10 | 0.21 | 1.85 | 0.26 | 0.01 | 1.88 | 0.20 | -0.31 | 2.07 | 0.02 |
| 2015-11 | -1.24 | 2.23 | 0.73 | 1.22 | 2.30 | 0.69 | 1.92 | 3.00 | 0.54 |
| 2015-12 | 0.37 | 2.54 | 0.45 | -1.28 | 3.24 | 0.21 | -2.20 | 3.77 | 0.11 |

**Table H4.** Analysis of the residual between CMAQ prior and posterior simulation and Cape Grim for 2015. Averaged bias (Bias), Root-mean-square error (RMSE) and Pearson's coefficient (R).

| | | | | | | | | | |
|---|---|---|---|---|---|---|---|---|---|
| | | | | Posterior | | | Posterior | | |
| Months | | Prior | | LNLG | | | LNLGOG | | |
| YYYY-MM | Bias | RMSE | R | Bias | RMSE | R | Bias | RMSE | R |
| 2015-01 | -2.31 | 3.38 | 0.28 | -2.33 | 3.22 | 0.28 | -0.54 | 2.46 | 0.27 |
| 2015-02 | -2.61 | 3.68 | 0.57 | -2.59 | 3.91 | 0.53 | -1.44 | 3.83 | 0.40 |
| 2015-03 | -1.25 | 2.02 | 0.53 | -2.70 | 3.07 | 0.29 | -0.97 | 1.72 | 0.32 |
| 2015-04 | -2.33 | 3.22 | 0.41 | -2.16 | 3.54 | 0.19 | -2.69 | 3.78 | 0.17 |
| 2015-05 | -1.85 | 3.27 | 0.36 | -0.60 | 2.82 | 0.46 | -0.22 | 2.64 | 0.54 |
| 2015-06 | -1.80 | 2.84 | 0.14 | -0.88 | 2.28 | 0.20 | -0.76 | 2.22 | 0.22 |
| 2015-07 | -0.96 | 2.05 | 0.10 | -2.18 | 3.20 | -0.03 | -1.85 | 3.19 | -0.01 |
| 2015-08 | -1.91 | 2.93 | -0.05 | -2.12 | 3.22 | 0.02 | -1.82 | 3.06 | 0.12 |
| 2015-09 | -2.22 | 3.60 | -0.02 | -4.18 | 4.94 | 0.16 | -1.69 | 3.93 | 0.02 |
| 2015-10 | -2.37 | 3.47 | 0.08 | -2.52 | 3.75 | 0.00 | -2.42 | 3.84 | -0.07 |
| 2015-11 | -2.35 | 3.32 | 0.34 | -3.07 | 4.28 | -0.06 | -1.88 | 3.65 | -0.28 |
| 2015-12 | -1.88 | 2.54 | 0.58 | -2.34 | 2.89 | 0.49 | -1.32 | 2.36 | 0.49 |

## Appendix I:  Australian fluxes derived by MIP *in situ* and OCO-2 (LNLG) global inversions for 2015

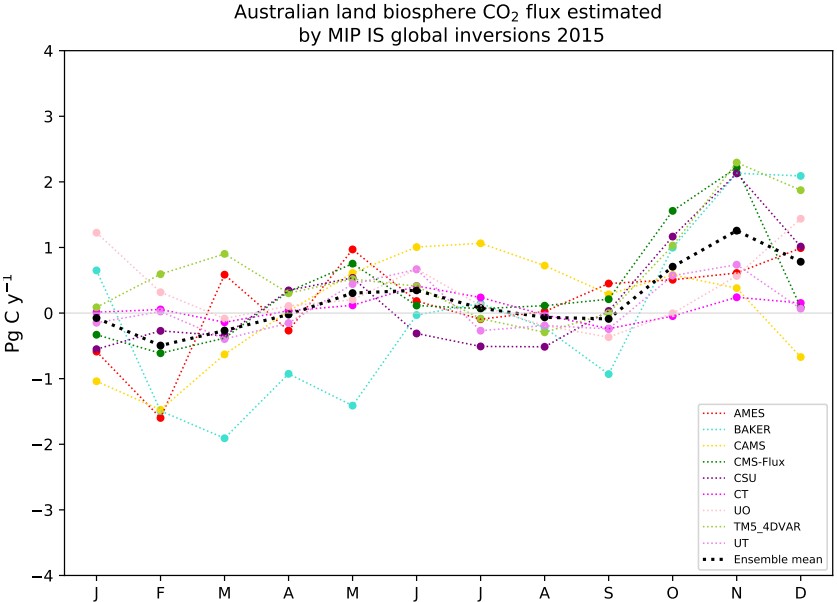

**Figure I1.** Time series of monthly mean carbon fluxes derived by MIP *in situ* (IS) global inversion for 2015.

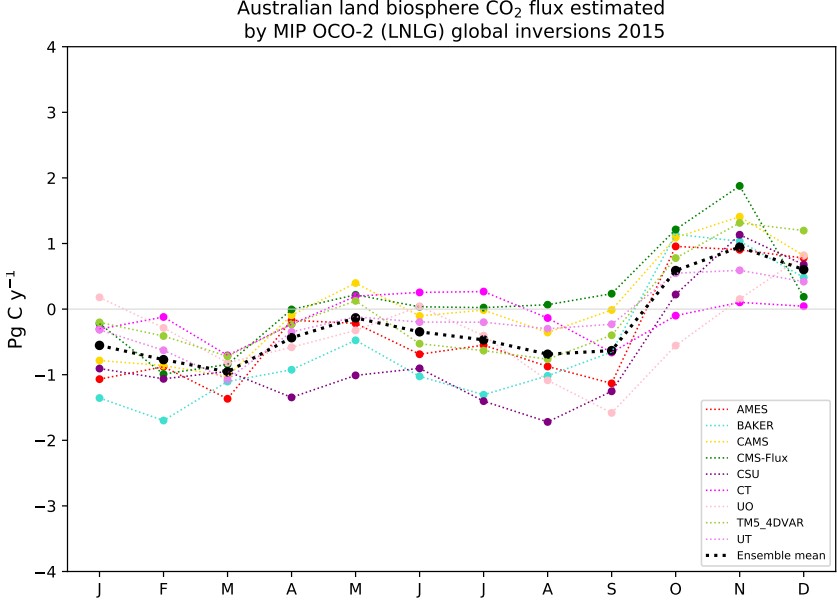

**Figure I2.** Time series of monthly mean carbon fluxes derived by MIP OCO-2 (LNLG) global inversion for 2015.

*Data availability.* The surface gridded fluxes are available in Zenodo repository under the identifier 10.5281/zenodo.5636113.

*Code availability.* The code of the inversion system is available at https://github.com/steven-thomas/py4dvar (Thomas, 2020).

*Author contributions.* YV prepared all the input data required to run the inversion system and performed data analysis of the fluxes. YV was responsible for post-processing the TCCON and *in-situ* measurements, then developing the paper and figures. ST was the principal developer of the inversion system code. PR and JR also contributed to developing the inversion code, provided guidance for the manuscript's preparation and interpretation of the results. JK and VH ran CABLE-BIOS3 and provided the biosphere fluxes required for the inversion. JK reviewed and commented on the final manuscript. ZL provided data from the ground-based *in-situ* measurements (Cape Grim, Ironbark, Burncluith and Gunn Point) and gave comments on the paper. DP reviewed and comments on the TCCON Lauder site. ND and DG reviewed the final manuscript.

*Competing interests.* The authors declare that they have no conflict of interest.

*Acknowledgements.* This work was funded by the National Agency for Research and Development (ANID) scholarship, Becas Chile (grant no. 72170210) and supported by the Education Infrastructure Fund of the Australian Government, and the Australian Research Council (ARC) of the Centre of Excellence for Climate Extreme (CLEX, grant no. CE170100023). The authors would like to thank the to all MIP inverse modellers: Matthew Johnson, Frédéric Chevallier, Junjie Liu, Andrew Schuh, Andy Jacobson, Sean Crowell, David Baker, Sourish Basu, and Feng Deng for contributing with their *in situ* and OCO-2 (LNLG) global inversion product. The authors would also like to thank the institutions that provide data from the TCCON sites. Darwin and Wollongong TCCON stations are supported by ARC grants DP160100598, LE0668470, DP140101552, DP110103118 and DP0879468, and Darwin through NASA grants NAG5-12247 and NNG05-GD07G. NMD is funded by an ARC Future Fellowship, FT180100327. This research was undertaken with the assistance of resources and services from the National Computational Infrastructure (NCI), which is supported by the Australian Government, and the resources of the High-performance Computing Centre of the University of Melbourne, SPARTAN.

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
