# Peer review of "Was Australia a sink or source of CO2 in 2015? Data assimilation using OCO-2 satellite measurements"

_Atmospheric Chemistry and Physics, 2021_

## Referee Comment (RC1)

The manuscript by *Villalobos et al.* (2021) evaluates terrestrial biosphere carbon dioxide ($CO_2$) fluxes in Australia for the year 2015. This analysis was conducted with a regional-scale inverse modeling framework which assimilated Orbiting Carbon Observatory-2 (OCO-2) retrievals of column-averaged $CO_2$ (XCO2). The main result of the study was the larger biospheric uptake of $CO_2$ in Australia during 2015 compared to years prior. The study suggests that the main biomes causing this larger update were the northern savannas, Mediterranean regions, and sparsely vegetative areas. Additional information is evaluated to suggest processes which caused these anomalous fluxes. The article is relatively well-written, results are novel and presented effectively, and overall conclusions are interesting. This study is also commendable in the fact it addresses an increasingly important frontier for using OCO-2 retrievals to derive sub-regional biospheric $CO_2$ fluxes. The study design and results are appropriate for the journal; however, in the current form, I can not recommend this paper for publication in Atmospheric Chemistry and Physics (ACP). As described in the comments below, many key features in the observations and modeling framework, which could have significant impact on the model results, are not described/evaluated in sufficient detail. It is a concern of the reviewer that these oversights could have influenced the model results which is heavily relied on in this study. I do however feel that if the authors can sufficiently address the major comments presented here that it could potentially be published in ACP in the future.

**Minor Comments**

1. Line 7. "the Mediterranean ecotype" instead of "Mediterranean".

2. Line 12. "concentrations".

3. Line 29. "(CO₂)".

4. Line 48. "to the period".

5. Line 110-111. "Haverd et al. (2020) ran…".

6. Line 267. I think the authors want to refer to Fig. A1.

7. Figure 6. Please use the same y-axis values for all figure panels to avoid any unnecessary confusion.

**Major Comments**

1. To help provide some background/estimate about the uncertainty of global inverse model estimates of biospheric $CO_2$ fluxes in Australia when assimilating satellite and in situ data, in addition to the text provided already in the introduction section of this study, the authors could

access gridded results of the version 9 OCO-2 Multi-model Intercomparison Project (MIP) (https://www.esrl.noaa.gov/gmd/ccgg/OCO2_v9mip/index.php). This data set provides prior and posterior estimates of Net Biome Exchange (NBE) from up to 10 global models for the four-year interval of 2015-2018. This data set could have also been used to further compare to, and evaluate, some of the results of this study. This study does compare the results to 5 global inverse models. However, the OCO-2 MIP is a controlled experiment which can help with interpreting results due to specific processes (e.g., transport model, spatial resolution, a prior fluxes, observation modes, etc.).

One thing that should be taken into consideration, which has been demonstrated with OCO-2 MIP results, is that notable differences in terrestrial carbon flux estimates are derived depending on which transport model (e.g., GEOS-Chem or TM5) is used for the inversion. Using a single transport model (i.e., WRF) could result in biased biospheric $CO_2$ fluxes simply due to a specific model's transport errors. Using a model ensemble, such as that derived by the v9 OCO-2 MIP, can help better understand these potential biases. This is just a suggestion to the authors to provide a data set to help interpret results of this study and should not be considered a requirement for application here.

2. The authors state they use "fixed patterns" for initial and boundary conditions and then solve for scaling factors. Can more detail be provided about this? Boundary conditions can be very important for the accuracy of regional-scale inversion estimates for long-lived species. First, how large is the domain used in this study? This information would be good to present to the reader prior to discussing the boundary conditions. Are the boundary conditions provided as daily, monthly, seasonal, or annual averages? Are the scaling factors derived hourly or daily to reflect variability in the boundary conditions of $CO_2$? It is difficult to understand what exactly the authors did for this. Also, what is meant by the upper and lower areas of each quadrant? Please provide actual altitude or pressure levels which separate these areas.

It would be very helpful if the authors could provide some information about the sensitivity of the results of this study to the boundary conditions used in the modeling framework.

3. It sounds as if the prior biospheric $CO_2$ model fluxes did not cover the entire domain investigated in this study. Once again, it would be helpful for the reader to know the domain dimensions prior to this discussion. For the areas not covered by the BIOS3 product the authors incorporated monthly biosphere fluxes from Australia CABLE-POP global simulations. What spatial resolution is the global model provided at? How different are the global CABLE-POP results to the fluxes derived from the BIOS3 product?

4. Why did the authors decide to exclude small fires in their prior biomass burning emission inventory? The authors should provide some reasoning for this. Do small fires not contribute much to the overall biomass burning emission total for Australia? How were the GFED emissions scaled to an hourly resolution? Also, are biofuel emissions considered in the prior $CO_2$ flux estimate?

5. The description of how prior flux error/uncertainty estimates for all flux sources is missing. Also, what flux sources are constrained in the CMAQ simulations? Are all $CO_2$ sources and sinks allowed to be adjusted when assimilating OCO-2 data? What spatial and temporal resolution are these flux constraints calculated at? Does the domain include oceanic regions? Are these ocean fluxes allow to be adjusted? Given that Australia is both upwind and downwind of oceanic regions, these emission fluxes will have a large impact on XCO2 values over the Australian continent. This needs to be described in more detail.

In the results section the authors do state that they "do not allow much freedom for ocean fluxes" implying that ocean flux prior uncertainties were set to be small values. This could greatly impact posterior land flux estimates due to the fact Australia's XCO2 values will be significantly impacted by ocean fluxes.

6. Details about how the observational error/uncertainty matrix was calculated (e.g., transport error, model-data mismatch, etc.) needs to be described as well. Are individual soundings of OCO-2 retrievals used for comparison to the model, or is there some temporal/spatial averaging? The authors point to a past study for these methods, but the way OCO-2 data is treated in this study is important enough to be presented in this manuscript.

7. How do the authors extrapolate the vertical $CO_2$ profile in CMAQ above 50 mb? Will some of the offset in the model-data XCO2 comparison be due to the fact the model does not account of this part of the vertical profile? Satellites (e.g., OCO-2, GOSAT) have sensitivity to this stratospheric $CO_2$ and will contribute to the overall retrieved XCO2 values.

8. A major result presented by the authors is that "either prior uncertainties or observational uncertainties were too high" in the model setup used in this study. Can the results of the posterior fluxes be trusted due to this? Just because the posterior $CO_2$ concentrations compare better to the observations, compared to the prior, does in no way mean that the posterior fluxes are more accurate. The authors need to expand upon this and provide evidence of why the posterior fluxes are realistic.

9. When the authors compare monthly and annual terrestrial $CO_2$ fluxes, do these include biomass burning emissions? Were fires anomalous in 2015 compared to prior years? Were the fires adjusted significantly due to the assimilation of OCO-2 observations? This is an important point because if prior fire emissions are not treated correctly in the inversion, and the fires were significant different during this year, the inversion could bias the land sink high or low. Something similar could be said for oceanic fluxes. The authors should present values, and spatial maps, for the source attribution of prior and posterior $CO_2$ fluxes for the domain. Since the description about what fluxes were constrained in the inversion, and how the individual source prior errors were attributed, it is difficult to understand and trust the results of the study.

10. Are the 0-50 mm rainfall anomalies for southeast Australia in July 2015 significant? What is the fractional increase in rainfall for this region this equates to? There are regions of Australia in the same year that received up to 700 mm more rainfall in a respective month. Also, depending

on when the rainfall was occurring in July, it might not even have much effect on the EVI values and could be impacted by months prior. It appears June 2015 had a slightly higher anomaly in rainfall in the same region. The authors are quick to attribute the increase in biospheric carbon uptake to increased EVI and rainfall, which may be true, but more analysis/explanation would help.

10. What data is used to derive the six bioclimatic classes used in this study? The authors point to a past study, but the information is needed here.

11. One of the most striking and surprising results is the large increase in carbon uptake in the sparsely vegetated (mainly desert) region of Australia. The authors state this "might be associated with an underestimation of the GPP by CABLE-BIOS3". This is true of course, but why would a sparsely vegetative region, which had decreased vegetation (negative EVI anomalies in Fig. S1) and experienced a negative anomaly in rainfall for much of the year (see Fig. S2), have such large values of carbon uptake? The March – September 2015 posterior carbon uptake values in the sparsely vegetative regions in Figure 6 are larger than any other biome in Australia. Is this not counter-intuitive and highly unexpected? Are the larger values due to very large carbon uptake, or is simply due to the large spatial extent of the biome? Could this be due to the choices in prior flux or observational uncertainties which are known to be incorrect (as stated by the authors earlier in the manuscript)? The comparison between CABLE-BIOS3 GPP and MODIS GPP is helpful but does not explain why the carbon uptake in this sparsely vegetative region is so large. This is a very interesting result, but it needs to be explained and interpreted more thoroughly.

12. There are a couple experimental setups that could just as likely lead to these results instead of it actually occurring in nature. The first thing that needs to be expanded on, and potentially investigated more, is the impact of boundary conditions on the inversion. Small errors in the boundary conditions can have large impacts on regional-scale inversion models. This is evident in Figure 7, as very large adjustments in posterior land fluxes had only small impacts on the XCO2 values (typically ~0.25 ppm) at the TCCON locations. These same results could be simulated if you had ~0.25 ppm or more errors in boundary conditions and had too small prior error uncertainties or did not adjust boundary conditions correctly. Can the authors provide evidence this is not the case? Knowing that observational and prior error uncertainties were not set correctly, how can the mean posterior fluxes values, and spatial distributions, be expected to be accurate enough to make the claims in the results of this study? Also, the authors clearly state that prior uncertainties are too stiff for ocean fluxes, how do we know that inaccuracies in the ocean prior aren't being redistributed to posterior land fluxes? All three of these concerns could easily lead to similar results/conclusions presented in this study.

13. Could this study have assimilated in situ data to infer $CO_2$ fluxes in Australia? It would be interesting, and perhaps provide more confidence in the results presented here, to see if in situ data assimilation results in similar conclusions compared to inverse model estimates using OCO-2 XCO2 data. Are there any other sources of in situ measurement data in Australia besides the data applied in this study for model evaluation?

14. Have the authors conducted the inversion using OCO-2 XCO2 data for other years than 2015? It would be interesting to see if the model results had any inter-annual variation. Was 2015 selected simply to see if the El Nino had impact on Australia carbon fluxes? If 2015 was in fact an anomalous year, it would be interesting to see if the model framework would not simulate the larger biospheric uptake for later years (e.g., 2017). This could help increase the robustness of the conclusions of this study.

---

## Author Comment (AC1)

The manuscript by Villalobos et al. (2021) evaluates terrestrial biosphere carbon dioxide  $(CO_2)$  fluxes in Australia for the year 2015. This analysis was conducted with a regionalscale inverse modeling framework which assimilated Orbiting Carbon Observatory-2 (OCO-2) retrievals of column-averaged  $CO_2$  (XCO2). The main result of the study was the larger biospheric uptake of  $CO_2$  in Australia during 2015 compared to years prior. The study suggests that the main biomes causing this larger update were the northern savannas, Mediterranean regions, and sparsely vegetative areas. Additional information is evaluated to suggest processes which caused these anomalous fluxes. The article is relatively well-written, results are novel and presented effectively, and overall conclusions are interesting. This study is also commendable in the fact it addresses an increasingly important frontier for using OCO-2 retrievals to derive sub-regional biospheric  $CO_2$  fluxes. The study design and results are appropriate for the journal; however, in the current form, I can not recommend this paper for publication in Atmospheric Chemistry and Physics (ACP). As described in the comments below, many key features in the observations and modeling framework, which could have significant impact on the model results, are not described/evaluated in sufficient detail. It is a concern of the reviewer that these oversights could have influenced the model results which is heavily relied on in this study. I do however feel that if the authors can sufficiently address the major comments presented here that it could potentially be published in ACP in the future.

**Minor Comments**

- 1. Line 7. "the Mediterranean ecotype" instead of "Mediterranean".
- 2. Line 12. "concentrations".
- 3. Line 29. "(CO2)".
- 4. Line 48. "to the period".
- 5. Line 110-111. "Haverd et al. (2020) ran...".
- 6. Line 267. I think the authors want to refer to Fig. A1.
- 7. Figure 6. Please use the same y-axis values for all figure panels to avoid any unnecessary confusion.

All the minor comments have been corrected in the manuscript.

**Major comments**

1. To help provide some background/estimate about the uncertainty of global inverse model estimates of biospheric  $CO_2$  fluxes in Australia when assimilating satellite and in situ data, in addition to the text provided already in the introduction section of this study, the authors could access gridded results of the version 9 OCO-2 Multi-model Intercomparison Project (MIP) (https://www.esrl.noaa.gov/gmd/ ccgg/0C02\_v9mip/index.php). This data set provides prior and posterior estimates of Net Biome Exchange (NBE) from up to 10 global models for the four-year interval of 2015-2018. This data set could have also been used to further compare to, and evaluate, some of the results of this study. This study does compare the results to 5 global inverse models. However, the OCO-2 MIP is a controlled experiment which can help with interpreting results due to specific processes (e.g., transport model, spatial resolution, a prior fluxes, observation modes, etc.). One thing that should be taken into consideration, which has been demonstrated with OCO-2 MIP results, is that notable differences in terrestrial carbon flux estimates are derived depending on which transport model (e.g., GEOS-Chem or TM5) is used for the inversion. Using a single transport model (i.e., WRF) could result in biased biospheric  $CO_2$  fluxes simply due to a specific model's transport errors. Using a model ensemble, such as that derived by the v9 OCO-2 MIP, can help better understand these potential biases. This is just a suggestion to the authors to provide a data set to help interpret results of this study and should not be considered a requirement for application here.

We thank the reviewer for their suggestion. We compared our results with OCO2 Multi-model Intercomparison Project (MIP). This dataset has been valuable for us to evaluate our posterior fluxes further and provide more confidence in our findings.

The analysis of the ensemble annual mean of MIP-OCO2 global inversion suggests that the carbon budget for Australia was  $-0.17 \pm 0.53$  PgC y-1 compared to our posterior estimate  $(-0.30 \pm 0.09 \text{ PgC y}^{-1})$  (Fig. 1, in this revision document). We can see in Fig. 1 that the seasonal pattern between MIP ensemble mean flux and our posterior carbon fluxes are quite similar, and both estimates fall within 1-sigma uncertainty. July is one exception to this agreement with a posterior flux of  $-1.71 \pm 0.39$  PgC y-1 compared to the MIP ensemble mean (-0.33  $\pm$  0.53 PgC y-1). Analysing the spatial distribution of these two carbon flux estimates, we found that the distribution of the MIP ensemble mean in July is similar to our posterior fluxes (both carbon sinks are located in southeastern Australia). However, the magnitude of our posterior sink is stronger compared to the MIP ensemble mean (see Fig 2(g) and Fig. 3(g)). To further analyse our results, we also assess how well our posterior monthly spatial maps agree with these ten global inversions individually. We plotted monthly maps for each global inversion (see appendix of this document). Looking at the monthly spatial maps of these global inversions for 2015, we found that our posterior flux distribution across Australia agree well with at least five global inversions (TM5, CAMS, PCTM, LoFi AMES). We believe that this intercomparison is valuable for Australia because it shows that our results are reliable with a better spatial resolution than global inversion. We will include these results in the discussion section of our manuscript, and I will contact all the OCO-2 MIP modelling group before submitting the final version of the paper to the Journal.

Figure 1: Dot grey point in (a) shows the ensemble annual mean of ten MIP-OCO2 global inversion (AMES, PCTM, CAMS, CMS-Flux, CSU, CT, LoFI, OU, TM5-4DVAR, UT) for 2015. The error bar in this graph shows the spread of these models. The orange and blue points show the annual mean of the prior and posterior biosphere land  $CO_2$  flux associated with its uncertainties (Note that the posterior uncertainties in this study were calculated by OSSEs). The grey dots in (b) show the ensemble monthly mean of the ten global inversions described in (a), and the error bars represent the monthly mean spread of this models. The orange and blue points in (b) represent the prior and posterior flux estimated (All fluxes in this comparison are without fossil fuel emissions.

---

## Author Response (AR1)

**Response to Referees' Comments**

Yohanna Villalobos Cortés

August 13, 2021

This document presents a point-by-point reply to the reviewers' comments on manuscript ACP-2021-16 (entitled "Was Australia a sink or source of $CO_2$ in 2015? Data assimilation using OCO-2 satellite measurements"). This reply is written on behalf of all co-authors.

We would like to thank to Sourish Basú and the anonymous referee for their comments and efforts towards improving our manuscript. We also thank David Baker for his careful reading of the manuscript and for his constructive remarks. His comments help us to improve and better describe our results in the paper. The first part of this document describes the main changes of the manuscript; then, we address the comments from each reviewer. The reviewer's comments are given in Roman type, and our replies are shown in blue.

**1 Summary of changes**

The main changes in the manuscript can be summarised as follows:

1. In the previous version of the manuscript (Section 3.1; Evaluation of the inversion), we indicated that our optimization system did not reach the theoretical minimum of the cost function (often defined as equal to half of the number of observations assimilated by the inversion). For this reason, we decided to re-run our inversion and adjust the uncertainties assumed in our system. A description of the prior and observation uncertainties were incorporated in Sections 2.3 and 2.5 of the manuscript. The description of these uncertainties are the same as the ones incorporated in the Observing System Simulation Experiments (OSSEs) performed by (Villalobos et al., 2020). In the current study we only increase them by a factor of 1.2. Reviewers can see in Section 3.1 that after the adjustment of the uncertainties in our system (prior and observation), we reached the required value of the cost function at its minimum, showing that our system is self-consistent. After iteration 27, we obtained a cost function of 4392, which was close to half of the total number of OCO-2 assimilated observations for 2015 (N = 9563). Figure in this section was also updated by a box plot for the 1st y-axis.

2. After re-running the inversion system, all figures, numbers, and tables in the previous manuscript were updated. The reviewers should note that the new inversion run did not change much the results found in the previous manuscript. Our assimilated fluxes still suggest that Australia was a carbon sink of -0.41 Pg $y^{-1}$ compared to prior estimate $\pm$ 0.20 PgC $y^{-1}$ (excluding fossil fuel emissions).

3. A better description of how initial and boundary condition are treated by our data assimilation system is included in Section 2.2 (Definition of the control variables) as follows:

   Paragraph added to the manuscript (Line 108-125, Pages 5 and 6): In our data assimilation system, we solve for monthly-average surface fluxes at 81 km grid-cell scale resolution as the multipliers of the principal eigenvectors of the prior error covariance matrix $\mathbf{B}$, computed as $\mathbf{W^T w}^{-1/2}(\mathbf{x} - \mathbf{x}^b)$, where $\mathbf{W}$ was defined as the matrix of eigen-vectors and $\mathbf{w}$ as a diagonal matrix of corresponding eigenvalues (Villalobos et al., 2020, Section 2.2.). In order to avoid the impact of the initial conditions (ICs) and boundary conditions (BCs) on our assimilated fluxes, we also solved them within the control vector $\mathbf{x}$. We did not optimize them in the same way as the fluxes in order to not increase the control vector size, so we treat the unknowns related to the BCs and ICs as scaling factors of the emissions added to the CMAQ model. Lateral BCs were solved as eight boundary regions divided by the upper and lower boundary areas within the CMAQ domain (south, east, north and west). In Fig. 2, we provide a representation of these boundaries. In this Figure, we can see that our study domain not only covers the Australian continent (AUS) but additionally other countries such as Indonesia (IND), Papua New Guinea (PNG) and New Zealand (NZ). The extension of this domain was created as an extra precaution to minimize the influence of the boundaries over Australia.

   Lower boundary layers were defined to cover from $\sigma = 1$ to $\sigma = 0.96$, which correspond (on average) to a pressure of 972.5 hPa, while the upper boundary layer was defined to cover from 972.5 hPa up to 50 hPa. As mentioned before, our inversion system solves for these lateral boundary components, while surface fluxes are also being optimized. Boundary conditions are provided to the CMAQ model as daily averages, but we optimize them as monthly averages. BCs and ICs dataset were taken from the CAMS global $CO_2$ atmospheric inversion product (version v19r1) (Chevallier, 2019). Uncertainties for the ICs were set at 1% (approximately 4 ppm), and the uncertainties in the lateral BCs were assumed as the standard deviation ($1\sigma$ uncertainty) of CAMS concentration data in the perimeter of the boundary.

4. We also performed (as an independent experiment) an inversion run using the combination of both land nadir and land glint (LNLG) along with ocean glint OCO-2 satellite observations (LNLGOG). We did this experiment to assess whether they had a strong impact on the Australian flux estimate compared to the inversion that only uses LNLG OCO-2 observations. We found that adding ocean OCO-2 glint observation to our system did not significantly alter the annual carbon sink flux estimated for Australia. The OCO-2 (LNLGOG) flux estimate was -0.36 PgC y$^{-1}$ compared to -0.41 PgC y$^{-1}$ flux inversion that only uses LNLG. We mentioned these findings in section 3.2 in the manuscript; however, they are not discussed later in the following sections of the manuscript. We decided that it was better not to include it in the main results because ocean glint observations still have undetermined biases, which might mislead the Australian flux estimate.

5. In Section 3.4, we also included the annual mean flux for 2015 classified by bioclimatic region across Australia (Fig.8 in the manuscript).

6. In Section 3.5. (Evaluation of the inversion with independent data) we decided to examine whether biases in our posterior concentration could get better when incorporating glint ocean observations into the inversion. From Fig.10 to Fig13 were updated with this new information. Now the comparison is not only using LNLG but also with LNLGOG.

7. We expanded Section 4 in the manuscript (Discussion Section). In this new version of the manuscript, we compared our assimilated fluxes against the ensemble mean of ten different global inversions (AMES, Baker, CAMS, CMS-Flux, CSU, CT, LoFI, OU, TM5−4DVAR, UT) that participate in the OCO-2 model intercomparison (MIP). We added this extra analysis as recommendation of referee #1 and David Baker to assess whether our assimilated carbon flux estimates fall within the ensemble estimate of the OCO-2 MIP. We do not only compare our fluxes against OCO-2 MIP but also with *in situ* MIP inversion. In summary, we found that our posterior annual mean carbon flux estimate for Australia (-0.41 ± 0.08 PgC y$^{-1}$) falls within the annual ensemble mean estimate of the MIP OCO-2 (LNLG) flux inversion (-0.17 ± 0.26 PgC y$^{-1}$) (Fig. 14a in the manuscript) and that seasonally of our inversion produces a similar seasonal pattern to the ensemble monthly mean of MIP-OCO2 (LNLG) (except for July) (Fig.14b). In this section we also included a case were the monthly mean for July was interpolated between June and August. By doing this interpolation, our monthly mean estimate for July was -0.62 ± 0.34 PgC y$^{-1}$ similar to MIP OCO-2 (-0.33 ± 0.97 PgC y$^{-1}$), considering the spread of the uncertainties (Fig.15b). In this section we mentioned that July (winter season in Australia) was difficult to validate against *in situ* measurements (such as Ironbark and Burncluith), and mentioned that CMAQ model might present some limitation to match surface fluxes with column-average OCO-2 in winter. We included a brief description of the MIP OCO-2 and *in situ* dataset in Section 2.6.4 in the manuscript.

8. We also included within the supplementary document spatial distribution maps of the ensemble monthly mean of OCO-2 MIP global inversion (Supplementary, Fig.S13) as well as the spatial distributions maps of the ten independent OCO-2 global inversions (Supplementary, Figs. S14 to S23) so the reader can compare the spatial distribution map of our inversion (Fig.7 in the main manuscript).

9. We also updated the Appendix Section of the manuscript.

   - Appendix B, Fig. B1 shows the monthly time series of the Australian prior and posterior carbon classified by different six bioclimatic regions and divided by its area.
   - Appendix C, Fig. C2 shows the spatial distribution of OCO-2 soundings (LNLGOG) over the CMAQ domain for 2015.
   - Appendix F, Table F1 shows the Australian monthly terrestrial carbon flux estimated using LNLG and LNLGOG for 2015.
   - Appendix G, Table G1 to G4 shows the analysis of the residual between CMAQ prior and posterior simulation (LNLG and LNLGOG) and TCCON estimates (Darwin, Wollongong, and Lauder site) for 2015.

- Appendix H, Table H1 to H4 shows the analysis of the residual between CMAQ prior and posterior simulation (LNLG and LNLGOG) and *in situ* monitoring sites (Gunn Point, IronBark, Burncluith and Cape Grim) for 2015.

- Appendix I, Fig. 11 show the time series of monthly mean carbon fluxes derived by MIP *in situ* (IS) global inversion for 2015.

- Appendix I, Fig. 12 show the time series of monthly mean carbon fluxes derived by MIP OCO-2 (LNLG) global inversion for 2015.

**Referee I**
The manuscript by Villalobos et al. (2021) evaluates terrestrial biosphere carbon dioxide ($CO_2$) fluxes in Australia for the year 2015. This analysis was conducted with a regional-scale inverse modeling framework which assimilated Orbiting Carbon Observatory-2 (OCO-2) retrievals of column-averaged $CO_2$ (XCO2). The main result of the study was the larger biospheric uptake of $CO_2$ in Australia during 2015 compared to years prior. The study suggests that the main biomes causing this larger update were the northern savannas, Mediterranean regions, and sparsely vegetative areas. Additional information is evaluated to suggest processes which caused these anomalous fluxes. The article is relatively well-written, results are novel and presented effectively, and overall conclusions are interesting. This study is also commendable in the fact it addresses an increasingly important frontier for using OCO-2 retrievals to derive sub-regional biospheric $CO_2$ fluxes. The study design and results are appropriate for the journal; however, in the current form, I can not recommend this paper for publication in Atmospheric Chemistry and Physics (ACP). As described in the comments below, many key features in the observations and modeling framework, which could have significant impact on the model results, are not described/evaluated in sufficient detail. It is a concern of the reviewer that these oversights could have influenced the model results which is heavily relied on in this study. I do however feel that if the authors can sufficiently address the major comments presented here that it could potentially be published in ACP in the future.

**Minor Comments**

1. Line 7. "the Mediterranean ecotype" instead of "Mediterranean".

2. Line 12. "concentrations".

3. Line 29. "(CO2)".

4. Line 48. "to the period".

5. Line 110-111. "Haverd et al. (2020) ran...".

6. Line 267. I think the authors want to refer to Fig. A1.

7. Figure 6. Please use the same y-axis values for all figure panels to avoid any unnecessary confusion.

All the minor comments have been corrected in the manuscript.

**Major comments**

1. To help provide some background/estimate about the uncertainty of global inverse model estimates of biospheric $CO_2$ fluxes in Australia when assimilating satellite and *in situ* data, in addition to the text provided already in the introduction section of this study, the authors could access gridded results of the version 9 OCO-2 Multi-model Intercomparison Project (MIP) (`https://www.esrl.noaa.gov/gmd/ccgg/OCO2_v9mip/index.php`). This data set provides prior and posterior estimates of Net Biome Exchange (NBE) from up to 10 global models for the four-year interval of 2015-2018. This data set could have also been used to further compare to, and evaluate, some of the results of this study. This study does compare the results to 5 global inverse models. However, the OCO-2 MIP is a controlled experiment which can help with interpreting results due to specific processes (e.g., transport model, spatial resolution, a prior fluxes, observation modes, etc.). One thing that should be taken into consideration, which has been demonstrated with OCO-2 MIP results, is that notable differences in terrestrial carbon flux estimates are derived depending on which transport model (e.g., GEOS-Chem or TM5) is used for the inversion. Using a single transport model (i.e., WRF) could result in biased biospheric $CO_2$ fluxes simply due to a specific model's transport errors. Using a model ensemble, such as that derived by the v9 OCO-2 MIP, can help better understand these potential biases. This is just a suggestion to the authors to provide a data set to help interpret results of this study and should not be considered a requirement for application here.

We thank the reviewer for his/her suggestion. As mentioned earlier in this document, we compared our results with OCO2 Multi-model Intercomparison Project (MIP). This dataset has been valuable for us to evaluate our posterior fluxes further and provide more confidence in our findings. Details of OCO-2 and *in situ* MIP results have been included in Section 4 of the manuscript (Please see pages 32 to 36).

2. The authors state they use "fixed patterns" for initial and boundary conditions and then solve for scaling factors. Can more detail be provided about this? Boundary conditions can be very important for the accuracy of regional-scale inversion estimates for long-lived species. First, how large is the domain used in this study? This information would be good to present to the reader prior to discussing the boundary conditions. Are the boundary conditions provided as daily, monthly, seasonal, or annual averages? Are the scaling factors derived hourly or daily to reflect variability in the boundary conditions of $CO_2$? It is difficult to understand what exactly the authors did for this. Also, what is meant by the upper and lower areas of each quadrant? Please provide actual altitude or pressure levels which separate these areas. It would be very helpful if the authors could provide some information about the sensitivity of the results of this study to the boundary conditions used in the modeling framework.

We agree with the reviewer that regional inversions are sensitive to boundary conditions. To assess the impact of biases in the lateral boundaries of the CMAQ domain, we performed two sensitivity experiments. In both experiments we add a constant offset of 0.25 ppm to each grid-cell of the BCs. In the first experiment we solve for the BCs and use LNLG. This induces a bias in our posterior annual flux of -0.8 PgC. Solving for the BCs and using LNLGOG observations reduces the bias further to (-0.4 PgC $y^{-1}$). Adding 0.25 ppm everywhere is an extreme test since the global assimilation fields we use are unlikely to have such systematic errors against data they assimilate. The results do highlight the importance of solving for the BCs in a regional inverse system and also the importance of large domains with enough observations in a buffer region around our area of interest. These results are also reinforced by the good agreement that our assimilated fluxes have with MIP OCO-2. This text was added to Section 4 of the manuscript (Please see pages 32 to 36).

3. It sounds as if the prior biospheric $CO_2$ model fluxes did not cover the entire domain investigated in this study. Once again, it would be helpful for the reader to know the domain dimensions prior to this discussion. For the areas not covered by the BIOS3 product the authors incorporated monthly biosphere fluxes from Australia CABLE-POP global simulations. What spatial resolution is the global model provided at? How different are the global CABLE-POP results to the fluxes derived from the BIOS3 product?

In section 2.2 (Defining the control vector) we included a description of our study domain (Fig.2 in the manuscript). In this section we mentioned that our study domain covers the Australia region and other countries such as New Zealand (NZ), Indonesia (IND) and Papúa New Guinea (PNG). As mentioned in section 2.3 of the manuscript, CABLE model runs in the BIOS-3 setup and rely on their regional driver and observations, which do not exist outside the domain of the Australian continent. These data (BIOS-3) provide the advantage of the best possible prior for Australia. Land areas that are not covered by CABLE BIOS-3 simulations were derived using CABLE-POP global runs (runs included in TRENDY intercomparison version 8). The spatial resolution of this global product were given at 1 degree compared to CABLE BIOS-3, which was given at 0.25 degree.

4. Why did the authors decide to exclude small fires in their prior biomass burning emission inventory? The authors should provide some reasoning for this. Do small fires not contribute much to the overall biomass burning emission total for Australia? How were the GFED emissions scaled to an hourly resolution? Also, are biofuel emissions considered in the prior $CO_2$ flux estimate?.

We did not exclude fires emission from GFED product, because we used the fourth version of the Global Fire Emissions Database (GFED 4.1s). We forgot to update this information in the manuscript. We will correct the text which says that GFED emissions were scaled to an hourly resolution because we assumed GFED was constant across the months, and we did not include biofuel emissions in the prior flux estimate.

5. The description of how prior flux error/uncertainty estimates for all flux sources is missing. Also, what flux sources are constrained in the CMAQ simulations? Are all $CO_2$ sources and

sinks allowed to be adjusted when assimilating OCO-2 data? What spatial and temporal resolution are these flux constraints calculated at? Does the domain include oceanic regions? Are these ocean fluxes allow to be adjusted? Given that Australia is both upwind and downwind of oceanic regions, these emission fluxes will have a large impact on XCO2 values over the Australian continent. This needs to be described in more detail. In the results section the authors do state that they "do not allow much freedom for ocean fluxes" implying that ocean flux prior uncertainties were set to be small values. This could greatly impact posterior land flux estimates due to the fact Australia's XCO2 values will be significantly impacted by ocean fluxes.

We agree with the author that a better description of the prior and uncertainties should have been provided in the document. We did not include much detail of them, because a complete description of prior characterization was made in Villalobos et al. (2020). The text in section 2.3 in this manuscript was updated with more details about how we constructed the prior and their uncertainties.

Regarding the reviewer's questions, we constrain anthropogenic fluxes, fires, land and ocean fluxes in the CMAQ simulations, and all of them were adjusted by the optimizer when assimilating OCO-2 data. We solve for monthly averaged fluxes at the CMAQ horizontal grid-resolution (81 km). Our study domain as seen in Fig.2 of the manuscript includes ocean regions. Emissions over ocean were taken from the latest version of CAMS greenhouse inversion (Chevallier, 2019). We did adjust ocean fluxes, but not much (this is because our prior error covariance matrix was decomposed along its eigenvectors and eigenvalues, where most of the variance was captured over land). We did not give much flexibility for the inversion to move ocean fluxes because the ocean uncertainties were assumed to be small in comparison with emission uncertainties over land. For details please see details see section 2.3.

6. Details about how the observational error/uncertainty matrix was calculated (e.g., transport error, model-data mismatch, etc.) needs to be described as well. Are individual soundings of OCO-2 retrievals used for comparison to the model, or is there some temporal/spatial averaging? The authors point to a past study for these methods, but the way OCO-2 data is treated in this study is important enough to be presented in this manuscript.

For details of how we created the observational uncertainties please see Section 2.5 (OCO-2 satellite information and its uncertainties). With regards to the reviewer's questions, we averaged all the OCO-2 soundings for comparison to the CMAQ model followed the advise of David F. Baker (personal communication). To construct this average, we used a two-step process, first we grouped OCO-2 sounding across 1-second spans, then we grouped those across 11-second spans. This averaging procedure is similar to the one used by Crowell et al. (2019).

7. How do the authors extrapolate the vertical $CO_2$ profile in CMAQ above 50 mb? Will some of the offset in the model-data XCO2 comparison be due to the fact the model does not account of this part of the vertical profile? Satellites (e.g., OCO-2, GOSAT) have sensitivity to this stratospheric $CO_2$ and will contribute to the overall retrieved XCO2 values.

We did not extrapolate the CMAQ vertical $CO_2$ profile above 50 hPa. We assigned everything

above the model-top (50 hPa) to the top layer of the model, which only represents 5% of the mass of the atmosphere. In order to assess how much impact this assumption would have in the model-data XCO2 comparison, we made a simple extra comparison where we assumed that the CMAQ XCO2 column-average concentration derived from our interpolation only accounts for 95% mass of the atmosphere, and the rest (5%) corresponds to the upper model-top concentration in the CMAQ model. We found that the difference between "the model-data XCO2 comparison" and this simple extra comparison was minimal, and on average, for the whole CMAQ domain, this difference only represents about 0.03 ppm. These findings were expected because the vertical profile variation above 20 km shows less variability than below this height.

8. A major result presented by the authors is that "either prior uncertainties or observational uncertainties were too high" in the model setup used in this study. Can the results of the posterior fluxes be trusted due to this? Just because the posterior $CO_2$ concentrations compare better to the observations, compared to the prior, does in no way mean that the posterior fluxes are more accurate. The authors need to expand upon this and provide evidence of why the posterior fluxes are realistic.

We meant to say on page 1.3 in the manuscript that either the prior or observation were too low in the model set-up. We understand the concern of the reviewer related to the prior and observational uncertainties. As stated earlier in this document, we ran again the inversion adjusting the uncertainties. We increased both the prior and observations uncertainties by a factor of 1.2. For more details please section 3.1. (Inversion Evaluation: Analysis of the residual between CMAQ simulation and OCO-2).

9. When the authors compare monthly and annual terrestrial $CO_2$ fluxes, do these include biomass burning emissions? Were fires anomalous in 2015 compared to prior years? Were the fires adjusted significantly due to the assimilation of OCO-2 observations? This is an important point because if prior fire emissions are not treated correctly in the inversion, and the fires were significant different during this year, the inversion could bias the land sink high or low. Something similar could be said for oceanic fluxes. The authors should present values, and spatial maps, for the source attribution of prior and posterior $CO_2$ fluxes for the domain. Since the description about what fluxes were constrained in the inversion, and how the individual source prior errors were attributed, it is difficult to understand and trust the results of the study.

We cannot determine if the fires emissions were adjusted significantly due to the assimilation of OCO-2 observations because we did not solve the fire flux separately. Instead, we solve for the net flux, which includes fires, fossil fuel and land emissions. To answer the review question relating to anomalies fires in 2015, we calculate the GFED fire anomalies for 2015 relative to the mean 2000–2014. We included two new Figures in the Supplementary information (Fig.S4 and Fig.S5). Fig.S4 shows fires emission across Australia for 2015, and Fig. S4 shows their anomalies.In these Figures the reviewer can see high fire activity in the Northern Australia

from April to September, and November (dry season in Australia).

10. Are the 0-50 mm rainfall anomalies for southeast Australia in July 2015 significant? What is the fractional increase in rainfall for this region this equates to? There are regions of Australia in the same year that received up to 700 mm more rainfall in a respective month. Also, depending on when the rainfall was occurring in July, it might not even have much effect on the EVI values and could be impacted by months prior. It appears June 2015 had a slightly higher anomaly in rainfall in the same region. The authors are quick to attribute the increase in biospheric carbon uptake to increased EVI and rainfall, which may be true, but more analysis/explanation would help.

We plotted the fractional increase of rainfall per grid-cell for each month of the year 2015 (relative to mean 2000–2014) (see Supplementary, Fig. S3). We found that about 50 mm in southeast Australia in July 2015 represents about 60% of a increase of rainfall (relative to mean 2000-2014). To a certain extent we agree with the reviewer that the rainfall anomalies across southern-east Australia in May and June (previous to July) also contributing to increase in vegetation in this region. However during wet times in this region, pasture may grow quickly and become dense.

11. What data is used to derive the six bioclimatic classes used in this study? The authors point to a past study, but the information is needed here.

The six bioclimatic classes used in this study correspond to an aggregation of the 18 agro-climatic zones generated by Hutchinson et al. (2005). The climatic classification in Hutchinson et al. (2005) was adapted from an existing global agro-climate classification (Hutchinson, 1992), which was refined and closely aligned with natural vegetation formations and common land uses across Australia using 182 weather climate stations and the Interim Biogeographic Regionalisation for Australia (IBRA). We included this information in Section 3.4 (Australian $CO_2$ flux estimate classified by bioclimatic zones).

12. One of the most striking and surprising results is the large increase in carbon uptake in the sparsely vegetated (mainly desert) region of Australia. The authors state this "might be associated with an underestimation of the GPP by CABLE-BIOS3". This is true of course, but why would a sparsely vegetative region, which had decreased vegetation (negative EVI anomalies in Fig. S1) and experienced a negative anomaly in rainfall for much of the year (see Fig. S2), have such large values of carbon uptake? The March – September 2015 posterior carbon uptake values in the sparsely vegetative regions in Figure 6 are larger than any other biome in Australia. Is this not counter-intuitive and highly unexpected? Are the larger values due to very large carbon uptake, or is simply due to the large spatial extent of the biome? Could this be due to the choices in prior flux or observational uncertainties which are known to be incorrect (as stated by the authors earlier in the manuscript)? The comparison between CABLE-BIOS3 GPP and MODIS GPP is helpful but does not explain why the carbon uptake in this sparsely vegetative region is so large. This is a very interesting result, but it needs to be explained and interpreted more thoroughly.

In Appendix B (Fig.B1), we plotted the prior and the posterior flux estimates not with the same units ($PgC\,y^{-1}$) as same the one shown in Fig.9 (see manuscript). In this case, the carbon fluxes were divided by the area of each bioclimatic ecosystem. It is evident in Fig.B1 that the significant increase in carbon uptake in the sparsely vegetated region of Australia is because a "small shift" in the carbon fluxes over this large ecosystem cause an important impact on the total carbon net flux calculated for the whole country. We include this information in our results. For more details see section 3.4. (Page 22).

13. There are a couple experimental setups that could just as likely lead to these results instead of it actually occurring in nature. The first thing that needs to be expanded on, and potentially investigated more, is the impact of boundary conditions on the inversion. Small errors in the boundary conditions can have large impacts on regional-scale inversion models. This is evident in Figure 7, as very large adjustments in posterior land fluxes had only small impacts on the XCO2 values (typically $\approx$0.25 ppm) at the TCCON locations. These same results could be simulated if you had  0.25 ppm or more errors in boundary conditions and had too small prior error uncertainties or did not adjust boundary conditions correctly. Can the authors provide evidence this is not the case? Knowing that observational and prior error uncertainties were not set correctly, how can the mean posterior fluxes values, and spatial distributions, be expected to be accurate enough to make the claims in the results of this study? Also, the authors clearly state that prior uncertainties are too stiff for ocean fluxes, how do we know that inaccuracies in the ocean prior aren't being redistributed to posterior land fluxes? All three of these concerns could easily lead to similar results/conclusions presented in this study.

    This question related to boundary conditions (BCs) was answered in the first section of this document. For details of how we handle BCs in the manuscript see Section 2.2 (Defining the control vector), and Section 4 (discussion).

14. Could this study have assimilated *in situ* data to infer $CO_2$ fluxes in Australia? It would be interesting, and perhaps provide more confidence in the results presented here, to see ifv*in situ* data assimilation results in similar conclusions compared to inverse model estimates using OCO-2 XCO2 data. Are there any other sources of *in situ* measurement data in Australia besides the data applied in this study for model evaluation?

    We agree with the reviewer that performing a regional inversion with in-situ measurement would have been ideal for quantifying the Australian carbon fluxes with more confidence. But unfortunately, there are not many other sources of *in situ* measurements in Australia besides the ones applied in this study. For this reason, we only perform a regional inversion using OCO-2 data. If the reviewer wants to know more about Australia monitoring $CO_2$ site, Ziehn et al. (2014) performed a study over Australia which proposed expanding the current monitoring $CO_2$ stations over the country. In this work, the authors mention that the current monitoring sites (around six across Australia, which are not all operational) provide no meaningful constraint on Australian fluxes.

15. Have the authors conducted the inversion using OCO-2 XCO2 data for other years than 2015? It would be interesting to see if the model results had any inter-annual variation. Was 2015

selected simply to see if the El Nino had impact on Australia carbon fluxes? If 2015 was in fact an anomalous year, it would be interesting to see if the model framework would not simulate the larger biospheric uptake for later years (e.g., 2017). This could help increase the robustness of the conclusions of this study.

We ran a regional inversion for Australia for 2015–2019 using OCO-2 data (version 9) to assess the inter-annual variability of the Australian $CO_2$ surface fluxes. These five years inversion is a working paper that focuses on understanding the main climate drivers that may cause inter-annual variability and/or temporal trends in carbon fluxes over Australia. One of the questions in this new article is to evaluate whether the Australian semi-arid ecosystems followed the same patterns observed during 2015 or whether such patterns became stronger or weaker due to precipitation and temperature changes throughout 2015–2019. Australia is a very interesting study case because despite being affected by ENSO in 2015, not all months in 2015 had a deficit of rain, which certainly impacted the net carbon fluxes across Australia (see Supplementary, Fig.S2).

**Referee 2 - Sourish Basú**

The manuscript is a clear and straightforward presentation of the Australian NBP derived from OCO2 XCO2 retrievals in 2015. The inversion methodology and the analysis of results is sound, and I would recommend publication once the authors have addressed the following questions.

1. Line 48: Is the anomaly of 0.40-0.61 PgC/yr positive or negative? It is presented as a positive anomaly, but that goes against an expected negative anomaly in 2011.

   We have restructured the paragraph that starts in line 29 on page 2, and clarify that the anomaly flux found by Trudinger et al. (2016) over Australia was negative.

   Modified text (Line 53-54): Trudinger et al. (2016) also found a similar carbon sink anomaly for this period (ranging between -0.40 to -0.61 PgC $y^{-1}$)

2. Line 57: What is the "BoM, 2015" reference?

   The Bureau of Meteorology (BoM) is Australia's national weather, climate and water agency, which provide to the public report of the weather forecasts, warnings and observations. We have modified the reference "BoM, 2015" to Annual Climate Statement, Bureau of Meteorology, Australia, 2015. Please see line 63 on page 3

3. Figure 1: The uncertainty in the top right yellow box should be $R^{-1}$, not $B^{-1}$

   We have modified Figure 1 on page 4. We replaced the uncertainty in the top right yellow box by $R^{-1}$.

4. Line 119: ODIAC certainly has bunker fuels, including aviation and marine transport. For at least the past five years, ODIAC files have had a variable called "int bunker" containing bunker emission estimates. Since the authors assumed that ODIAC did not have bunker fuels, are they double counting that category by adding EDGAR bunker emissions?

We did not double count bunker emissions because we used a ODIAC version file that exclude them (e.g., odiac2019_1km_excl_intl_1501.tif.gz file).

5. Line 123: Which version of EDGAR did you use? And which sectors of EDGAR did you sum up to get just the marine and aviation sectors? Can you be specific?

We used version 5.0 of the EDGAR emission inventory, and the aviation sectors selected were aviation climbing and descent, aviation cruise, and aviation landing and takeoff. These aviation emissions were also distributed along the CMAQ vertical levels. Given that we used CAMS emissions over ocean, we did not include EDGAR shipping emission, because CAMS also include them. We included this information in section 2.3 (Prior information and its uncertainties).

6. Line 128: Why were small fires excluded? And why was version 4 used instead of 4.1s? The latter has been out for at least 4 years now.

We used the fourth version of the Global Fire Emissions Database (GFED 4.1s). We forgot to update this information in the manuscript, but this information has been corrected, see section 2.3 (Prior information and its uncertainties).

7. Line 148: Can you say exactly how you chose "good" quality soundings? OCO2 quality flags are 0 and 1, not good and bad, and I am asking for specifics because earlier this year I reviewed another manuscript that used the wrong value of the quality flag to select their soundings.

We only selected OCO-2 soundings flagged equal to 0, and reject all soundings flagged to soundings with low quality (quality flag $\neq$ 0).

Modified text (Line 192-194, page 9): We only used OCO-2 retrievals with quality flag 0 and only soundings that were bias-corrected, as described by (Kiel et al., 2019).

8. Line 172: "Monthly daily mean averages" denotes a lot of averaging. Why not compare the observations with the model co-sampled at exactly the correct times?

We did co-sample the model and the observation at exactly time (every hour) and location, then we only selected data between 12:00 and 5:00 p.m. (local time) to perform the average. To clarify this in the main manuscript, we have restructured the paragraph.

Modified text (Line 235, page 10): We used hourly data from these monitoring sites, but the monthly mean averaged data shown in Section 3.5.2 were calculated using local time averages between midday and 05:00 p.m.

9. It is customary for satellite inversions to do an *in situ* only inversion as a "baseline", to compare OCO2 inversions with and assess the added information content and possible biases in the OCO-2 data. Why was no *in situ* inversion performed in this study?

We did not perform a Australia regional inversion with in-situ measurements because there are not many other sources besides the ones applied in the validation Section in this study (Cape Grim, Ironbark, Gunn Point and Burncluith). We believe the four stations across Australia do not provide a meaningful constraint on Australian fluxes to assess possible bias in OCO-2 data.

10. Line 199: Why is the correlation calculated between the flux correction and MODIS EVI? It would make more sense (to me) to calculate the correlation between MODIS EVI and just the posterior flux.

Our results showed that OCO-2 produced a considerable shift in the carbon sink (relative to the prior) over the savanna and sparsely vegetated region. We wanted to see whether this shift was due to CABLE model simulation was not able to capture these anomalies. We found strong negative correlations (R> 0.8) at grid-cell scale between the EVI anomalies and the posterior and prior difference in northern Australia (savanna ecosystem) and in the western region of the sparsely vegetated ecosystem, which align with the spatial pattern of rainfall in that area. These results suggest that our OCO-2 inversion better capture the anomalies in comparison with the biosphere land model. We include these findings in the manuscript so that the reader can understand our point.

11. Line 233: I think the cost function being too high is a symptom of uncertainties that are too low, not too high.

We agree with the reviewer on this point. What we meant to say on page 1.3 in the manuscript was that either the uncertainties in the prior or observation were too low in the model set-up. We decided to re-run the inversion again and increase the both uncertainties (prior and observations) by a factor of 1.2 (Please see information early in the manuscript).

12. Lines around 245: What is the uncertainty in the Australian fossil $CO_2$ emission? Say the range between different inventories?

We included the Australian fossil fuel uncertainties for 2015 (0.01 Pg $y^{-1}$) in the manuscript. Fossil-fuel uncertainties were created by multiplying the emissions dataset with a factor of 0.44. This factor was derived by linear regression between the mean fluxes and the spread of an ensemble of 25 realizations posterior $CO_2$ fluxes, following Asefi-Najafabady et al. (2014).

13. Lines around 365: My impression from the TCCON plots is that the posterior sticks pretty close to the prior and does not necessarily approach the TCCON data in most months. Given that two of the three TCCON stations are exactly where the largest flux adjustments are (northern and south-eastern Australia), isn't that bad news for the reliability of your flux estimates? Or can you show that these TCCON mismatches are primarily due to boundary conditions at the edges of your CMAQ domain?

The largest flux adjustment was found over the savanna and sparsely vegetated ecosystem (see Fig. 8 in the manuscript). Unfortunately, we do not have TCCON monitoring stations in these categories. TCCON Darwin is located in the Tropics, and TCCON Wollongong is located in a warm temperate region as stated our conclusion (Section 5) of the manuscript. The scarcity of monitoring observations across the Australian continent, mainly over the savanna and sparsely vegetated ecosystem, restrict our ability to conclude with confidence whether the stronger carbon sink (relative to the prior) found in those ecosystems are real or not. However, the comparison with the annual and monthly ensemble means of the OCO-2 MIP intercomparison reinforces and supports our results.

14. Figure 10: Looking at JJA at Barncluith and Ironbark, I get the impression that some of the deep JJA winter sink is an artifact, because the posterior is too low compared to *in situ* $CO_2$. Correct?

We agree with the reviewer, and some discussions of these results were incorporated in Section 4. We added the following paragraph to the manuscript

Paragraph (Lines 596-601, page 32): As we saw in Section 3.5 the validation of our inversion against Ironbark and Burncluith sites suggest that the anomalous sink seen in July might likely be related to errors in the transport model. If we interpolate July between June and August, we reproduce a monthly mean (-0.62 0.34 PgC $y^{-1}$), which is closer to the ensemble mean of MIP-OCO-2 (-0.33 ± 0.52 PgC PgC $y^{-1}$), considering the range of the spread the model and the uncertainties of our posterior fluxes (see Fig. 14b). By doing this interpolation, we shift our posterior annual flux from -0.41 to -0.32 PgC $y^{-1}$, which is also closer to the annual ensemble mean of MIP-OCO-2 (-0.17 PgC $y^{-1}$).

15. Line 412: The -0.30 ± 0.09 PgC/yr is the prior, not the posterior, correct?

The number quoted by the reviewer correspond to the posterior flux estimate in the previous version the manuscript. After re-running the inversion, and adjust the uncertainties in the system, the new posterior estimate for Australia was -0.41 ± 0.08 PgC $y^{-1}$)

16. Lines around 415: Not all global inverse models are made equal. If you have the models' mole fractions, you can check how well these five models fit the *in situ* data you have over Australia, which indicates how well you can trust their fluxes in that region. I strongly suspect that the realistic "model uncertainty" or range you will get out of that exercise will be smaller.

We agree with the reviewer, and the model uncertainties (the ensemble model spread) quoted in the paper was an error. We decided to exclude these results from the paper and replaced them with the ensemble mean of the MIP *in situ* inversion to be consistent with the comparison with MIP OCO-2. We re-wrote all the findings in Section 4 of the manuscript. In summary, we found the ensemble annual mean for the MIP *in situ* was 0.21 ± 0.39 PgC $y^{-1}$) compared to our assimilated fluxes (-0.41 ± 0.39 PgC $y^{-1}$ and the MIP OCO-2 inversion (-0.17 ± 0.26 PgC $y^{-1}$). For more details, please see section 4 (Discussion).

17. Table 4: Of the models listed here, I understand that ORCHIDEE or CASA may have the wrong seasonality over Australia, since most of the ecosystem data used to evaluate those models do not come from Australia. Why would BIOS-CABLE3, a model built specifically for Australia, also have the minimum in the wrong month?

A BIOS model evaluation with OzFlux sites carried out by (Trudinger et al., 2016) indicates that the CABLE model performs well in representing the seasonal cycles of the GPP flux in regions driven by both monsoonal rainfall seasonality (Tropical savanna in northern Australia) in by radiation seasonality (cool temperate ecosystem). However, the performance of the model in dry periods does not perform very well and underpredicts the GPP flux. Part of the GPP underprediction is due to the algorithm used for partitioning LAI. In dry periods, the CABLE model assumes that most of LAI is attributable to grass cover vegetation with shallow-root

systems resulting in severe water limitation in the dry season. A similar explanation is for ecosystems with sparse vegetation in north-central Australia, which are not well simulated by the model.

18. General question about error bars: Are the error bars reported here the analytical errors from flux and measurement uncertainty, or do they also contain the uncertainty due to the boundary condition at the edges of the CMAQ domain?

    We provided a better description of how we built the prior uncertainties incorporated in the inversion system (Section 2.3; Prior information and its uncertainties) and the observation uncertainties (Section 2.5; OCO-2 satellite information and its uncertainties). The prior and posterior flux uncertainties quoted in the manuscript do not include the uncertainties assumed in boundary conditions (BCs). However, we solved them in our inversion, not in the same way we solve for the surface fluxes, as they were were treated as nuisance variable.

    Citation: `https://doi.org/10.5194/acp-2021-16-RC2`

**Response - David Baker**

This study uses a state-of-the-art regional atmospheric $CO_2$ flux inversion system to assimilate column-average $CO_2$ measurements from the OCO-2 satellite over Australasia. Monthly and annual mean flux estimates for Australia, as well as six eco-climate sub-regions inside it, are analyzed for the calendar year 2015. The results for the domain as a whole are compared to similar estimates from six global flux inversions (which, however, used only *in situ* data instead of the OCO-2 data). To help interpret the results, vegetation greenness information in the form of satellite EVI data, as well as rainfall products, were examined. Since the results are presented in terms of deviations from a carbon model prior, deviations of the GPP given by this model from GPP estimates derived from the MODIS instrument are also examined. The main finding is that the OCO-2 data indicate that the annual mean flux for Australia for 2015 was -0.3 PgC/yr, a substantial sink, as opposed to the small source of about +0.1 PgC/yr given by their carbon model prior or the substantial source of +0.6 PgC/yr given by the mean of the six global inversions. (Though see below my question about the latter number – it may be closer to +0.1 or +0.2 PgC/yr if the monthly fluxes plotted in Figure 11 are correct.) This result suggests that the OCO-2 measurements cause the land biospheric flux for Australia to change from the source given by the *in situ* data into a robust sink.

1. The paper is well-organized and well-presented. It should not, however, be published in its present form, for the following reasons: 1) It is not clear from the work presented here that the main result, this shift of about a half PgC/yr or more from source to sink in the 2015 annual mean, is real and not simply an artifact of local biases in the OCO-2 data, or of the impact of the coverage of the OCO-2 data on the regional inversion (i.e., the partitioning of the OCO-2 information between fluxes in the interior of the domain versus fluxes outside the domain, interpreted here as errors in the boundary conditions).

   To respond to the reviewer's concern, we give a better description in the manuscript of how the initial and boundaries conditions are treated in the inversion (see Section 2.2 of the

main manuscript). Here, the reviewer should see that the target area was well-buffered from boundaries to avoid edge effects.

2. Going beyond this shift in flux to look at the absolute values of both the prior and the posterior fluxes for the region, further uncertainty is introduced by potential errors in the transport model used here, the prior concentrations used (in particular, the $CO_2$ gradients produced by running the prior fluxes through the transport model combined with the global offset needed), and the impact of drift in the regional inversion (the fact that the constraint on the land + ocean $CO_2$ flux total obtained in the global inversions from the difference of the fossil fuel input and the atmospheric increase is no longer available in the regional inversion).

   The global trend is rarely fitted explicitly in global inversions and the trend as an implicit constraint is present here too. The boundaries conditions (BCs) does contribute though but it is very unlikely that the trend in BCs is wrong since it comes from an CAMS assimilation OCO-2 data.

3. One cannot rely on the errors obtained from the OSSEs done here to capture these error sources, unless the OSSEs somehow captured all the transport model, inversion method, and measurement bias errors along with the random measurement errors that are usually addressed (terms that were not mentioned in the text and that I doubt were addressed).

4. We addressed some of the systematic error mentioned above. We did not only consider the uncertainties OCO-2 observation, we followed the algorithm that the reviewer provided in a personal communication few years ago. To clarify the sources of error included in the inversion, we included the following text in the manuscript (section 2.5, OCO-2 satellite information and its uncertainties). The 1-second weighted averaging process is described in detail in Section 2.3. in Villalobos et al. (2020). In summary, to obtain the uncertainties of these 1-second averaging processes, we considered three different forms of uncertainty calculation, similar to Crowell et al. (2019). First, we averaged OCO-2 uncertainties assuming that these were correlated in a 1-second span (uncertainties defined as $\sigma_s$). Second, given that the average of OCO-2 uncertainties are sometimes low because they neglect systematic errors, we also used the spread of the OCO-2 retrievals in the 1-second average (uncertainties defined as $\sigma_r$). Third, we also defined baseline uncertainties (defined as $\sigma_b$) for cases where the number of soundings was not enough to compute a realistic spread. The values for our baseline uncertainties were assumed to be 0.8 ppm over land and 0.5 ppm over ocean. Finally, we selected the maximum value between these three uncertainties ($\sigma_s$, $\sigma_r$, and $\sigma_b$). We also added (in quadrature) to this term 0.5 ppm as the contribution of the model uncertainty (defined as $\sigma_m$). We also added an extra increase of these uncertainties (factor 1.2) to get the best fits between both observations and the prior estimate.

5. By focusing on the annual mean fluxes for this region, the authors are going after the toughest part of the estimation problem, more difficult than getting the seasonal cycle of flux correct, for example.

   We agree that annual to the annual mean flux at region scale is the most challenging problem.

However, our annual flux estimated by our inversion (-0.41 $\pm$ 0.08 PgC y$^{-1}$) falls within the ensemble mean estimate of the MIP OCO-2 (-0.17$\pm$ 0.26 PgC y$^{-1}$ (see section 4, Fig.14a), same as the seasonal cycle (Fig.14b), except for July. We discuss in Section 4 of the manuscript the caveats of the inversion in winter.

6. It has been shown that errors in representing *in situ* measurements (especially those affected by continental air) in the global transport models result in measurement biases that get translated into annual mean flux biases in the global inversions. The early TransCom comparisons of regional fluxes from different global inversions regularly showed timeseries that moved up or down together with a similar seasonal and interannual variability, but were separated by offsets caused by these biases. Regional inversions are affected by the same sort of errors through the boundary conditions assumed (taken often from these same global inversions). The shift from the mainly surface-based *in situ* data to column-integrated measurements from satellites should reduce the bias problem simply by having a much denser and globally-distributed data constraint. However, counter-balancing this are two added problems: 1) biases in the retrieval of CO2 from the measured radiances (these are on the order of a half ppm in the column at the scale of continents and seasons), and 2) the need to rely on transport models more to relate the information in the upper part of the column to fluxes far afield. Were it not for the retrieval bias problem, in particular, more satellite-based flux inversion results of the sort of study being done here would have appeared in the literature before now.

   We agree that biases in OCO-2 observations are a problem and may lead to erroneous posterior flux estimates just like biases in the lateral boundaries in regional inversion. However, as we stated earlier in this document, our inversion is able to handle potential biases introduced by boundary conditions (BCs). Besides, our BCs rely on the latest version of CAMS global inversion product, which also assimilate OCO-2 data in the inversion.

7. The comparison of the a posteriori CO$_2$ concentrations obtained here to independent data is a strength of the manuscript. However, the systematic differences of 1 ppm or more over half the year that are obtained when comparing to the three local TCCON sites, as well as the half-ppm difference of opposite sign obtained in comparison to the *in situ* data at Cape Grim, do not give one confidence that the annual mean fluxes are correct. Differences of half to 1 ppm are not negligible and are the sorts of differences that a half or full PgC/yr error in the annual mean for the region might cause. Not knowing whether the flux estimate should be pushed more in the direction of the TCCON or Cape Grim data is also obviously a problem (i.e. we don't know whether our validation data themselves are reliable).

   It is clear that we obtain better reduction of the prior biases in summer and Autumn season at TCCON sites (biases less than 0.5), and reaming biases of about 1 ppm are seen in winter. We explain in the manuscript (Section 3.5.1) 1 ppm bias might be explained by spurious OCO-2 soundings affected by biomass burning aerosols seen in that period. In northern Australia, winter occurs in dry season, and it is highly impacted by wildfires (see Supplementary; Fig. S4). OCO-2 spectrometers measure reflected sunlight from Earth's surface, and regions heavily affected by fires can lead to a modification of the light path length because the instrument

struggles to distinguish between photons reflected by intermediate scatterers and photons reflected from Earth's surface (O'Dell et al., 2018).

In the main manuscript (Section 3.5.2), we also showed that adding ocean glint observations in our inversion improves the prior biases at the Cape Grim site. Cape Grim is an oceanic monitoring site that is "strongly" impacted by ocean fluxes, and therefore does not represent the air masses that cross the Australian continent the majority of the time. It is a challenge to validate our results with few TCCON and *in situ* sites around the country (which only represent a few grid-cell over Australia). It would be ideal to have more stations around the savanna or sparsely vegetated ecosystem to compare posterior concentration. Unfortunately, we do not have these stations in Australia. But, still, the comparison with MIP OCO-2 lends confidence to our results.

8. For this paper to be publishable, the authors should either back off from the emphasis on the absolute flux levels and focus instead on variability, or else add the sensitivity studies necessary to quantify the uncertainty in the annual mean fluxes related to the systematic errors mentioned above.

   David Baker's comments have been taken into consideration fully in the revised manuscript. We have performed three different sensitivity experiments (explained in the first section of this document) to show that our regional inversion is self-consistent, and the flux estimates over the Australian continent fall within the most recent estimates made by the OCO-2 community (MIP). The focus on the variability of the carbon flux over Australia is a working paper under development.

9. Even if the first approach is taken (focusing only on variability, meaning here the seasonal cycle), some quantification of the impact of the systematic errors on the flux variability would still be required. One approach that might help with this would be to look at the comparison of OCO-2 flux estimates from a suite of different global inversions collected as part of the OCO-2 flux inversion model intercomparison project (MIP). Fluxes obtained for the Australia or Australasia region from those runs could be collected and the spread used as a measure of some of these sources of uncertainty (the different estimates in the suite having been obtained from inversions that used different transport models, prior fluxes, and inversion methods).

   We compare our assimilated fluxes with the OCO-2 and in-situ MIP global flux inversion. For details please see section 4 (Discussion).

10. The key issue of biases in the OCO-2 data could perhaps be examined also using these MIP results, as well as by doing an inversion using the OCO-2 ocean glint data inside the regional domain.

    We performed an independent experiment where we added to the inversion OCO-2 ocean glint data. We decided not to discuss the Australia flux results based on this inversion because ocean glint observations (version 9) still have undetermined biases (O'Dell et al., 2018) that might mislead the results over the Australian carbon flux estimate. However, we included these findings in Appendix F, Table 1, and decided to examine whether the biases in the posterior

concentration improve the agreement with independent observations. For details, see section 3.5 (Evaluation of the inversion with independent data). The MIP OCO-2 comparison is made in Section 4 (Discussion).

11. Another approach might be to try to use a larger domain (with the inversion run at a coarser resolution) to see how much the annual mean flux result changes.

    We believe that running the inversion at a coarser resolution would require quite a lot of work, and our study domain already includes a considerable buffer around Australia (Please see Fig.2 of the main manuscript).

    All this is unfortunately a lot of work, but it seems to be needed to get an idea of how robust the sink obtained here actually is. It should be noted that this applies to not just this study, but to regional inversions of satellite $CO_2$ retrievals more generally.

    We agree that there can be issues with the regional inversions with satellite $CO_2$ retrievals, but the BCs adjustment should deal with the systematic errors from long-distance transport.

    Finally, I would offer my opinion that regional inversions of satellite $CO_2$ data are not the ideal way of handling these data, since much of the information contained in the column pertains to fluxes emitted outside the domain (and conversely, much of the information useful for constraining fluxes inside the domain, especially their annual means, is contained in satellite measurements taken outside of the regional domain). A global inversion that uses all the data together, perhaps with a high-resolution model nested over the region of interest, would be a better approach. All this is due to the full-column nature of the satellite measurements regional inversions do much better when the data are taken close to the surface and reflect local fluxes more directly. If the authors are intent on using a regional inversion for this problem, they should do a better job of quantifying the uncertainty in their flux estimates due to their treatment of the boundary conditions.

    We respect David Baker's opinion, but it is hard to get high-resolution regional posterior fluxes without a regional inverse modelling study. The only other way would be to use a global transport model with a stretched grid (e.g. MPAS, adapted to include passive tracer transport) that **also** has an adjoint. We agree that interference from the boundaries is a problem with regional flux inversions for satellites, and that better solutions (such as noted above) are likely to appear, but for the time being the regional scale flux inversion with a boundary condition adjustment represents a compromise.

**Detailed comments (for line numbers indicated):**

1. 4: Does a negative sink indicate a source? Be careful to distinguish between fluxes (which can be + or -) and a source or sink, which should be unsigned (the sign of the flux being indicated by whether it is called a source or sink)

   In this paper we adopt the atmospheric convention where a negative flux indicates removal from the atmosphere (a sink, quoted with a negative sign), and a positive value indicates an addition to the atmosphere (source).

2. 15-16: I'm glad you bring out this difficulty in matching both column and surface data in the presence of transport model error – this is often ignored.

3. 115-124: It is not clear which fields EDGAR estimates are being used for just the aviation and marine transport emissions missing from ODIAC? Please clarify in the text.

   We have modified the paragraph as follows: Anthropogenic fluxes were created by the combination of two different inventory data sets: the Open-source Data Inventory for Anthropogenic $CO_2$ (ODIAC) (Oda et al., 2018) and the Emissions Database for Global Atmospheric Research (EDGAR), version 5 (Crippa et al., 2020). The combination of these two anthropogenic inventories (each used to cover different source sectors) was necessary because the version of ODIAC dataset selected did not contain emissions from aviation. The EDGAR emissions combined with ODIAC were aviation climbing and descent, aviation cruise, and aviation landing and takeoff dataset. Aviation emissions were also distributed across the vertical layers of the CMAQ domain. EDGAR is a gridded product, which has spatial resolution of $0.1°$ $\times 0.1°$ with monthly temporal resolution. ODIAC (version 2019) is also a gridded product, which has a spatial resolution of $1 \times 1$ km at grid-cell scale. Monthly ODIAC fluxes were modified by incorporating a diurnal scale factor estimated by (Nassar et al., 2013).

4. 232-233: The explanation given here for the a posteriori cost function values being too high by a factor of two is correct IF the errors are unbiased and gaussian. Another possibility that ought to be mentioned is that they are not gaussian and independent but rather have errors correlated in time and space (including flat biases) that render the statistical assumptions made in deriving the estimation method invalid and lead to a higher cost function than expected.

   We thank to the reviewer for his comments. We have included them in Section 2.3 of the manuscript as follows

   The error statistics of $\mathbf{x^a}$ are embodied in the posterior error covariance matrix ($\mathbf{A}$). In this study, $\mathbf{A}$ was computed by a series of observing system simulation experiments (OSSEs) carried out by (Villalobos et al., 2020, Section 2.4.). However, here we adjusted the prior and observation uncertainties assumed in Villalobos et al. (2020) by a factor of 1.2. We made this adjustment to satisfy the theoretical assumption in the variational optimization, which indicate the value of the cost function in its minimum has to be approximately equal to half of the number of observations (for more details see Section 3.1). In general, errors assumed in the inversion are not gaussian and independent but rather have errors correlated in time and space (including flat biases) that render the statistical assumptions made in deriving the estimation method invalid and lead to a higher cost function than expected. A description of how the prior and observation uncertainties were assumed in our study is found in Sections 2.3 and 2.5. Appendix D (Figs. D1 and D2) show the spatial distribution of the prior and posterior that we reference in this study.

5. 233-234: "To compensate, the posterior uncertainties estimated in Villalobos et al. (2020) were increased by $\sqrt{2}$." I think you mean prior uncertainties instead of posterior, no? Also, please indicate whether the uncertainties on the measurements were increased in addition to those on the prior fluxes.

The text mentioned above no longer exist in the manuscript, because we re-run the inversion and increase both prior and observations by a factor 1.2. We decided to adjust both to make the changes conservative.

6. 264: the prior overestimates the measured $CO_2$ when?

Line 264 was modified for lines 352-354, page 19): From the inversion viewpoint, the significant shift between prior and posterior fluxes occurs because the prior column averaged simulated by the CMAQ model overestimates the column-average retrieval by OCO-2 in these periods (see Fig. 4).

7. 265: Are these posterior-prior differences that you refer to FLUX differences or differences in CO2 concentration? If the latter, are they XCO2 or something else? [Figure B1g does not relate to either, so I can't tell...]
We pointed to the wrong Figure. We should have referred to (Appendix A; Fig. A1). because this plot shows the spatial distribution of posterior and prior flux for each month in 2015. This error has been amended in the manuscript (Line 364, page 19).

8. 267: The figure pointed to here (B1g) only shows the number of OCO-2 data points over the domain in July 2015. It is not at all clear how that relates to the posterior-prior differences referred to in this sentence. Why are you pointing to that figure? Do you mean A1g?

As we have answered in the previous question, we pointed to the wrong plot. We should have referred to Appendix A (Fig.A, panel g) (Line 364, page 19).

9. 272: "...most of the increased in the posterior flux" By "increase in the posterior flux", do you mean a shift towards more positive values, reflecting the greater prevelence of red values in SE Australia in A1h&i? Or an increase in the uptake from the previous month (blue values in A1)?

We have restructure the paragraph to better convey our results.

Modified text (line 370-374, page 19): An increase in carbon uptake estimated by our inversion in August comes from the northern and southern region of Australia(with the exception of coastal areas in the southern-east corner of Australia), which mainly shows a release of carbon (Appendix370A; Fig. A1, panel h). The release of carbon by the land in this coastal region is likely attributed to a decrease in land productivity(Supplementary information, Fig. S1, panel h). The subtle decrease of photosynthesis activity in the coastal area is likely associated to a decrease of rainfall seen in June and July (Supplementary Fig S2, panel h).

10. 273: "(Supplementary Fig S1, panel g, h)" The flux changes are shown in Figure A1 – do you mean to point to that figure, instead?

This has been corrected.

11. 273-275: "...which again lines up with a higher than usual increase in land productivity." The largest signal I see in SE Australia in Aug-Sept 2015 in Fig A1 h&i are the (red) positive flux anomalies next to the coast, which seem to overwhelm the (blue) negative anomalies further inland. These cannot be explained by the higher EVI values in your argument.

We agree with the reviewer that we only focus on the spatial pattern of the negative flux (blue color) seen from August to September (Appendix A, panel h and i), and we did not discuss much about the positive fluxes seen in the coastal areas in Southern Australia. To give a better explanation of our results, we have restructured the paragraph from the previous manuscript.

In September, the posterior carbon uptake primarily comes from the southeast corner of Australia (with a slight exception seen in the southeast and east coast of Australia), which shows a release of carbon into the atmosphere (Appendix A; Fig. A1,panel i). The carbon uptake seen in the southeast corner of Australia aligns up with a higher than usual increase in land productivity, as reflected by the positive EVI anomalies in that region (Supplementary Fig S1, panel h), likely benefited by the positive rainfall anomalies seen in August in that area. In September, we also see that positive EVI anomalies were not as strong. These findings are probably associated with the fact that rainfall in September decreased considerably for most parts of the country, where rainfall was lower than average (negative rainfall anomalies) (Supplementary Fig S2, panel i).

12. 274: "Positive EVI anomalies in this period was not as strong as in July." To my eye, the strong EVI anomalies seen in July (Fig S1g) more or less continue at the same magnitude, or very close to it, in the following two months (S1h & i) – and yes, perhaps waning a bit in September. By "these findings", do you mean the reduction in EVI? That makes sense, but the sitll-strong uptake in the south from your inversion does not agree with it... Maybe you could reword this to be clearer?

Please see previous answer.

13. 277-278: "Spatial patterns of the EVI anomalies during these months are expected because rainfall is one of the most important drivers of ecosystem dynamics and productivity."

Agreed, however, the agreement between your EVI plots and your rainfall plots does not really support this well. In particular, in the August-September time frame that you are discussing here, most of the EVI anomalies in SE Australia are positive (green) at a time when the rainfall anomalies are switching to negative (dry). The two areas in southern Australia in September that have negative EVI anomalies (orange) are both located in areas of positive rainfall anomalies. Basically, the argument you make in the text (which is reasonable) is not well- supported by the figures you point to.

We do not fully agree with the reviewer. It is true that positive rainfall anomalies switch from positive (August) to negative (dry) in September in Southern Australia. However, If the reviewer carefully sees the spatial distribution of the stronger negative flux anomalies (blue grid cells colour) in September (Appendix A, Fig.A, panel i), he should notice that they follow a similar spatial pattern of the positive rainfall anomalies seen in August, suggesting that most of the growing in vegetation in September is benefited by the rain in the previous month.

14. 294-306: In this whole discussion of the savanna fluxes, it surprises me that you don't point at all to your EVI or rainfall maps in the supplementary material. The strong rainfall anomaly in January over the savanna in the north supports your argument, as does the positive EVI

values there in the same month. However, these favorable growth conditions quickly change to unfavorable ones in February (strong negative EVI values (red) broadly across the savanna in the NE, and dry conditions starting in March) and stay unfavorable all the way through September, a period when your inversion is giving you an anomalous uptake there the whole time. To make these two lines of evidence agree, you would have to argue that the benefit of the wet conditions in January allow increased growth and carbon uptake that last all the way through the dry half year that follows.

We agree with the reviewer and we have restructured the paragraph from the previous manuscript.

Modified text (line 408-416): Fig. 9 shows the monthly time series of the prior and posterior terrestrial flux aggregated into these bioclimatic regions.Over the savanna ecosystem (Fig 9b), our inversion indicates that from January to June, this ecosystem acted as carbon sink. In February, in this ecosystem, we see that the prior sink (-0.48 $\pm$ 0.40 PgC y$^{-1}$ ) strengthens to a posterior of -1.07 $\pm$ 0.10 PgC y$^{-1}$. The stronger carbon sink (relative to the prior) from January to March coincides with an increase of greenness in vegetation (positive EVI anomalies) in this ecosystem (see Supplementary; Fig. S1, panel a and b), benefited by anomalous rainy conditions in January (see Supplementary; Fig. S2, panel a). Thus, it seems that the anomalous increase of rainfall in northern Australia in January benefits the increase in vegetation growth and carbon uptake recorded in February. However, it is difficult to draw conclusions about the posterior carbon uptake seen in months subsequent to March because of unfavourable raining conditions and negative EVI anomalies in these periods.

15. 307-312: Similarly for the warm temperate region, your finding of reduced C uptake in the posterior does not agree with the EVI data, which show increased greenness across the region for most of this time.

We restructured the paragraph in the manuscript

Modified text (line 426-435, page 22): Over the warm temperate region, from February to April our posterior estimate suggests a carbon source (Fig 9c). For this425period, we cannot determine if the prior flux estimate was a carbon sink or source due to its uncertainty range. In February, the prior flux (-0.05 $\pm$ 0.08 PgC y$^{-1}$) becomes a posterior carbon source of 0.17 $\pm$ 0.06 PgC y$^{-1}$. In March, the prior estimate was nearly neutral (0.04 $\pm$ 0.05 PgC y$^{-1}$) compared to the posterior carbon source estimate (0.17 $\pm$ 0.05 PgC y$^{-1}$). The reduced carbon uptake estimated by the inversion in this period does not agree with the positive EVI anomalies seen in this region,however it is likely that this extra carbon release to the atmosphere is related to a increase of leaf respiration in response to high temperatures recorded in 2015 for the majority of Australia (Annual climate statement, Bureau of Meteorology, 2015). Another possible reason for the relatively small shift from the prior in this period was most likely because the CABLE-BIOS3 GPP over-estimates MODIS GPP (Appendix E1, Fig. E1, panel c). For the warm temperate category, the correlation of CABLE-BIOS3and MODIS GPP flux is high (R = 0.86).

16. 313-321: For the cool temperate region, there does not seem to be any indication in the EVI or

rainfall data of any strong driver for these shifts to C release that you obtain in your inversion. For this region, as in those above, your posterior flux estimates don't seem to correlate well with the EVI or rainfall metrics. At least you bring in the MODIS GPP metric and discuss errors in your CABLE-BIOS3 prior, which is relevant because you discuss your results in terms of a shift away from the prior.

We can't entirely agree with the reviewer on this point. If he carefully sees Fig. S1, panels d and e in the supplementary information, he will notice that the cool temperature ecosystem was affected by both positive and negative EVI anomalies. In April, we see predominately negative EVI anomalies in the southern corner of Australia (mainland) and Tasmania located to the south of the continent. We have modified the text to better explain these findings.

17. 325: Here, the rainfall and (especially) EVI maps do support the greater uptake in August. I would point also to the EVI map, as that is more convincing than the rainfall map.

In order to improve our results, we calculated the percentage of increase in rainfall for 2015 (relative the mean 2000–2014), and we found that the positive rainfall anomalies was high in August (greater than 60%) over the Mediterranean ecotype. We have restructured the paragraph that discusses the results over the Mediterranean ecosystem, so the reader can understand much better the results discussed here.

Modified text (line 448-453, page 22): Another disagreement between the prior and posterior terrestrial flux estimates is seen over the Mediterranean ecoregion in August (Fig.9e). Our posterior estimate is a flux of -0.35 $\pm$ 0.08 PgC y$^{-1}$ compared to the prior of -0.12 $\pm$ 0.12 PgC y$^{-1}$. An increase in vegetation productivity may also be the reason for the increase in the amount of carbon taken up by this category (positive EVI anomalies, supplementary, Fig S1, panel h). This larger carbon uptake was likely a consequence of an increase of rainfall in this category (greater than 60% on average (relative to the mean 2000–2014) for some areas of this ecosystem, supplementary S3, Fig, h). We also found that CABLE-BIOS3 underestimates MODIS GPP by 0.2 PgC y$^{-1}$.

18. 350: I could not find the indicated Table A1 in Appendix A – where is it?

We have hyperlinked Table A1 located in Appendix A.

19. 345-353: This whole discussion of the biases with respect to the Darwin TCCON site seems a bit beside the point. The broad view is that there are very large seasonal differences between the TCCON estimates and both the prior and posterior estimates from the inversion, and that assimilating OCO-2 data does not significantly reduce them. Some discussion of why this might be the case and what it means is warranted. The OCO-2 data themselves are bias-corrected using the TCCON data, so why shouldn't they drive the result to agree with the TCCON data? (One answer: there are still site-to-site biases between OCO-2 and TCCON, some of which we may be seeing here.) Could the difference be due to errors in the TCCON retrievals, maybe? If the positive bias is due to improper treatment of fire carbon emissions in the prior, as suggested, why wouldn't the posterior estimate correct these strongly (the inversion assimilating column CO2 data, which should be less sensitive to the vertical location

of the emissions)? The TCCON data also see any fire emissions in the column, again being less sensitive to the emission level.

We understand the reviewer's point of view, so we have restructured the paragraph that describe the results at the Darwin TCCON site. Note that Section 3.5.1 (Comparison with TCCON observations) was modified. Now, the comparison against TCCON includes not only the posterior concentration derived by fluxes that assimilate land nadir and land glint OCO-2 observations (LNLG) but also posterior fields derived from both land (nadir and glint) and ocean glint observations (LNLGOG).

Modified text (line 490-505, page 25): In Fig. **??**a, we see that in late spring, summer and early autumn in Australia (November to March) the posterior column-average simulated by CMAQ model (LNLG) is in better agreement with TCCON Darwin estimates than the prior. In this period, prior mean biases were reduced by approximately $30-80\%$. For example, in November and December, the prior concentration biases were reduce from -0.25 (RMSE = 0.51) to 0.11 (RMSE = 0.42), and from -0.34 (RMSE = 0.48) to -0.02 (RMSE = 0.31) respectively. Large seasonal differences (approximately 1 ppm) are seen between the TCCON observation, the prior and posterior column-average concentrations (LNLG) from June to September. Despite the fact that we see an improvement of the prior biases in this period, assimilating OCO-2 data does not significantly reduce them. The remaining posterior concentration biases of about $+1$ ppm might be explained by spurious OCO-2 soundings affected by biomass burning aerosols seen in that period. In northern Australia, winter occurs in dry season, and it is highly impacted by wildfires (see Supplementary; Fig. S4). OCO-2 spectrometers measure reflected sunlight from Earth's surface, and regions heavily affected by fires can lead to a modification of the light path length because the instrument struggles to distinguish between photons reflected by intermediate scatterers and photons reflected from Earth's surface (O'Dell et al., 2018). In terms of the posterior bias improvement when fluxes have been assimilated by OCO-2 LNLGOG data, we can see that the improvements are negligible, and in some periods such as January or May, the posterior biases get worse. This result suggests that the uncharacterized OCO-2 glint ocean bias degrades the performance of the inversion. We also found that our posterior column-average concentrations were better correlated with TCCON in comparison with the prior concentration (Appendix G, Table G1).

20. 357-364: Again, in the discussion of the Wollongong site, the key point is not captured: strong systematic differences between TCCON and the OCO2-driven inversion remain, even after the OCO-2 data are assimilated. Why? A 1 ppm difference extending across over half the year is not a "small" difference in terms of what it implies about flux errors.
362-364: There is little suggestion in the figures that are pointed to that the large difference in August at Wollongong is related to data density – the winds are mostly coming from the west then, where the OCO-2 data are located (i.e., over the continent).

We agree with the reviewer, and a better explanation of the results at TCCON Wollongong site has been included in Section 3.5 (Evaluation of the inversion with independent data). Paragraph 357-364 has been modified as follows:

Modified text (Line 506-525, pages 25, 16): At Wollongong site, we see a relatively slight reduction of the prior biases in February, March, and November (spring, summer and early autumn season in Australia) (Fig.10b). In November, for example, prior negative biases of about -0.74 ppm (RMSE = 1.22) were reduced to -0.40 ppm (RMSE = 1.13). The small reduction of the biases in this period is likely associated with strong winds coming from the ocean to the TCCON station. Wollongong TCCON site is strongly affected by ocean fluxes, which are less restricted by our inversion when we only use LNLG observations. In late autumn and winter at Wollongong, we also see a consistent overestimation of the posterior column average (LNLG) simulated by CMAQ. July is the only month in winter that we see slight improvement of the prior biases (Fig.11b). Significant high positive posterior biases (range between 1.1 and 1.61 ppm) seen in the winter season are likely related to OCO-2 than TCCON biases. It has been found that passive satellite instruments have difficulties measuring at high and middle latitudes in winter because the sun stays low in the sky (Wunch et al., 2017). A low solar altitude angle corresponds to a high solar zenith angle and high air mass, which means it takes longer for the sunlight to reach the satellite instrument. Biases related to high airmasses ("long path length") can be obtained because the absorption spectra tend to saturate at the line centre, causing the column line shape of the absorption line to be more sensitive (Jacobs et al., 2020). To evaluate whether the zenith angle influences TCCON retrieval bias, we filtered the TCCON dataset and only selected solar zenith angles $\geq 40$ and $\leq 50$ (see Appendix G, Table Table G3). We found a slight improvement of the posterior biases, which means TCCON retrieval bias is not likely the reason for the biases seen in winter. Similar to the Darwin site, we did not find an improvement of the prior biases by adding ocean glint data to the inversion. Besides, ocean glint data (as it is shown in Appendix C, Fig.C2, panel d-h) is quite sparse around the Wollongong site providing not much constraint on carbon fluxes around this location.

21. Why would winds blowing off of the ocean necessarily lead to negative biases? Are you saying that there is some problem with your prior $CO_2$ fields over the ocean? What is causing that?

We set up small uncertainties over the ocean compared to land, so somehow we restrict the ability of OCO-2 to modify ocean fluxes when are using LNLG only. In the current version of the manuscript, we showed that adding ocean glint data help to restrict ocean carbon fluxes ( mainly when winds blow from the ocean to the monitoring site).

Modified text (line 539-548, pages 28,29): As illustrated in Fig.12a, the inversion using only LNLG OCO-2 observations does not match Gunn Point observed concentrations well except in September. Most biases are negative, indicating that the posterior simulation at the surface of the CMAQ model underestimates the observations. The prior concentration indicates a better agreement, but biases are still significant. One possible explanation for the large negative posterior biases in January, February, March and December might be related to strong westerly winds that blow from the ocean to this site location (see Supplementary; Fig. S9). Using only LNLG observations restricts our inversion to optimized ocean fluxes because the uncertainties set-up it over ocean were lower compared to land. However, when we added ocean glint observations the posterior concentration biases get better. These results are not unexpected

because Gunn Point is a coastal site largely affected by ocean carbon fluxes. In February, and when the wind comes from the ocean, the posterior bias using LNLGOG shows a significant improvement compared to prior bias concentration. Here, we see a reduction of the bias from 1.93 (RMSE = 4.21) to 0.65 (RMSE = 3.79).

22. 380-384: It is argued that the difference at Gunn Point is due to a high bias in the *in situ* data, a single high reading possibly related to fires. However, in Figure 9a, the absolute values for both the *in situ* and estimated $CO_2$ levels are given, and the July *in situ* value seems to be in line with seasonally-expected values, lying in between the June and August values. It is the estimated $CO_2$ value which is the outlier. So the explanation given does not agree with this.

The reviewer is right in this point, so we have restructured the paragraph that describe the results at the Gunn Point site.

Modified text (Lines 539-557, pages 28, 29): As illustrated in Fig.12a, the inversion using only LNLG OCO-2 observations does not match Gunn Point observed concentrations well except in September. Most biases are negative, indicating that the posterior simulation at the surface of the CMAQ model underestimates the observations. The prior concentration indicates a better agreement, but biases are still significant. One possible explanation for the large negative posterior biases in January, February, March and December might be related to strong westerly winds that blow from the ocean to this site location (see Supplementary; Fig. S9). Using only LNLG observations restricts our inversion to optimized ocean fluxes because the uncertainties set-up it over ocean were lower compared to land. However, when we added ocean glint observations the posterior concentration biases get better. These results are not unexpected because Gunn Point is a coastal site largely affected by ocean carbon fluxes. In February, and when the wind comes from the ocean, the posterior bias using LNLGOG shows a significant improvement compared to prior bias concentration. Here, we see a reduction of the bias from 1.93 (RMSE = 4.21) to 0.65 (RMSE = 3.79).

In winter (June to August), we see that the posterior biases using either OCO-2 LNLG or LNLGOG show no improvement, and the prior biases are in better agreement with the observations. One possible explanation might be related to the fact the column-integrated $CO_2$ measurements are less sensitive to near-surface dynamics compared to *in-situ* measurements (Lauvaux and Davis, 2014), or to remaining bias in the OCO-2 data. Despite the fact that version 9 has an improvement in the biases correction, in the recent study performed by OCO-2 MIP (Peiro et al., 2021) shows that LNLG data still has large negative latitudinal biases in the Southern Hemisphere. Another potential explanation could be associated with an inaccurate representation of vertical transport within the planetary boundary layer in winter by the CMAQ model. Incorrect vertical transport might lead to erroneous horizontal distributions of air masses (Lauvaux and Davis, 2014). Therefore, correcting the prior column-average simulated by CMAQ to match OCO-2 might not improve near surface simulations.

23. 402: "...savanna, Mediterranean and sparsely vegetated ecosystem drove this higher posterior uptake." I would suggest removing the Mediterranean region from this list. The tropical region actually had a larger shift from the prior, yet you don't include it. Only July saw a significant

change from the prior in the Mediterranean ecoregion...

We have restructured the paragraph (Lines 578-584, page 32): We saw that semi-arid ecosystems in Australia, such as savanna and areas with sparse vegetation, are responsible to some extent for the stronger carbon sink (relative to the prior flux) recorded in 2015. We associated this carbon uptake with an increase in vegetation productivity (positive EVI anomalies) and an underestimation of the GPP flux by the CABLE land surface model. We speculate that the LAI estimated by the land-surface model CABLE-BIOS3 fails to capture the abrupt response of the terrestrial biosphere to rainfall over areas with sparse vegetation. This hypothesis could be tested by comparing CABLE-BIOS3 LAI with satellite-LAI. However, this is beyond the scope of this study and will be taken up in a forthcoming article.

24. 411: The annual mean for the ensemble of global models that you give ($+0.63$ PgC/yr) does not seem to agree with the monthly numbers you give for the ensemble mean in Figure 11b. If you add up the monthly numbers in 11b and divide by 12, you get something more like $+0.1$ or $+0.2$. Which is correct, the annual mean you give or the monthly numbers plotted in 11b?

The reviewer is correct; the annual ensemble of the five models included in the previous manuscript was about $0.21$ PgCy$^{-1}$. We effectively forgot to divide the fluxes by 12. We decided to replace these results for the annual ensemble mean MIP *in situ* global inversions in the current version of the manuscript. The annual ensemble mean of the MIP *in situ* observation was $0.21 \pm 0.39$ PgC y$^{-1}$. For details, please see Section 4 (Discussion).

25. 415: It is not clear where the seasonal amplitude of $0.22$ PgCy$^{-1}$ for the global models comes from. Based on what is plotted in Figure 11b, it should be more like 2.7 PgC/yr. Are you sure you didn't accidentally divide by 12? The 3.46 number for the inversion results also looks a bit high, based on what is shown in Fig 11b.

As we answered in the previous question, the results from the ensemble monthly mean of five *in situ* global inversions was replaced by MIP *in situ*. In sum, the analysis of seasonal variation shown in Table 4 indicates the flux inversion made by PCTM model is the one with the largest seasonal amplitude (5.78 PgC y$^{-1}$), and CT is the one with the lowest amplitude (1.27 PgC y$^{-1}$). For more details about these results, please see Section 4 (Discussion). In Appendix I (Fig I1 and Fig I2), we plotted the monthly seasonal cycle of these ten independent models (the MIP *in situ* and OCO-2 (LNLG) for 2015). Our inversion seasonal amplitude was 3.07 PgC y$^{-1}$.

26. 423-424: "Monthly biases of simulated concentrations compared to TCCON sites at Darwin, Wollongong and Lauder generally improved using posterior rather than prior fluxes." I would have to disagree – based on what you have shown in your plots, the improvement is only very slight. Mostly, the seasonal biases that are there in the prior remain there in the posterior.

We understand the reviewer point of view. However, after re-running the inversion again and adjusting the uncertainties in our inversion system. We were able to get a better reduction of the biases at the TCCON Darwin site. We saw a decline of the prior biases to less than 0.1 ppm (mainly in summer months). In winter, we also saw a reduction of the biases, but this

one was not as significant as the summer months. We explained the potential reason in the paper for these findings. TCCON and Lauder site the reduction was slight, but these stations are strongly influenced by ocean fluxes, which we did not give much freedom to be modified by the inversion. For more details, see section 3.5.1 (Comparison with TCCON observations), and Tables in Appendix G.

27. 424-428: To me, this seems like an overly rosy view of the results. It seems to me that there are persistent biases between the results of the inversion and the independent data: the $CO_2$ values are too low across the year when compared to the three TCCON sites (except during the austral Spring for the two sites over Australia), and too high when compared to the Cape Grim *in situ* site. What causes these biases, and why are the inversion results lower than TCCON in the column, but higher than Cape Grim near the surface? For a paper trying to pin down the absolute level of annual mean flux for this region, these systematic biases with respect to the available independent data are a big cause for concern. A 1 ppm difference in the column average is not a small bias, nor is a 2 ppm difference at a background site like Cape Grim.

We were optimistic about the results because we knew the challenges we were going to face trying to validate our posterior field against monitoring stations in close proximity to ocean grid cells. As we showed in the paper, adding ocean glint data helps us to significantly reduce the prior biases at Cape Grim and Gunn Point sites. Now, the question is, what will be the best option for optimizing carbon fluxes over Australia, knowing that glint data still have some uncharacterized bias that might mislead the national carbon budget.

28. 432-443: The potential of higher-resolution inversions noted here is well taken. However, the overall result of an annual mean uptake for 2015, as well as the factors affecting how it was obtained, have more to do with broad-scale features of the inversion that would not be substantially changed by going to finer resolutions. First, the data used (column data only over the land areas, mostly over Australia) – local biases in this, perhaps related to the near-desert nature of much of the area, might factor into the result. What would your inversion give if you included the OCO-2 ocean glint data as well? Yes, that data might be affected more by retrieval biases, but you could solve for that in the same manner that you correct for biases in the boundary fluxes coming into the domain.

We found that adding OCO-2 glint ocean data does not change much the annual mean uptake for 2015 (-0.36 PgCy$^{-1}$ compared to -0.41 PgCy$^{-1}$ using only land nadir and land glint data. For more details, please see section 3.2 (Australian $CO_2$ flux estimate). With regards to desert local biases, it is possible that local biases are still hidden within OCO-2 retrievals, however as indicated by Kiel et al. (2019), improvements of the surface pressure estimate in OCO-2 version 9 reduce the errors in the bias-corrected process for OCO-2. In this study, we used bias-corrected data and selected only soundings with good quality.

29. Second, the short span of the inversion – how important are errors in the initial condition to the result you obtain? If you vary your prior for the period leading up to the start of 2015, how do your annual mean results change?

We forgot to mentioned in the manuscript the we gave a spin up of 1 month to the inversion to deal with potential errors in the initial conditions (ICs). Besides, we also solve for them within our optimization. For more details, please see section 2.2 (Defining the control vector).

30. Third, and probably most important, what is the impact of using only a small amount of column-integrated satellite $CO_2$ data in a regional inversion of this sort on constraining the local fluxes? Most of the $CO_2$ information in the mid- to upper-column comes from fluxes emitted outside of the regional domain – only the lowermost part of the column contains information on the local fluxes. How do you ensure that errors in modeling the upper part of the column (in terms of your prior fluxes and CO2 fields, and your atmospheric transport) do not overwhelm the actual local information that you do have, resulting in spurious flux estimates? Sensitivity studies are crucial for handling these sorts of issues, and you really have not done any of those here. Also, given the importance of transport model errors in inversions of this sort, getting an idea of the model-related errors is important. One approach you could take to address this is to see where your annual mean results for Australia fall within the suite of inverse modeling results given by the OCO-2 flux inversion MIP (see Crowell et al, 2019 for the OCO-2 v7 MIP and Peiro et al. for the v9 MIP, the latter having just been submitted to ACPD).

As mentioned in the previous question, adding more OCO-2 soundings to our domain (e.g ocean glint data) does not has a impact in the annual flux mean for Australia. We do not believe that error in the upper part of the column should have much impact on the fluxes at the surface, because concentration of $CO_2$ gradually decreases from surface to tropopause ( 6 to  20 km) and exponentially decline in the stratosphere. Besides, our boundary-condition adjustment help us to correct any biases that CAMS global inversion data might have from the upper column of the atmosphere.

31. 448-449: "In general, they compared well with TCCON data when wind directions coupled our estimated fluxes to these observations." This is a generous interpretation of your flux estimates. In my view, the systematic errors with respect to the three TCCON sites and with respect to the background site at Cape Grim open legitimate questions as to the reliability of the annual mean flux estimates obtained here.

This question has been answer before. Please see question number 7.

32. 453-454: This focus on the Mediterrenean ecoregion, when it played a relatively minor role in the greater uptake estimates obtained, seems misplaced.

Agree with the reviewer. In the discussion section, we only focus on the savanna and sparsely vegetated ecosystem.

33. 455-456: "We also noted an increased seasonal cycle of flux, also suggesting greater productivity than the prior estimate." Is the increase in the seasonal cycle amplitude obtained here in comparison to the global models' still significant once the error in the global ensemble mean number (the 0.22 PgC/yr) noted above is corrected?

Yes, we found that Australian carbon fluxes assimilated with OCO-2 data show a larger seasonal amplitude than the ensemble mean of 5 *in situ* global inversions. However, and as mentioned in previous questions, we replaced the ensemble mean of 5 global inversions with MIP *in situ*.

34. Figure 4 vs. Figure 11: why are the blue lines on these two plots different? Aren't both of them the net land biospheric flux for Australia given by the regional OCO-2 inversion for 2015?

    Fig.4 (in the previous version of the manuscript), dotted lines showed a smooth line of the posterior fluxes. Fig.11 showed the same line as Fig.7, but the line was not smooth. In the current version of the manuscript, we use a consistent line between both Figures.

**Editorial comments:**

1. 6: I would put a comma after "southern Australia", for clarity

   We re-wrote the abstract.

2. 7: it seems odd to refer to "Mediterranean" by itself over Australia – I would continue to call it "the Mediterranean ecotype" if it were me...

   The word "Mediterranean" was removed from the abstract.

3. 7: define "EV" here

   We have defined the word "EVI" in the abstract.

4. 13: replace "which" with "that" (?)

   Corrected.

5. 14: add a comma before "though"

   We re-wrote the abstract, so line 6 to line 14 were not necessary.

6. 15: standardize your usage of "in situ" throughout document. Earlier you had a hyphen in it. I like it without the hyphen and italicized, but check the ACP preference on Latin words and go with that. No hyphen, in any case...

   We have standardize the usage of "in sit" throughout document.

7. 16: I'd replace "to match" with "in matching"

   Corrected.

8. 24: You are missing an end parenthesis for this beginning one

   Corrected.

9. 28-29: add commas before "along" and after "$CO_2$ data"

   Corrected.

10. 30-31: add commas around "launched in 20xx" (two places)

    Corrected.

11. 33: put "and none in Australia" either in parentheses or between commas

    Corrected.

12. 39: "are run at" ?

    Corrected.

13. 41: "run at 1 deg"; also, change the comma before "for example" to a semi-colon

    Corrected.

14. 43: change "the inversion" to "an inversion"?

    Corrected.

15. 44: add a comma before "since"

    Corrected.

16. 47-48: please indicate whether the anomaly that Poulter et al found was for the year of 2011 indicated in previous sentence, or some other year. You could just add "in that year" after "-0.66" to do so.

    Corrected.

17. 50: "Ma et al. (2016) found that this carbon uptake rapidly diminished..." It is not clear here whether you are trying to say that Ma et al got a smaller estimate of uptake than the two previously-cited references, or whether they found a similarly large uptake that somehow rapidly diminished across the 2011-2012 time period. Please reword this to clarify.

18. Footnote on page 2: On this footnote, you could note that when you refer to uptake or emissions, you will still use the signed flux values, so that the reader is not confused by, say, a negative uptake value.

    Corrected.

19. 58: remove comma before "and"

    Corrected.

20. 59: define "EVI" here and in the abstract

    Corrected.

21. 60: replace "against" with "and"

    Corrected.

22. 64: add a comma before "we"

    Corrected.

23. 68: remove comma before "involves"

    Corrected.

24. 70: add "likelihood" after "maximum a posteriori"? Otherwise, does not make any sense. xxx

25. 71: get rid of the parenthesis before "Rayner" and move it before "2019"

26. 76: italicize the second "x"

    Corrected.

27. 77: the "b" should be super-scripted, not sub-

28. Figure D1, caption: You should indicate what quantity these uncertainties apply to. I think you mean them to apply to the annual mean flux estimate across 2015, but you need to indicate this.

29. 95: add a comma before "x" (in addition to the one after it that is already there)

    Corrected.

30. 97: When you say here that you solve for scaling factors on the boundary conditions, this conflicts with what you say in the following sentence, where you say that you solve for additive offsets to the boundary conditions. Please clarify this to indicate which of these two options you actually use.

    Modified text (Lines 51-59): Australia has recently been subject to attention from the global carbon cycle community due to a large terrestrial carbon sink anomaly recorded in 2011 (Poulter et al., 2014). Poulter et al. (2014) found that Australia's flux anomaly was -0.66 for 2011 (relative to the 2003–2012 mean). Trudinger et al. (2016) also found a similar carbon sink anomaly for this period (ranging between -0.40 to -0.61 PgC $y^{-1}$). These studies suggest that Australia's ecosystems might act as strong sinks of $CO_2$ in the future during extreme wet periods. However, the efficiency and the spatial distribution of these carbon sinks remain largely uncertain (i.e., Ma et al., 2016). Some studies (e.g., Ma et al. (2016)) found that the anomalous carbon uptake recorded in Australia in 2011 rapidly diminished in the following year ($\sim$ 0.08 PgC $y^{-1}$), suggesting that semi-arid ecosystem can act as carbon sink in the short term but not over longer periods compared to tropical forest ecotypes. An important unanswered question in carbon cycle research remains regarding the carbon sink strength of semi-arid ecosystems in non-wet years.

    We re-wrote section 2.2 (Defining the control vector) to clarify how we treat BCs in our system.

31. 102: add a comma before "so"

    Corrected.

32. 111: change to "ran the CABLE model in the BIOS-3 setup"

    Corrected.

33. 111: "hourly" does not need to be capitalized

    Corrected.

34. 162: instead of interpolating to the "CMAQ vertical profiles", I think you mean to the "CMAQ vertical levels", no?

    Corrected.

35. 163: would be useful if you wrote out this equation here in the text instead of forcing the reader to go to Conner et al to get it

    We have added Connor Equation to the manuscript. For details, please see section 2.5 (OCO-2 satellite information and its uncertainties).

36. 165: rewrite this for clarity. They aren't daily averages but rather 1000-1400 local time averages, correct?

    We agree with the reviewer, and we modified that text in the manuscript.

37. 168: "in situ"

38. 172: again, use of "daily" to describe a subset of the full day is a bit confusing – please reword

    Modified text: For the comparison with our model simulation, we used hourly data from these monitoring sites, but the monthly mean averaged data shown in Section 3.5.2 were calculated using local time averages between midday and 05:00 p.m.

39. 185: "data that comes from the land" – it would be better to reword this to "data that was collected over land", because we the origins of the air in the column can "come from" over the ocean quite easily...

    Corrected.

40. 213: remove the parentheses around "S76 and S78"

    Corrected.

41. 215: Add comma after "Table 3"

    Figure 3 caption: What are the units? Also, do the number of measurements in each month change substantially?

    Units included in the caption of Fig3.

42. 237-239: move the phrase "which is about 0.06 PgC y−1 (mostly constant for each month in 2015)" after "over Australia" – as it reads now, it sounds like this refers to the annual posterior flux, not the fossil fuel.

    Corrected.

43. 258: change "are shown" to "is given"

    Corrected.

44. 273 & 275: do you mean "h and i" instead of "g and h"? Those are the two panels showing August & Sept.

    Corrected.

45. 293: "are shown" xxx Corrected.

46. 294-295: change to "strengthens to a posterior of -1.05 $\pm$ 0.11 PgC y$-$1".

    Corrected.

47. 316: change to "to a posterior of 0.47 $\pm$ 0.05 PgC y$-$1".

    Corrected, but warm temperate results were modified.

48. 331: "suggests also an"

    Corrected.

49. 334: "strengthened"

50. Figure 8a: the title should read "Darwin" not "Wollongong", I think.

    Corrected.

51. 374: add a comma after "sites"

    Corrected.

52. 380: "OCO-2"

    Corrected.

53. 400: "Section 4" – the "s" should be capitalized because it is a proper name. Corrected.

54. 412: add a comma before "which"

55. 417: switch "which" to "that"

    We modified the first paragraph of Section 4 (Discussion), so changes in line 412 and 417 were not necessary.

56. Figure 11 caption, line 4: correct "ensemble monthly the mean"

    Figure 11 was replaced by Fig.14 (Section 4).

We have included a marked-up manuscript version (Latexdiff) showing all the changes made in the main document.

**References**

[revised manuscript text omitted]

Ziehn, T., Nickless, A., Rayner, P. J., Law, R. M., Roff, G., and Fraser, P. (2014). Greenhouse gas network design using backward lagrangian particle dispersion modelling - part 1: Methodology and australian test case. *Atmospheric Chemistry and Physics*, 14(17):9363–9378.

---

## Author Response (AR2)

**Response to Editor' Comments**

Yohanna Villalobos Cortés

October 4, 2021

On behalf all co-authors, I would like thank the editor Abhishek Chatterjee for reviewing and provide valuable feedback of the findings found in our manuscript. Below the editor will find detailed answers to his questions. Answers to editor comments are shown in blue font.

(a) It is certainly a great idea to tap into the inversion estimates from the OCO-2 MIP repository. I just want to make sure that you have made all the inversion modelers aware that you are using their product and following the data fair use policy and guidelines. In case you have, then please make sure to capture this in the acknowledgement section. In case you missed this, then please make sure to check in with Dr. Andrew Jacobson at NOAA about the fair use policy

Dr. Andrew Jacobson and the other modellers are aware that we are using their product. I emailed them on May 31, 2021 asking about the MIP data fair use policy and guidelines. We certainly overlooked inclusion of the contributors to the manuscript. This was amended in the acknowledgement section.

(b) Along the same lines, please note that LoFi is not a global inversion product as the other products listed in Tables 3 (or Table 4). You can find details about the LoFI system here - https://acp.copernicus.org/articles/21/9609/2021/. It is primarily a bias correction technique for surface fluxes using the AGR information from in situ sites. My strong recommendation would be to keep the global inversion products that are actually based on Bayesian inference techniques, thus making it a more apples-to-apples comparison. In case you do decide to retain LoFI, then the fact that it is not based on Bayesian inference (as the other global inversions) should be made clear for the benefit of the reader.

We thank the editor for the clarification with the LoFi product. We thought that LoFi was part of the ensemble mean of MIP product because it was included within the global attribution description of the netCDFile provided by MIP. I contacted Dr. Andrew Jacobson and he said that was error in the file.

After removing the LoFi inversion flux product from the ensemble mean of the OCO-2 (LNLG) MIP, we found that the annual carbon sink estimate for Australia was larger (-0.23 $\pm$ 0.12 PgC y$^{-1}$) than the previous estimate (-0.17 $\pm$ 0.26 PgC y$^{-1}$). Contrary to these results, the Australian annual ensemble mean derived by in situ MIP remain almost the same (0.20 $\pm$ 0.22 PgC y$^{-1}$). As consequence of this changes, Figs. 14 and 15 were updated accordingly. Please see more details in the discussion section of the manuscript.

Table 4 and Figs. I1 and I2 (Appendix I) were also updated because we found a small error in the code that aggregate the MIP flux estimates across Australia. We had aggregated the fluxes across the whole CMAQ domain instead of masking only Australia. The new estimates do not impact the main results presented in the discussion and idea we want to convey to the readers.

(c) Kindly take note that there are a few spelling mistakes throughout the manuscript (for e.g., see Table 4 headings, or in the body of the table). Please look through the manuscript carefully and revise, as necessary.

We have now conducted a careful review of the manuscript and amended all spelling mistakes found. These changes can be seen in the latex diff document attached to the manuscript.